# Provably Efficient Exploration in Inverse Constrained Reinforcement Learning

**Bo Yue**[1]  **Jian Li**[2]  **Guiliang Liu**[1][*]

## Abstract

Optimizing objective functions subject to constraints is fundamental in many real-world applications. However, these constraints are often not readily defined and must be inferred from expert agent behaviors, a problem known as Inverse Constraint Inference. Inverse Constrained Reinforcement Learning (ICRL) is a common solver for recovering feasible constraints in complex environments, relying on training samples collected from interactive environments. However, the efficacy and efficiency of current sampling strategies remain unclear. We propose a strategic exploration framework for sampling with guaranteed efficiency to bridge this gap. By defining the feasible cost set for ICRL problems, we analyze how estimation errors in transition dynamics and the expert policy influence the feasibility of inferred constraints. Based on this analysis, we introduce two exploratory algorithms to achieve efficient constraint inference via 1) dynamically reducing the bounded aggregate error of cost estimations or 2) strategically constraining the exploration policy around plausibly optimal ones. Both algorithms are theoretically grounded with tractable sample complexity, and their performance is validated empirically across various environments.

## 1. Introduction

Constrained Reinforcement Learning (CRL) tackles sequential decision-making problems with safety constraints and has achieved considerable success in various safety-critical applications (Gu et al., 2022). However, in many real-world environments, such as robot control (García & Shafie, 2020; Thomas et al., 2021) and autonomous driving (Krasowski et al., 2020), specifying the exact constraint that can con-

sistently guarantee safe control is challenging, which is further exacerbated when the ground-truth constraint is time-varying and context-dependent.

Instead of utilizing a pre-defined constraint, an alternative approach, Inverse Constrained Reinforcement Learning (ICRL) (Malik et al., 2021; Liu et al., 2025), seeks to learn constraint signals from the demonstrations of expert agents and imitate their behaviors by adhering to the inferred constraints. ICRL effectively incorporates expert experience into the online CRL paradigm and thus better explains how expert agents optimize cumulative rewards under their empirical constraints. Under this framework, existing ICRL algorithms often assume the presence of a known dynamic model (Scobee & Sastry, 2020; McPherson et al., 2021), or a generative transition model that responds to queries for any state-action pair (Papadimitriou et al., 2023; Liu et al., 2023). However, this setting has a considerable gap with real-world scenarios where such global transition models are often unavailable or even time-varying. In such cases, agents have to physically navigate to new states to learn about transition dynamics through exploration.

To mitigate the gap, some recent studies (Malik et al., 2021; Baert et al., 2023) explicitly maximized the policy entropy throughout the learning process, yielding soft-optimal policy representations that favor less-selected actions. However, this strategy often induces overly conservative constraints that forbid any behavior not present in the expert demonstrations. Moreover, such an uncertainty-driven exploration ignores the potential estimation errors in dynamic models and expert policies. To date, a theoretical framework is still lacking to demonstrate how well such exploration approaches facilitate the accurate estimation of constraints.

This paper introduces a strategic exploration framework for sampling to solve ICRL problems with guaranteed efficiency. Recognizing the inherent challenge of pinpointing the exact constraint that expert agents adhere to in their demonstrations (Ng et al., 2000), the objective of our framework is to recover a *set of feasible constraints*, rather than to specify a unique constraint with pre-defined heuristics or restrictions. This approach has the advantage of analyzing the intrinsic sample complexity of ICRL problems only, without being obfuscated by other factors (Lazzati et al., 2024b). By representing constraints with reward advantages, we bound

---

[*]Corresponding Author [1]School of Data Science, The Chinese University of Hong Kong, Shenzhen [2]Stony Brook University, New York. Correspondence to: Guiliang Liu <liuguiliang@cuhk.edu.cn>.

*Proceedings of the 42nd International Conference on Machine Learning*, Vancouver, Canada. PMLR 267, 2025. Copyright 2025 by the author(s).

estimation errors for feasible cost functions to the discrepancy between estimated and ground-truth ones regarding dynamics and the expert policy. Based on this quantifiable measure of estimation errors, a tractable upper bound for sample complexity can be derived.

Under our framework, we design two strategic exploration algorithms for solving ICRL problems: 1) a Bounded Error Aggregate Reduction (BEAR) algorithm, which guides the exploration policy to minimize the upper bound of cost estimation errors, and 2) a Policy-Constrained Strategic Exploration (PCSE) algorithm, which reduces the estimation error by selecting an exploration policy from a set of candidate policies. This collection of policies is rigorously established to encompass the optimal policy, thereby promising to accelerate the training process significantly. We provide a rigorous sample complexity analysis for both algorithms, furnishing deeper insights into their training efficiency.

To empirically study how well our method captures the accurate constraint, we conduct evaluations under different environments. The experimental results show that PCSE significantly outperforms other exploration strategies and applies to continuous environments.

## 2. Related Work

This section reviews the previous works that are most closely related to ours. Appendix B provides further discussions.

**Exploration in Inverse Reinforcement Learning (IRL).** Compared with the exploration strategies in RL for forward control (Amin et al., 2021; Ladosz et al., 2022), the exploration algorithms in IRL have relatively limited studies. Balakrishnan et al. (2020) utilized Bayesian optimization to identify multiple IRL solutions by efficiently exploring the reward function space. To learn a transferable reward function, Metelli et al. (2021) introduced an active sampling methodology that targets the most informative regions with a generative model to facilitate effective approximations of the transition model and the expert policy. A subsequent research (Lindner et al., 2022) expanded this concept to finite-horizon MDPs with non-stationary policies, crafting innovative strategies to accelerate the exploration process. To better quantify the precision of recovered feasible rewards, Metelli et al. (2023) recently provided a lower bound on the sample complexity for estimating the feasible reward set in the finite-horizon setting with a generative model. However, these methods study only reward functions under a regular MDP without considering the safety of control or the constraints in the environment.

**Inverse Constrained Reinforcement Learning (ICRL).** Inverse Constraint Learning (ICL) extended the IRL paradigm to account for safety issues. This line of research encompasses several notable works. Hugessen et al. (2024)

simplify inverse constraint inference to a variant of IRL by jointly identifying the cost function and Lagrange parameters, which are assumed to form a convex cone. Kim et al. (2024) generalized the IRL framework to infer tight safety constraints from multi-task expert demonstrations, offering both performance and constraint satisfaction guarantees. Building on this work, Qadri et al. (2025) revealed that inverse constraint inference recovers a dynamic-conditioned and failure-inevitable constraint set, rather than the original ground-truth constraint set. Another line of research in classical ICRL algorithms updated the cost functions by maximizing the likelihood of generating the expert dataset under the maximum (causal) entropy framework (Scobee & Sastry, 2020). This method has been scaled to both discrete (McPherson et al., 2021) and continuous state-action spaces (Malik et al., 2021; Baert et al., 2023; Liu et al., 2023; Qiao et al., 2023; Xu & Liu, 2024a;b; Zhao et al., 2025). To improve training efficiency, recent studies combined ICRL with bi-level optimization techniques (Liu & Zhu, 2022; Gaurav et al., 2023). However, neither have these ICRL methods investigated exploration strategies based on estimation errors nor conducted theoretical studies on the sample efficiency of their algorithms.

## 3. Preliminaries

**Notation.** Let $\mathcal{X}$ and $\mathcal{Y}$ be two sets. $\mathcal{Y}^{\mathcal{X}}$ represents the set of functions $f : \mathcal{X} \rightarrow \mathcal{Y}$. Let $\Delta^{\mathcal{X}}$ denote the set of probability measures over $\mathcal{X}$. Let $\Delta_{\mathcal{Y}}^{\mathcal{X}}$ denote the set of functions: $\mathcal{Y} \rightarrow \Delta^{\mathcal{X}}$. We define the vector infinity norm as $||a||_\infty = \max_i |a_i|$ and the matrix infinity norm as $||A||_\infty = \max_i \sum_j |A_{ij}|$. We define $\min_{x \in \mathcal{X}}^+ f(x)$ to return the minimum positive value of $f$ over $\mathcal{X}$. The complete notation is reported in Appendix A.

**Constrained Markov Decision Process (CMDP).** We model the environment as a stationary CMDP $\mathcal{M} \cup c :=$ $(\mathcal{S}, \mathcal{A}, P_{\mathcal{T}}, r, c, \epsilon, \mu_0, \gamma)$, where $\mathcal{S}$ and $\mathcal{A}$ are the finite state and action spaces, with each cardinality denoted as $S = |\mathcal{S}|$ and $A = |\mathcal{A}|$; $P_{\mathcal{T}}(s'|s,a) \in \Delta_{\mathcal{S} \times \mathcal{A}}^{\mathcal{S}}$ defines the transition distribution; $r \in [0, R_{\max}]^{\mathcal{S} \times \mathcal{A}}$ and $c \in [0, C_{\max}]^{\mathcal{S} \times \mathcal{A}}$ denote the reward and cost functions; $\epsilon$ defines the threshold (budget) of the constraint; $\mu_0 \in \Delta^{\mathcal{S}}$ denotes the initial state distribution; $\gamma \in [0, 1)$ is the discount factor. $\mathcal{M}$ denotes the CMDP without knowing the cost (i.e., CMDP\$\backslash c$). The agent's behavior is modeled by a policy $\pi \in \Delta_{\mathcal{S}}^{\mathcal{A}}$. $\Pi_{\mathcal{M} \cup c}^*$ denotes the set of all optimal policies for CMDP $\mathcal{M} \cup c$. The expert policy $\pi^E$ is optimal in the sense that $\pi^E$ maximizes the rewards while adhering to constraints, i.e., $\pi^E \in \Pi_{\mathcal{M} \cup c}^*$. Let $f \in \mathbb{R}^S$ and $g \in \mathbb{R}^{S \times A}$, we slightly abuse $P_{\mathcal{T}}$ and $\pi$ as operators: $(P_{\mathcal{T}} f)(s,a) = \sum_{s' \in \mathcal{S}} P_{\mathcal{T}}(s'|s,a) f(s')$ and $(\pi g)(s) = \sum_{a \in \mathcal{A}} \pi(a|s) g(s,a)$. We define the occupancy measure as $\rho_{\mathcal{M}}^\pi(s,a) = (1-\gamma) \sum_{t=0}^{\infty} \gamma^t \mathbb{P}_{\mu_0}^\pi(s_t = s, a_t = a)$ where $\mathbb{P}_{\mu_0}^\pi$ denotes the probability at $(s,a)$ at timestep

$t$ under the policy $\pi$ and the initial distribution $\mu_0$. We focus on a discrete finite state-action space within an infinite planning horizon in this work.

**Constrained Reinforcement Learning (CRL).** Within a CMDP, CRL learns a policy $\pi$ that maximizes the discounted cumulative rewards subject to a known constraint:

$$\arg \max_{\pi} \ \mathbb{E}_{\mu_0, \pi, p_{\mathcal{T}}} \Big[ \sum_{t=0}^{\infty} \gamma^t r(s_t, a_t) \Big], \qquad (1)$$

$$\text{s.t.} \ \ \mathbb{E}_{\mu_0, \pi, p_{\mathcal{T}}} \Big[ \sum_{t=0}^{\infty} \gamma^t c(s_t, a_t) \Big] \leq \epsilon, \qquad (2)$$

where $\epsilon = 0$ indicates a hard constraint and $\epsilon > 0$ represents a soft constraint since $c$ is non-negative.

**Value and advantage functions.** We distinguish two cases where the first superscript $r$ or $c$ specifies the actual rewards or costs evaluated and the subscript $\mathcal{M}$ or $\mathcal{M} \cup c$ specifies the environment. We denote the reward action-value function as $Q_{\mathcal{M}}^{r,\pi}(s, a) = \mathbb{E}_{\pi, P_{\mathcal{T}}} \left[ \sum_{t=0}^{\infty} \gamma^t r(s_t, a_t) | s_0 = s, a_0 = a \right]$, and the reward advantage function as $A_{\mathcal{M}}^{r,\pi}(s, a) = Q_{\mathcal{M}}^{r,\pi}(s, a) - V_{\mathcal{M}}^{r,\pi}(s)$, where the reward state-value function $V_{\mathcal{M}}^{r,\pi}(s) = \mathbb{E}_{\pi}[Q_{\mathcal{M}}^{r,\pi}(s, a)]$. Likewise, we denote the cost action-value function as $Q_{\mathcal{M} \cup c}^{c,\pi}(s, a) = \mathbb{E}_{\pi, P_{\mathcal{T}}} \left[ \sum_{t=0}^{\infty} \gamma^t c(s_t, a_t) | s_0 = s, a_0 = a \right]$ and the cost state-value function as $V_{\mathcal{M} \cup c}^{c,\pi}(s) = \mathbb{E}_{\pi}[Q_{\mathcal{M} \cup c}^{c,\pi}(s, a)]$.

# 4. Learning Feasible Constraints

This section formally defines the feasible cost set and investigates how estimation errors of the inferred cost functions can be bounded by imperfections in the estimates of both transition dynamics and the expert policy.

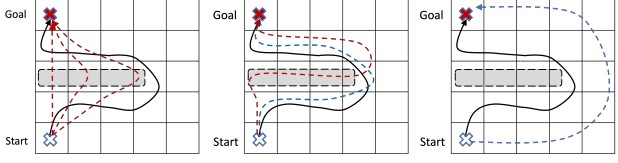

*Figure 1.* Trajectories of the expert policy (black) and exploratory policies (red and blue) in a Gridworld. The constraint (gray) is not observable. On the left, exploratory policies reach the goal in shorter paths and thus have larger rewards. In the middle, the rewards of exploratory policies are equal to the rewards of the expert policy. Their trajectories can either overlap with the constraint (red) or simply mismatch the expert's (blue). On the right, exploratory policies result in longer paths that gain fewer rewards.

## 4.1. Feasible Costs in CMDP

Since the expert policy maximizes rewards within certain constraints, two key insights emerge 1) if a policy achieves

higher rewards than the expert policy (shorter path in Figure 1, left), the underlying constraints *must be violated*, and we can detect unsafe state-action pairs by examining these infeasible trajectories; 2) if a policy achieves the same or lower rewards than the expert policy (equal or longer path in Figure 1, middle & right), this suggests an absence of notable constraint-violating actions, implying that the underlying constraints *may or may not be violated.* To effectively prevent overfitting and mitigate the combinatorial explosion of the constraint space, ICRL focuses on identifying the *minimal* set of constraints necessary to explain expert behaviors (Scobee & Sastry, 2020). In this sense, only the first insight is utilized to expand the cost set.

**Lemma 4.1.** *Suppose the expert policy $\pi^E$ of a CMDP $\mathcal{M} \cup c$ is known, and the current state is $s$. Let $\mathfrak{A}^E(s)$ denote the set containing all expert actions at state $s$, i.e., $\mathfrak{A}^E(s) = \{a \in \mathcal{A} \mid \pi^E(a|s) > 0\}$. Then, at least one of the following two conditions must be satisfied: 1) the cost function satisfies $\mathbb{E}_{\mu_0, \pi^E, P_{\mathcal{T}}} \left[ \sum_{t=0}^{\infty} \gamma^t c(s_t, a_t) \right] = \epsilon$;*
*2) $\forall a' \in \mathcal{A} \backslash \mathfrak{A}^E(s), A_{\mathcal{M}}^{r,\pi^E}(s, a') \leq 0$.*

Intuitively, if there is some unused constraint budget and a better action, there must exist a chance for policy improvement, which violates the optimality of the expert policy. The above lemma shows that cumulative costs of the expert policy must *reach the threshold*, i.e., use all the budget, if there exists a non-expert action yielding greater rewards than the expert policy (the first condition must be satisfied if the second condition is not satisfied). Thus, enforcing that any higher-reward action incurs greater costs than the expert policy suffices to establish a constraint-violation condition.

Let $\mathcal{Q}_c = \{(s, a) | Q_{\mathcal{M} \cup c}^{c,\pi^E}(s, a) - V_{\mathcal{M} \cup c}^{c,\pi^E}(s) > 0\}$ denote the set of state-action pairs with higher costs than the expert, given a cost function $c$. In scenarios with hard constraints, it simplifies to: $\mathcal{Q}_c = \{(s, a) | c(s, a) > 0\}$. We formally formulate the ICRL problem (Malik et al., 2021) as follows.

**Definition 4.2.** An ICRL problem is a pair $\mathfrak{P} = (\mathcal{M}, \pi^E)$. A cost function $c \in [0, C_{\max}]^{\mathcal{S} \times \mathcal{A}}$ is feasible for $\mathfrak{P}$ if $\pi^E$ is an optimal policy for the CMDP $\mathcal{M} \cup c$, i.e., $\pi^E \in \Pi_{\mathcal{M} \cup c}^*$. Let $\mathcal{F}_{\mathfrak{P}} = \{c | \pi^E \in \Pi_{\mathcal{M} \cup c}^*\}$ denote a general set of feasible cost functions. ICRL problem seeks to recover the minimal set of feasible cost functions for $\mathfrak{P}$, named feasible cost set that satisfies $\mathcal{C}_{\mathfrak{P}} = \{c^* | c^* = \arg \min_{c \in \mathcal{F}_{\mathfrak{P}}} |\mathcal{Q}_c|\}$.

We have defined the Q-function for non-expert actions. However, for stochastic expert actions under soft constraints, we only know that $\mathbb{E}_{a' \sim \pi^E}[Q_{\mathcal{M} \cup c}^{c,\pi^E}(s, a')] = V_{\mathcal{M} \cup c}^{c,\pi^E}(s) \geq 0$. To determine the exact value of the Q-function for expert actions, the following assumption is required.

**Assumption 4.3.** *(i) The expert policy $\pi^E$ is optimal w.r.t rewards among all safe policies;*
*(ii) The expert policy $\pi^E$ is deterministic for soft constraints.*

Based on these findings, we establish the feasible cost set.

**Lemma 4.4.** *(Feasible Cost Set Implicit). Under Assumption 4.3, $c$ is a feasible cost function for an ICRL problem $\mathfrak{P}$, i.e., $c \in \mathcal{C}_\mathfrak{P}$ if and only if $\forall (s,a) \in \mathcal{S} \times \mathcal{A}$:*

*(i) If $\pi^E(a|s) > 0$, i.e., $(s,a)$ follows the expert policy:*

$$Q^{c,\pi^E}_{\mathcal{M} \cup c}(s,a) - V^{c,\pi^E}_{\mathcal{M} \cup c}(s) = 0. \tag{3}$$

*(ii) If $\pi^E(a|s) = 0$ and $A^{r,\pi^E}_{\mathcal{M}}(s,a) > 0$, i.e., $(s,a)$ violates the constraint:*

$$Q^{c,\pi^E}_{\mathcal{M} \cup c}(s,a) - V^{c,\pi^E}_{\mathcal{M} \cup c}(s) > 0. \tag{4}$$

*(iii) If $\pi^E(a|s) = 0$ and $A^{r,\pi^E}_{\mathcal{M}}(s,a) \le 0$, i.e., $(s,a)$ is in the non-critical region:*

$$Q^{c,\pi^E}_{\mathcal{M} \cup c}(s,a) - V^{c,\pi^E}_{\mathcal{M} \cup c}(s) \le 0. \tag{5}$$

**Lemma 4.5.** *(Feasible Cost Set Explicit). Under Assumption 4.3, $c$ is a feasible cost function for an ICRL problem $\mathfrak{P}$, if and only if there exists $\zeta \in \mathbb{R}^{\mathcal{S} \times \mathcal{A}}_{>0}$ and $V^c \in \mathbb{R}^{\mathcal{S}}_{\ge 0}$:*

$$c = A^{r,\pi^E}_{\mathcal{M}} \zeta + (E - \gamma P_\mathcal{T}) V^c, \tag{6}$$

*where $E : \mathbb{R}^{\mathcal{S}} \to \mathbb{R}^{\mathcal{S} \times \mathcal{A}}$ is the expansion operator that satisfies $(Ef)(s,a) = f(s)$. Furthermore, $\|V^c(s)\|_\infty \le C_{\max}/(1-\gamma)$ and $\|\zeta\|_\infty = C_{\max}/\max^+_{(s,a)} |A^{r,\pi^E}_{\mathcal{M}}|$.*

Intuitively, the first term in (6) penalizes constraint-violating actions that deviate from the expert's policy yet achieve higher rewards (i.e., $A^{r,\pi^E}_{\mathcal{M}}(s,a) > 0$). This penalty ensures these actions violate the constraint in (2). The second term $V^c \in \mathbb{R}^{\mathcal{S}}$ can be interpreted as a cost-shaping operator that plays the role of translating the Q-function values by a fixed quantity. By utilizing the Bellman equation, we obtain that $V^c(s) = 0$ for hard constraints and $V^c(s) = V^{c,\pi^E}_{\mathcal{M} \cup c}(s)$ for soft constraints. Proofs of the above and the following theoretical results can be found in Appendix C.

## 4.2. Estimating Transition Dynamics and Expert Policy

Recall that our primary objective is to minimize the estimation errors of feasible cost functions. To obtain this error, we first introduce how we estimate the transition dynamics and the expert policy. We consider a model-based setting where the agent strategically explores the environment to learn the transition dynamics and the expert policy. We record the returns from querying a state-action pair $(s,a)$ by observing a next state $s' \sim P(\cdot|s,a)$, and the preferences of expert agents $a^E \sim \pi^E(\cdot|s)$ in each visited state. At iteration $k$, we denote by $n_k(s,a,s')$ the number of times we observe the transition tuple $(s,a,s')$. We further denote $n_k(s,a) = \sum_{s' \in \mathcal{S}} n_k(s,a,s')$. At iteration $k$, we denote by $n^E_k(s,a)$ the number of times we observe action $a$ as

an expert decision at state $s$. We further denote $n^E_k(s) = \sum_{a \in \mathcal{A}} n^E_k(s,a)$. We define four cumulative counts from iteration 1 to $k$ as $N_k(s,a,s') = \sum_{j=1}^{k} n_j(s,a,s')$ and $N_k(s,a) = \sum_{j=1}^{k} n_j(s,a)$, $N^E_k(s,a) = \sum_{j=1}^{k} n^E_j(s,a)$ and $N^E_k(s) = \sum_{j=1}^{k} n^E_j(s)$. Eventually, the transition model and the expert policy for a state-action pair at iteration $k$ are estimated as:

$$\widehat{P_\mathcal{T}}_k(s'|s,a) = \frac{N_k(s,a,s')}{N^+_k(s,a)}, \quad \widehat{\pi}^E_k(a|s) = \frac{N^E_k(s,a)}{N^{E^+}_k(s)}, \tag{7}$$

where $x^+ = \max\{1, x\}$.

## 4.3. Error Propagation

Building on the above estimations and the definition of cost functions, we obtain a set of *estimated* feasible cost functions. Next, we investigate the estimation error for the feasible cost function and analyze its underlying sources.

**Lemma 4.6.** *(Error Propagation). Let $\mathfrak{P} = (\mathcal{M}, \pi^E)$ and $\widehat{\mathfrak{P}} = (\widehat{\mathcal{M}}, \widehat{\pi}^E)$ be two ICRL problems where $\widehat{\mathcal{M}} = (\mathcal{M} \backslash P_\mathcal{T}) \cup \widehat{P_\mathcal{T}}$. For any $c \in \mathcal{C}_\mathfrak{P}$ satisfying $c = A^{r,\pi^E}_{\mathcal{M}} \zeta + (E - \gamma P_\mathcal{T}) V^c$ and $c \in [0, C_{\max}]^{\mathcal{S} \times \mathcal{A}}$, there exists $\widehat{c} \in \mathcal{C}_{\widehat{\mathfrak{P}}}$ and $\widehat{c} \in [0, C_{\max}]^{\mathcal{S} \times \mathcal{A}}$, $\forall (s,a) \in \mathcal{S} \times \mathcal{A}$:*

$$|c - \widehat{c}|(s,a) \le \frac{2(\chi(s,a) + \chi)}{1 + (\chi(s,a) + \chi)/C_{\max}}, \tag{8}$$

*where $\chi = \max_{(s,a) \in \mathcal{S} \times \mathcal{A}} \chi(s,a)$ and*

$$\chi(s,a) = \gamma \left| (P_\mathcal{T} - \widehat{P_\mathcal{T}}) V^c \right|(s,a) + \left| A^{r,\pi^E}_{\mathcal{M}} - A^{r,\widehat{\pi}^E}_{\widehat{\mathcal{M}}} \right| \zeta(s,a).$$

$\chi(s,a)$ is the distance at $(s,a)$ between the ground-truth cost function and a pseudo-estimated cost function with the same $\zeta$ and $V^c$, but does not necessarily fall into $[0, C_{\max}]$. The first part of $\chi(s,a)$ reflects the estimation error of the transition model, while the second part depends on the estimation error of the advantage function, which can be further decomposed as follows:

**Lemma 4.7.** *Let $\mathfrak{P} = (\mathcal{M}, \pi^E)$ and $\widehat{\mathfrak{P}} = (\widehat{\mathcal{M}}, \widehat{\pi}^E)$ be two ICRL problems. Then, we have*

$$\left| A^{r,\pi}_{\mathcal{M}} - A^{r,\widehat{\pi}}_{\widehat{\mathcal{M}}} \right| \le \tag{9}$$

$$\frac{2\gamma}{1-\gamma} \left| (\widehat{P_\mathcal{T}} - P_\mathcal{T}) V^{r,\widehat{\pi}^E}_{\widehat{\mathcal{M}}} \right| + \frac{\gamma(1+\gamma)}{1-\gamma} \left| (\pi - \widehat{\pi}) P_\mathcal{T} V^{r,\pi^E}_{\mathcal{M}} \right|.$$

To relate these error sources to the sample size, we derive confidence intervals for the transition model and expert policy using the Hoeffding inequality (see Lemma C.5). We show that the true transition model and expert policy lie within these intervals with high probability. Based on these results, we derive an upper bound on the estimation errors of feasible cost functions and prove that this upper bound is guaranteed with high probability as follows:

**Lemma 4.8.** *Let $\delta \in (0,1)$, with probability at least $1 - \delta$, for any pair of cost functions $c \in \mathcal{C}_{\mathfrak{P}}$ and $\widehat{c}_k \in \mathcal{C}_{\widehat{\mathfrak{P}}_k}$ at iteration $k$, we have*

$$|c(s,a) - \widehat{c}_k(s,a)| \leq \mathcal{C}_k(s,a) = \quad (10)$$

$$\min \left\{ \frac{2\sigma \left( \sqrt{\frac{\ell_k(s,a)}{2N_k^+(s,a)}} + \max_{(s,a)} \sqrt{\frac{\ell_k(s,a)}{2N_k^+(s,a)}} \right)}{1 + \frac{\sigma}{C_{\max}} \left( \sqrt{\frac{\ell_k(s,a)}{2N_k^+(s,a)}} + \max_{(s,a)} \sqrt{\frac{\ell_k(s,a)}{2N_k^+(s,a)}} \right)}, C_{\max} \right\}.$$

*where $\sigma = \frac{\gamma C_{\max} \left( R_{\max}(3+\gamma)/\max^+ \left| A_{\mathcal{M}}^{r,\pi^E} \right| + (1-\gamma) \right)}{(1-\gamma)^2}$ and $\ell_k(s,a) = \log \left( \frac{36 SA(N_k^+(s,a))^2}{\delta} \right)$.*

Intuitively, as the sample size grows, $\mathcal{C}_k(s,a)$ decreases, progressively reducing the estimation errors of feasible cost functions towards zero. However, this error does not necessarily need to be infinitesimal for the estimated feasible cost functions to sufficiently explain the expert's behavior. Next, we define a Probably Approximately Correct (PAC) optimality criterion for the estimated cost. The estimated feasible set $\mathcal{C}_{\widehat{\mathfrak{P}}}$ is considered 'close' to the exact feasible set $\mathcal{C}_{\mathfrak{P}}$, if for every cost function $c \in \mathcal{C}_{\mathfrak{P}}$, there exists one estimated cost function $\widehat{c} \in \mathcal{C}_{\widehat{\mathfrak{P}}}$ such that their Q-functions diverge within $\varepsilon$, and vice versa.

**Definition 4.9.** (Optimality Criterion). Let $\mathcal{C}_{\mathfrak{P}}$ be the exact feasible set and $\mathcal{C}_{\widehat{\mathfrak{P}}}$ be the feasible set recovered after observing $n \geq 0$ samples collected in the source $\mathcal{M}$ and $\pi^E$. We say that an algorithm for ICRL is $(\varepsilon, \delta, n)$-correct if with probability at least $1 - \delta$, it holds that:

$$\inf_{\widehat{c} \in \mathcal{C}_{\widehat{\mathfrak{P}}}} \sup_{\pi^* \in \Pi^*_{\mathcal{M} \cup c}} \left| Q_{\mathcal{M} \cup c}^{c,\pi^*}(s_0,a) - Q_{\mathcal{M} \cup \widehat{c}}^{\widehat{c},\pi^*}(s_0,a) \right| \leq \varepsilon, \forall c \in \mathcal{C}_{\mathfrak{P}},$$

$$\inf_{c \in \mathcal{C}_{\mathfrak{P}}} \sup_{\widehat{\pi}^* \in \Pi^*_{\widehat{\mathcal{M}} \cup \widehat{c}}} \left| Q_{\mathcal{M} \cup c}^{c,\widehat{\pi}^*}(s_0,a) - Q_{\mathcal{M} \cup \widehat{c}}^{\widehat{c},\widehat{\pi}^*}(s_0,a) \right| \leq \varepsilon, \forall \widehat{c} \in \mathcal{C}_{\widehat{\mathfrak{P}}},$$

where $\pi^*$ is an optimal policy in $\mathcal{M} \cup c$ and $\widehat{\pi}^*$ is an optimal policy in $\widehat{\mathcal{M}} \cup \widehat{c}$.

The criterion ensures estimation errors of feasible cost functions do not compromise the optimality of the expert policy. The first condition ensures completeness, requiring the recovered feasible cost set to track every true cost function. The second condition guarantees that there exists a true cost function close to every recovered cost function. This prevents the recovery of an excessively large feasible set that would overly prioritize completeness.

# 5. Efficient Exploration for ICRL

In this section, we introduce two algorithms for efficient exploration, addressing the challenge of collecting high-quality samples through interactions with the environment, thereby increasing the accuracy of our cost set estimations.

Unlike most existing ICRL works (Papadimitriou et al., 2023; Liu et al., 2022a; Yue et al., 2025) that rely on a generative model for sample collection, our exploration strategy must determine *which* states need more frequent visits and *how* to traverse to them from the initial state. We propose a Bounded Error Aggregate Reduction algorithm (BEAR, Section 5.1) and a Policy-Constrained Strategic Exploration algorithm (PCSE, Section 5.2) for solving ICRL problems in Algorithm 1.

---

**Algorithm 1** BEAR and PCSE for ICRL in an unknown environment

---

**Input:** significance $\delta \in (0,1)$, target accuracy $\varepsilon$, maximum number of samples per iteration $n_{\max}$;
Initialize $k \leftarrow 0$, $\varepsilon_0 = \frac{1}{1-\gamma}$;
**while** $\varepsilon_k > \varepsilon$ **do**
    Solve RL problem defined by $\mathcal{M}^{\mathcal{C}_k}$ to obtain the exploration policy $\pi_k$;
    Solve optimization problem in (15) to obtain the exploration policy $\pi_k$;
    Explore with $\pi_k$ for $n_e$ episodes;
    For each episode, collect $n_{\max}$ samples from $\mathcal{S} \times \mathcal{A}$;
    Update accuracy $\varepsilon_{k+1} = $
        $\max_{(s,a) \in \mathcal{S} \times \mathcal{A}} \mathcal{C}_{k+1}(s,a)/(1-\gamma)$;
    Update accuracy $\varepsilon_{k+1} = $
        $\|\mu_0^T (I_{\mathcal{S} \times \mathcal{A}} - \gamma P_{\mathcal{T}}\pi)^{-1} \mathcal{C}_k\|_{\infty}$;
    Update $\widehat{\pi}_{k+1}^E$ and $\widehat{P_{\mathcal{T}}}_{k+1}$ in (7);
    $k \leftarrow k + 1$.
**end while**

---

## 5.1. Exploration via Reducing Bounded Errors

To fulfill the optimality criterion in Definition 4.9, we begin by relating it to the cost estimation error.

**Lemma 5.1.** *Let $e_k(s,a;\pi^*) = |Q_{\mathcal{M} \cup c}^{c,\pi^*}(s,a) - Q_{\mathcal{M} \cup \widehat{c}_k}^{c,\pi^*}(s,a)|$ define the optimality error of state-action pair $(s,a)$ at iteration $k$ for $\pi^* \in \Pi^*_{\mathcal{M} \cup c}$. We upper bound it as follows:*

$$\|e_k(s,a;\pi^*)\|_{\infty} \leq \|\mu_0^T (I_{\mathcal{S} \times \mathcal{A}} - \gamma P_{\mathcal{T}}\pi)^{-1} \mathcal{C}_k\|_{\infty}. \quad (11)$$

We show in the lemma below that the exploration algorithm converges (satisfies Definition 4.9) at iteration $k$ if either one of the following statements is satisfied:

**Lemma 5.2.** *Let $\mathcal{C}_{\mathfrak{P}}$ be the ground-truth feasible set and $\mathcal{C}_{\widehat{\mathfrak{P}}_k}$ be the recovered feasible set after $k$ iterations. The conditions of Definition 4.9 are satisfied if either of the following statements are satisfied:*

$$(1) \quad \frac{1}{1-\gamma} \max_{(s,a)} \mathcal{C}_k(s,a) \leq \varepsilon; \quad (12)$$

$$(2) \quad \max_{\pi \in \Pi^\dagger} \max_{\mu_0 \in \Delta^{\mathcal{S}}} \left| \mu_0^T (I_{\mathcal{S} \times \mathcal{A}} - \gamma P_{\mathcal{T}}\pi)^{-1} \mathcal{C}_k \right| \leq \varepsilon, \quad (13)$$

$$\Pi^{\dagger} = \left( \bigcap_{c \in \mathcal{C}_{\mathfrak{P}}} \Pi^{*}_{\mathcal{M} \cup c} \right) \cup \left( \bigcap_{\widehat{c} \in \mathcal{C}_{\widehat{\mathfrak{P}}_k}} \Pi^{*}_{\widehat{\mathcal{M}}_k \cup \widehat{c}_k} \right)$$

Based on (12), we introduce BEAR in Algorithm 1 (highlighted in teal), which derives the $k$-th iteration exploration policy $\pi_k$ by solving the RL problem where the reward function $r = \mathcal{C}_k$. In practice, any RL solver can be applied to determine the exploration policy. Next, we analyze the sample complexity of BEAR.

**Sample Complexity.** Due to stochasticity in the environmental dynamics, we employ pseudo-counts to calculate the number of visitations to state-action pairs during traversal induced by the exploration policy. Let $\eta_k^h(s, a|s_0), h \in [n_{\max}]$ be the probability of state-action pair $(s, a)$ reached in the $h$-th step following a policy $\pi_k \in \Pi_{\mathcal{M}^{c_k}}$ starting in state $s_0$. We can compute it recursively

$$\eta_k^0(s, a|s_0) := \pi_k(a|s)\mathbb{1}_{\{s=s_0\}},$$
$$\eta_k^{h+1}(s, a|s_0) := \sum_{a',s'} \pi_k(a|s)P_{\mathcal{T}}(s|s', a')\eta_k^h(s', a'|s_0),$$

**Definition 5.3.** (Pseudo-counts). We introduce the pseudo-counts of visiting a specific state-action pair $(s, a)$ after $k$ iterations as:

$$\bar{N}_k(s, a) = \mu_0 \sum_{h=1}^{n_{\max}} \sum_{i=1}^{k} \eta_i^h(s, a|s_0).$$

Similar to (7), we define $\bar{N}_k^+(s, a) = \max\{0, \bar{N}_k(s, a)\}$. The following lemma provides an upper bound on the actual error in terms of the error induced by pseudo-counts.

**Lemma 5.4.** *With probability at least $1 - \delta/2$, $\forall s, a, h, k \in \mathcal{S} \times \mathcal{A} \times [n_{\max}] \times \mathbb{N}^+$, we have:*

$$\min\left\{ \sigma\sqrt{\frac{\ell_k(s, a)}{2N_k^+(s, a)}}, C_{\max} \right\} \leq \check{\sigma}\sqrt{\frac{2\bar{\ell}_k(s, a)}{\bar{N}_k^+(s, a)}}, \quad (14)$$

*where $\bar{\ell}_k(s, a) = \log(36SA(\bar{N}_k^+(s, a))^2/\delta)$ and $\check{\sigma} = \max\{\sigma, \sqrt{2}C_{\max}\}$.*

Subsequently, we derive the sample complexity of BEAR:

**Theorem 5.5.** *(Sample Complexity of BEAR). If Algorithm BEAR terminates at iteration $K$ with the updated accuracy $\varepsilon_K$, then with probability at least $1 - \delta$, it fulfills Definition 4.9 with a number of samples upper bounded by*

$$n \leq \widetilde{\mathcal{O}} \left( \frac{\check{\sigma}^2(2C_{\max} - \varepsilon_K(1 - \gamma))^2}{(1 - \gamma)^2\varepsilon_K^2 C_{\max}^2} \right).$$

The above theorem has taken into account the sample complexity of the *RL phase*. In fact, further improvements can be made to enhance the algorithm's performance.

## 5.2. Exploration via Constraining Candidate Policies

The above exploration strategy has limitations, as it aims to minimize uncertainty across *all policies*, whereas it could be more effective by focusing on reducing uncertainty only for *potentially optimal policies*. To address this, we propose PCSE for ICRL in Algorithm 1 (highlighted in purple). At each iteration, we intentionally restrict the search to policies that yield a value function close to the estimated optimal one regarding both rewards and costs. This allows us to focus solely on plausibly optimal policies, and we formulate the optimization problem as follows:

$$\varepsilon_{k+1} = \sup_{\substack{\mu_0 \in \Delta^{\mathcal{S}} \\ \pi \in \Pi_k}} \mu_0^T(I_{\mathcal{S} \times \mathcal{A}} - \gamma P_{\mathcal{T}}\pi)\mathcal{C}_{k+1}, \quad (15)$$

$$\text{s.t. } \Pi_k = \Pi_k^c \cap \Pi_k^r,$$

$$\Pi_k^c = \left\{ \pi : \sup_{\mu_0 \in \Delta^{\mathcal{S}}} \mu_0^T \left( V_{\widehat{\mathcal{M}}_k \cup \widehat{c}_k}^{c,\pi} - V_{\widehat{\mathcal{M}}_k \cup \widehat{c}_k}^{c,*} \right) \leq 4\varepsilon_k + \epsilon \right\},$$

$$\Pi_k^r = \left\{ \pi : \inf_{\mu_0 \in \Delta^{\mathcal{S}}} \mu_0^T \left( V_{\widehat{\mathcal{M}}_k}^{r,\pi} - V_{\widehat{\mathcal{M}}_k}^{r,\widehat{\pi}_k^*} \right) \geq \mathfrak{R}_k \right\},$$

where $\mathfrak{R}_k = \frac{2\gamma R_{\max}}{(1-\gamma)^2}\|P_{\mathcal{T}} - \widehat{P_{\mathcal{T}}}_k\|_{\infty} + \frac{\gamma R_{\max}}{(1-\gamma)^2}\|(\pi^* - \widehat{\pi}_k^*)\|_{\infty}$.

The rationale in $\Pi_k$ can be attributed to two aspects: 1) $\Pi_k^c$ constrains exploration policies to visit states within an additional cost budget, thereby ensuring *resilience* to estimation error when searching for optimal policies; 2) $\Pi_k^r$ states that exploration policies should focus on states with potentially higher cumulative rewards, where possible constraints lie. As the estimation error decreases, the gap (i.e., $\mathfrak{R}_k$) also diminishes, eventually converging to zero, which ensures the *optimality* of constrained policies. We have shown in Appendix C.12 that optimal policies in basic environments are captured by $\Pi_k$.

To solve the optimization problem (15), we express its Lagrangian objective as $L(\rho_{\mathcal{M}}^{\pi}, \lambda) =$

$$- \langle \rho_{\mathcal{M}}^{\pi}, \mathcal{C}_{k+1} \rangle + \lambda_2 \left( (1 - \gamma)(V_{\widehat{\mathcal{M}}_k}^{r,\widehat{\pi}_k^*} + \mathfrak{R}_k) - \langle \rho_{\mathcal{M}}^{\pi}, r \rangle \right)$$
$$+ \lambda_1 \left( -(1 - \gamma)(V_{\widehat{\mathcal{M}}_k \cup \widehat{c}_k}^{c,*} + 4\varepsilon_k + 2\epsilon) + \langle \rho_{\mathcal{M}}^{\pi}, \widehat{c}_k \rangle \right),$$

where $\lambda = [\lambda_1, \lambda_2]^T$ records two Lagrangian multipliers. The dual problem of (15) can be defined as

$$\min_{\rho_{\mathcal{M}}^{\pi}} \max_{\lambda \geq 0} L(\rho_{\mathcal{M}}^{\pi}, \lambda). \quad (16)$$

To solve this dual problem, we assume that Slater's condition is fulfilled, and we follow the two-timescale stochastic approximation (Borkar & Konda, 1997; Konda & Tsitsiklis, 1999). The following two gradient steps alternated until convergence.

$$\rho_{\mathcal{M},k+1}^{\pi} = \rho_{\mathcal{M},k}^{\pi} - a_k(L_{\rho}'(\rho_{\mathcal{M},k}^{\pi}, \lambda_k) + W_k),$$

$$\lambda_{k+1} = \lambda_k + b_k(L'_\lambda(\rho^\pi_{\mathcal{M},k}, \lambda_k) + U_k),$$

where coefficients $a_k \ll b_k$, satisfying $\sum_k a_k = \sum b_k = \infty$, $\sum a_k^2 < \infty$ and $\sum b_k^2 < \infty$; $W_k$ and $U_k$ are two zero-mean noise sequences. Under this condition, the convergence is guaranteed in the limit (Borkar, 2009). At each iteration $k$, the exploration policy is calculated as: $\pi_k(a|s) = \rho^\pi_{\mathcal{M},k}(s,a)/\sum_a \rho^\pi_{\mathcal{M},k}(s,a)$.

**Sample Complexity.** The convergence condition for PCSE for ICRL, given by (13), determines its sample complexity. To present this result, we additionally define the cost advantage function as $A^{c,*}_{\widehat{\mathcal{M}}\cup\tilde{c}}(s,a) = Q^{c,*}_{\widehat{\mathcal{M}}\cup\tilde{c}}(s,a) - V^{*,c}_{\widehat{\mathcal{M}}\cup\tilde{c}}(s)$, in which $\tilde{c} \in \arg\min_{c\in\mathcal{C}_{\mathfrak{B}}} \max_{(s,a)\in\mathcal{S}\times\mathcal{A}} |c(s,a) - \widehat{c}_K(s,a)|$ is the cost function in the true feasible cost set $\mathcal{C}_{\mathfrak{B}}$ closest to the estimated cost function $\widehat{c}_K(s,a)$ at iteration $K$.

**Theorem 5.6.** *(Sample Complexity of PCSE). If Algorithm PCSE terminates at iteration $K$ with accuracy $\varepsilon_K$ and the accuracy of previous iteration is $\varepsilon_{K-1}$, then with probability at least $1 - \delta$, it fulfills Definition 4.9 with the number of samples upper bounded by*

$$n \leq \widetilde{\mathcal{O}}\Bigg( \min\Bigg\{ \widetilde{\mathcal{O}}\left( \frac{\check{\sigma}^2(2C_{\max} - \varepsilon_K(1-\gamma))^2}{(1-\gamma)^2\varepsilon_K^2 C_{\max}^2} \right), \\ \frac{\sigma^2(6\varepsilon_{K-1} + \epsilon)^2 SA}{\min_{(s,a)}\left(A^{c,*}_{\widehat{\mathcal{M}}\cup\tilde{c}}(s,a)\right)^2 \varepsilon_K^2} \Bigg\}\Bigg).$$

The first term matches the sample complexity of the BEAR strategy since the convergence of BEAR is stricter than PCSE, i.e., (13) is always satisfied if (12) is satisfied. As a result, the sample complexity of BEAR constitutes a lower bound for that of PCSE. The second term depends on the ratio $(6\varepsilon_{K-1} + \epsilon)/\varepsilon_K$ and the minimum cost advantage function $\min_{(s,a)} A^{c,*}_{\widehat{\mathcal{M}}\cup\tilde{c}}$. On one side, the ratio depends on both $n_{\max}$ and $n_e$. If the two values are high, the ratio is high because the accuracy reduces faster from iteration $K-1$ to $K$ with more collected samples. Otherwise, the ratio is small because the accuracy remain similar between two iterations. A smaller $\epsilon$, namely a tighter constraint, benefits the sample efficiency. On the other side, the cost advantage function $\min_{(s,a)} A^{c,*}_{\widehat{\mathcal{M}}\cup\tilde{c}}$ shows that the larger the suboptimality gap, the easier to infer the constraint.

# 6. Empirical Evaluation

We empirically compare our algorithms against other methods in both discrete and continuous environments, where the agent navigates from a starting location to a target location (receiving a positive reward) while satisfying the constraints. Our implementation of code for discrete environments is adapted from (Liu et al., 2023), and for continuous environments, it is adapted from (Lazcano et al., 2024).

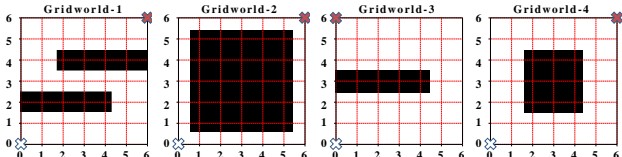

*Figure 2.* Four different Gridworld environments with white, red, and black markers indicating the starting, target, and constraint locations, respectively.

**Experiment Settings.** The evaluation metrics include: 1) *discounted cumulative rewards*, which measure the optimality of the learned policy; 2) *discounted cumulative costs*, which assess the safety of the learned policy; and 3) *Weighted Generalized Intersection over Union (WGIoU)* (refer to Appendix D.2 for details), which robustly evaluates the similarity between inferred cost functions and ground-truth cost functions.

**Comparison Methods.** We compare BEAR and PCSE with four other exploration strategies: random exploration, $\epsilon$-greedy exploration, maximum-entropy exploration, and upper confidence bound exploration.

## 6.1. Evaluation under Discrete Environments

Figure 2 illustrates four discrete environments, each characterized by distinct constraints. The expert policy is trained under ground-truth constraints, while two ICRL algorithms and four baselines operate without knowledge of these constraints. These environments are stochastic, with the environment executing a randomly sampled action with probability $p = 0.05$. Figure 3 and 5 (in Appendix) demonstrate the training process of three metrics for six exploration strategies in four Gridworld environments, along with the performance of expert policy (represented by the grey line). It can be shown that the performance of the optimal policy in $\mathcal{M} \cup \widehat{c}$ gradually converges to the performance of the optimal policy in $\mathcal{M} \cup c$. Also, we find that PCSE (represented by the red curve) converges the fastest among the six exploration strategies. In Gridworld-2 and Gridworld-4, WGIoU converges to a degree of similarity less than 1 (ground-truth). This is because some of the ground-truth constraints lie in non-critical regions, and ICRL infers the *minimal* set of constraints required to explain expert behavior. We demonstrate constraint learning processes of six exploration strategies for four Gridworlds in Appendix Figure 7, 8, 9, and 10, respectively. These learned constraints are recovered as visiting these states leads to higher cumulative rewards, while other unrecovered constraints do not impact the optimality of expert behaviors.

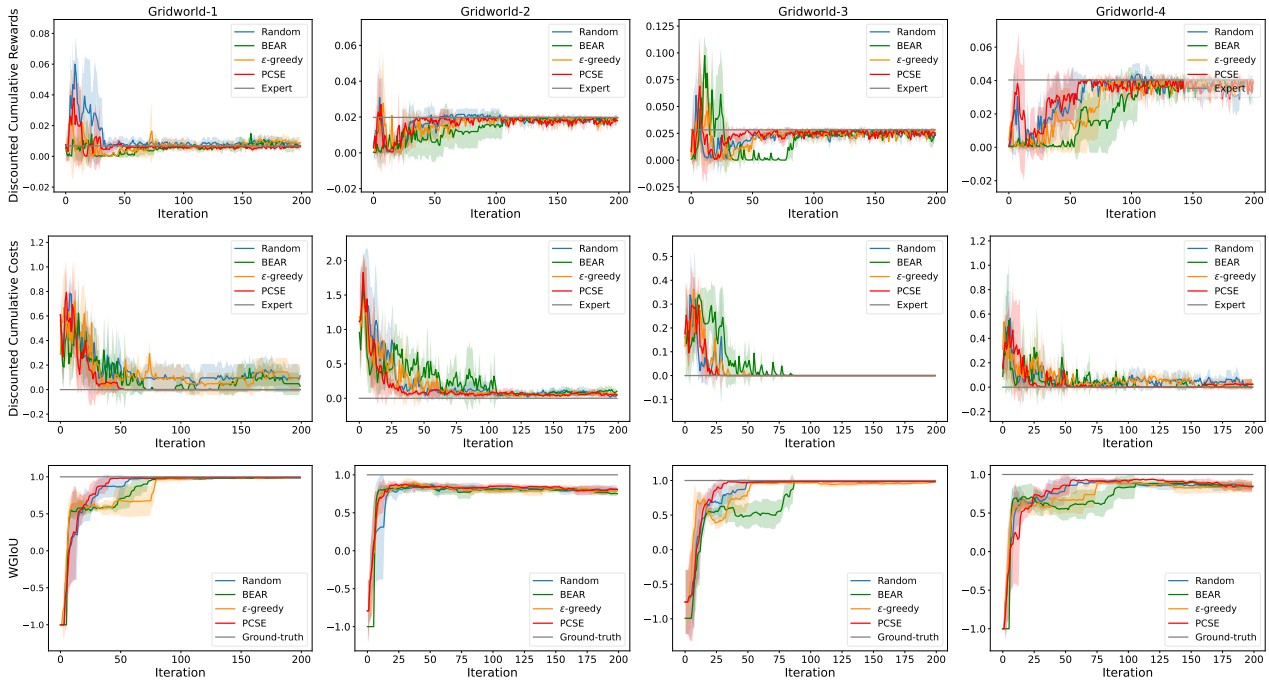

*Figure 3.* Training curves of discounted cumulative rewards (top), discounted cumulative costs (middle), and WGIoU (bottom) for four exploration strategies in four Gridworld environments, respectively.

## 6.2. Evaluation under Continuous Environments

Figure 4 (leftmost) shows the Point Maze environment, where the green agent has a continuous state space. The agent's goal is to reach the red ball inside the maze with pink walls. The environment is stochastic due to the noises imposed on the observed states. Figure 4 (middle left) demonstrates the inferred constraints (represented by blue dots) obtained through PCSE, with the center of the entire maze at $(0, 0)$. Figure 4 (middle right and rightmost) reports the discounted cumulative rewards and costs during training.

For all the experiments discussed above, please check Appendix D and E for more experimental details.

## 7. Conclusions

**Summary.** This paper introduces a strategically efficient exploration framework for ICRL problems. We conduct theoretical analysis to investigate the influence of estimation errors in expert policy and environmental dynamics on the estimation of feasible constraints. Building upon this, we propose two exploration algorithms, namely BEAR algorithm and PCSE. BEAR explores the environment to minimize the aggregated bounded error of cost estimations. Moreover, PCSE algorithm goes a step further by constraining the exploration policies to plausibly optimal ones, thus enhancing the overall sample efficiency. We provide tractable sample complexity analyses for both algorithms. To validate the ef-

fectiveness of our method, we perform empirical evaluations in various environments.

**Limitations and Future Work.** As is also a pressing challenge in the field, our method faces the limitation of struggling to scale to problems with large or continuous state spaces. This is due to the fact that our sample complexity is directly dependent on the size of the state space, and real-world problems often involve large or continuous spaces. Several future research directions warrant attention. First, extending the analysis to finite-horizon settings and deriving lower bounds for sample complexities would provide a more comprehensive understanding of the performance limits. Second, investigating the transferability of feasible constraints across different environments would be valuable in determining the generalizability of our approach. Additionally, relaxing the assumption that the expert policy is safe and reward optimal, either to a safe expert policy (not necessarily optimal) or offline expert demonstrations, would be an interesting direction for future work. We believe it is also valuable to investigate how the sub-optimality of expert agents influences inverse constraint inference and transferability. Finally, it is also intriguing if the hypothesis space of admissible constraint functions can be restricted in the first place, leading to more practical, scalable, and efficient constraint inference.

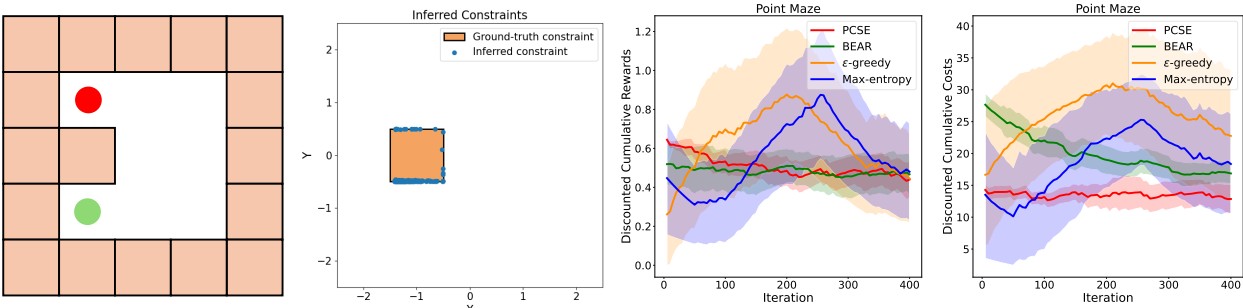

*Figure 4.* Point Maze environment (leftmost), inferred constraints (middle left), discounted cumulative rewards (middle right) and discounted cumulative costs (rightmost).

## 8. Discussions

**Set-recovery Framework.** In contrast to prior ICRL methods, our set-recovery framework defers the commitment to specific constraints until a later stage, thereby exposing the fundamental complexity of inverse constraint inference. Building on this perspective, we characterize two distinct modes of constraint selection for the downstream optimization procedure on a novel task. For hard constraints, all feasible constraints in the set are equivalent for a novel task, as the cost function value does not matter ($c(s, a) = 1$ and $c(s, a) = 2$ both prohibit $(s, a)$). Thus, any feasible constraint can be selected for transfer. For soft constraints, feasible constraints in the set differ for a novel task. The value of each cost function matters due to task differences in dynamics and rewards. Therefore, the generalizable learned constraints should come from the intersection of feasible sets from old and novel tasks. The selection criterion should depend on differences in dynamics and rewards.

**Affine Space Perspective.** Linear algebraic analyses provide a more rigorous and inherent framework for this set-recovery approach, making them valuable for investigating ICRL theories. For instance, by defining a subspace $\mathcal{U} = \text{im}(E - \gamma P_\mathcal{T})$, the cost functions within a feasible cost set become equivalent on the quotient space $\mathbb{R}^{\mathcal{S} \times \mathcal{A}} / \mathcal{U}$. Furthermore, we can measure the distance between the recovered and expert costs within this quotient space.

**Scaling to Practical Environments.** Sample complexity analysis has primarily focused on discrete state-action spaces (Agarwal et al., 2019). Existing algorithms for learning feasible sets (Metelli et al., 2023; Zhao et al., 2023; Lazzati et al., 2024a) struggle to scale effectively to problems with large or continuous state spaces. This limitation arises because their sample complexity depends directly on the size of the state space, and real-world problems frequently involve large or continuous spaces. Scaling feasible set learning to practical problems with large state spaces remains a pressing challenge in the field (Lazzati et al., 2024b).

One critical issue is the exploration challenge where algorithms need to specify where and how to explore at each iteration. Another key difficulty is the estimation of the ground-truth expert policy, which is hard to obtain in an online setting. A potential solution involves extracting the expert policy from offline datasets of expert demonstrations. However, these datasets often contain a mix of optimal and sub-optimal demonstrations, leading to sub-optimal expert policies. Addressing this issue could involve i) treating the dataset as noisy and applying robust learning algorithms designed to handle noisy demonstrations or ii) combining offline demonstrations with online fine-tuning, where feasible, to refine the learned policy. Finally, the scalability of learning in continuous spaces is frequently hindered by the curse of dimensionality. Dimensionality reduction techniques can mitigate this challenge by simplifying state and action representations while retaining the features essential for effective policy learning.

## Acknowledgments

This work is supported in part by Shenzhen Fundamental Research Program (General Program) under grant JCYJ20230807114202005, Guangdong-Shenzhen Joint Research Fund under grant 2023A1515110617, Guangdong Basic and Applied Basic Research Foundation under grant 2024A1515012103, and Guangdong Provincial Key Laboratory of Mathematical Foundations for Artificial Intelligence (2023B1212010001).

## Impact Statement

This paper presents work whose goal is to advance the field of Machine Learning. We do not see any direct negative consequences of our work.

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

# Appendix

## Table of Contents

# A. Notation and Symbols

In Table 1, we report the explicit definition of notation and symbols applied in our paper.

*Table 1.* Overview of notation and symbols

| Symbol | Name | Signature |
|---|---|---|
| $\mathcal{M}$ | CMDP without knowing the cost (CMDP$\backslash c$) | $(\mathcal{S}, \mathcal{A}, P_{\mathcal{T}}, r, \epsilon, \mu_0, \gamma)$ |
| $\mathcal{M} \cup c$ | CMDP | $(\mathcal{S}, \mathcal{A}, P_{\mathcal{T}}, r, c, \epsilon, \mu_0, \gamma)$ |
| $\mathcal{S}$ | State space | / |
| $\mathcal{A}$ | Action space | / |
| $P_{\mathcal{T}}$ | Transition dynamics | $\Delta_{\mathcal{S} \times \mathcal{A}}^{\mathcal{S}}$ |
| $s_0$ | Initial state | $\mathcal{S}$ |
| $\pi$ | Policy | $\Delta_{\mathcal{S}}^{\mathcal{A}}$ |
| $\pi^E$ | Expert policy | $\Delta_{\mathcal{S}}^{\mathcal{A}}$ |
| $r$ | Reward function | $[0, R_{\max}]^{\mathcal{S} \times \mathcal{A}}$ |
| $c$ | Cost function | $[0, C_{\max}]^{\mathcal{S} \times \mathcal{A}}$ |
| $\epsilon$ | Threshold of constraint or budget | $\mathbb{R}^{\mathcal{S}}$ |
| $V_{\mathcal{M}}^{r,\pi}$ | Reward state-value function for $r$ of $\pi$ in $\mathcal{M}$ | $\mathbb{R}^{\mathcal{S}}$ |
| $Q_{\mathcal{M}}^{r,\pi}$ | Reward action-value function for $r$ of $\pi$ in $\mathcal{M}$ | $\mathbb{R}^{\mathcal{S} \times \mathcal{A}}$ |
| $A_{\mathcal{M}}^{r,\pi}$ | Reward advantage function for $r$ of $\pi$ in $\mathcal{M}$ | $\mathbb{R}^{\mathcal{S} \times \mathcal{A}}$ |
| $V_{\mathcal{M} \cup c}^{c,\pi}$ | Cost state-value function for $c$ of $\pi$ in $\mathcal{M} \cup c$ | $\mathbb{R}^{\mathcal{S}}$ |
| $Q_{\mathcal{M} \cup c}^{c,\pi}$ | Cost action-value function for $c$ of $\pi$ in $\mathcal{M} \cup c$ | $\mathbb{R}^{\mathcal{S} \times \mathcal{A}}$ |
| $\mathcal{C}_{\mathfrak{P}}$ | Ground-truth feasible cost set | / |
| $\mathcal{C}_{\widehat{\mathfrak{P}}}$ | Recovered feasible cost set | / |
| $\eta_k^h(s, a\|s_0)$ | State action pair visitation frequencies | $\Delta^{\mathcal{S} \times \mathcal{A}}$ |
| $\rho_{\mathcal{M}}^{\pi}$ | Occupancy measure of $\pi$ in $\mathcal{M}$ | $\Delta^{\mathcal{S} \times \mathcal{A}}$ |
| $\varepsilon$ | Target accuracy | $\mathbb{R}^+$ |
| $\delta$ | Significance | $(0, 1)$ |
| $n_e$ | Number of exploration episodes | $\mathbb{N}^+$ |
| $E$ | Expansion operator | $\mathbb{R}^{\mathcal{S}} \to \mathbb{R}^{\mathcal{S} \times \mathcal{A}}$ |
| $I_{\mathcal{S} \times \mathcal{A}}$ | Identity matrix on $\mathcal{S} \times \mathcal{A}$ | / |
| $I_{\mathcal{S}}$ | Identity matrix on $\mathcal{S}$ | / |
| $[a]$ | Set that contains integers from 0 to $a$ | $\{0, 1, \ldots, a\}, a \in \mathbb{N}$ |

# B. Extra Related Works

**Sample Efficiency**. Sample-efficient algorithms have been explored across various RL directions, yielding significant advancements. To find the minimal structural assumptions that empower sample-efficient learning, (Jin et al., 2021) introduced the Bellman Eluder (BE) dimension and proposed a sample-efficient algorithm for problems with low BE dimension. (Liu et al., 2024) introduced a sample-efficient RL framework called Maximize to Explore, which reduces the computational cost and enhances compatibility. In the field of imitation learning, (Liu et al., 2022b) addressed both online and offline settings, proposing optimistic and pessimistic generative adversarial policy imitation algorithms with tractable regret bounds. In the realm of model-free RL, (Jin et al., 2018) developed a Q-learning algorithm with upper confidence bound exploration, achieving a regret bound of $\sqrt{T}$ in episodic MDPs. (Wachi et al., 2018) modeled state safety values using a Gaussian Process and proposed a more efficient approach to balance the trade-off between exploring the safety function, exploring the reward function, and exploiting knowledge to maximize rewards. In the context of (CRL), (Miryoosefi & Jin, 2022) bridged reward-free RL and CRL, providing sharp sample complexity results for CRL in tabular MDPs. Focusing on episodic finite-horizon Constrained MDPs (CMDPs), (Kalagarla et al., 2021) established a probably approximately correct guarantee on the number of episodes required to find a near-optimal policy, with a linear dependence on the state and action spaces and a quadratic dependence on the time horizon. From a meta-learning perspective, (Liu & Zhu, 2023) framed the problem of learning an expert's reward function and constraints from a few demonstrations as a bi-level optimization, introducing a provably efficient algorithm to learn a meta-prior over reward functions and constraints. In terms of sample

efficiency in IRL, (Lazzati et al., 2024a) redefines offline IRL by introducing the feasible reward set to address limited data coverage, proposing approaches to ensure inclusion monotonicity through pessimism. (Lazzati & Metelli, 2024) extends IRL to Utility Learning, introducing a framework for capturing agents' risk attitudes via utility functions. (Lazzati et al., 2024b) tackles scalability in online IRL by introducing reward compatibility and a state-space-independent algorithm for Linear MDPs, bridging IRL and reward-free exploration. For misspecification in IRL, (Skalse & Abate, 2023) provides a framework and tools to evaluate the robustness of standard IRL models (e.g., optimality, Boltzmann rationality) to misspecification, ensuring reliable inferences from real-world data. (Skalse & Abate, 2024) quantifies IRL's sensitivity to behavioral model inaccuracies, showing that even small misspecifications can result in significant errors in inferred reward functions.

**Constraint Inference.** Constraint learning in reinforcement learning has advanced significantly to address safety requirements and extend application scenarios. Chou et al. (2018) introduced a method to infer shared constraints across tasks using safe and unsafe trajectories, leveraging hit-and-run sampling and integer programming with theoretical guarantees. Kim & Oh (2022) proposed a sample-efficient RL method with CVaR constraints that addresses distributional shift via surrogate functions and trust-region constraints, achieving high returns and safety in complex tasks. To ensure stable convergence, Moskovitz et al. (2023) developed ReLOAD, which guarantees last-iterate convergence and overcomes limitations of gradient-based methods in CRL. For scenarios with unknown rewards and dynamics, Lindner et al. (2024) introduced a CMDP method that constructs a convex safe set from safe demonstrations, enabling task transferability and outperforming IRL-based approaches. Kim et al. (2024) extended IRL framework to infer tighter safety constraints from diverse expert demonstrations, addressing the ill-posed nature of constraint learning and enhancing multi-task generalization. These prior works either require multiple demonstrations across diverse environments or rely on additional settings to ensure the uniqueness of the recovered constraints. In contrast, our approach infers a feasible cost set encompassing all cost functions consistent with the provided demonstrations, eliminating reliance on additional information to address the inherent ill-posedness of inverse problems.

**Bayesian IRL.** In Bayesian IRL, posterior sampling over reward functions serves as a foundational mechanism to guide policy inference (Ramachandran & Amir, 2007). In contrast, PCSE shifts the focus from reward estimation to constraint-based reasoning by leveraging a structured policy set to guide exploration. Additionally, Balakrishnan et al. (2020) employed Bayesian optimization to facilitate active exploration of reward functions in IRL. PCSE extends this idea to the ICRL setting by actively exploring constraint structures rather than rewards, and crucially, it removes reliance on generative models. This design directly addresses scalability limitations highlighted by (Chan & van der Schaar, 2021), where traditional Bayesian IRL methods relying on Markov Chain Monte Carlo (MCMC) sampling exhibited poor scalability in high-dimensional state spaces.

# C. Proofs of Theoretical Results in the Main Paper

In this section, we provide detailed proofs of theoretical results in the main paper.

### C.1. Proof of Lemma 4.1

*Proof.* If neither case happens, i.e., $\mathbb{E}_{\mu_0, \pi^E, P_\mathcal{T}}\left[\sum_{t=0}^\infty \gamma^t c(s_t, a_t)\right] < \epsilon$ and $\exists\, a' \in \mathcal{A}$ that satisfies both $a' \notin \mathfrak{A}^E(s)$ and $A_{\mathcal{M}}^{r, \pi^E}(s, a') > 0$, we can always construct a new policy, which only differs from the expert policy $\pi^E$ in state $s$, as $\pi'(a|s) = \begin{cases} \theta & , a = a' \\ 1 - \theta, & a \sim \pi^E \end{cases}$. There must $\exists\, \theta \in (0, 1]$ that uses some (or all) of the leftover budget $\tau = \epsilon - \mathbb{E}_{\mu_0, \pi^E, P_\mathcal{T}}\left[\sum_{t=0}^\infty \gamma^t c(s_t, a_t)\right]$ while having a larger cumulative reward, which makes $\pi^E$ not an optimal policy. This makes a contradiction.

The existence of such $\theta$ can be proved as follows. By recursively using the Bellman Equation, we can obtain

$$\mathbb{E}_{\mu_0}\left[V_{\mathcal{M} \cup c}^{c, \pi^E}(s_0)\right] = \alpha(P_\mathcal{T}, \pi^E, \gamma, c) + \beta(P_\mathcal{T}, \pi^E, \gamma, c) \cdot \mathbb{E}_{\pi^E}\left[Q_{\mathcal{M} \cup c}^{c, \pi^E}(s, a^E)\right]. \tag{17}$$

where coefficients $\alpha \geq 0$, $\beta > 0$. $\beta$ can not equal to 0, since state $s$ has to be visited with at least some probability. Otherwise, we do not need to explain $\pi^E(s)$. Note that if $Q_{\mathcal{M} \cup c}^{c, \pi^E}(s, a') \leq \mathbb{E}_{\pi^E}\left[Q_{\mathcal{M} \cup c}^{c, \pi^E}(s, a^E)\right]$, $\pi'$ is a strictly better policy than the expert policy for any $\theta \in (0, 1]$ (larger rewards with equal or less costs). This clearly makes a contradiction. Hence,

we focus on $Q_{\mathcal{M}\cup c}^{c,\pi^E}(s,a') > \mathbb{E}_{\pi^E}\left[Q_{\mathcal{M}\cup c}^{c,\pi^E}(s,a^E)\right]$. In this case, we can always obtain a $\theta > 0$, by letting

$$\mathbb{E}_{\mu_0}\left[V_{\mathcal{M}\cup c}^{c,\pi^E}(s_0)\right] + \tau' = \alpha(P_{\mathcal{T}},\pi^E,\gamma,c) + \beta(P_{\mathcal{T}},\pi^E,\gamma,c)\cdot\left[(1-\theta)\mathbb{E}_{\pi^E}\left[Q_{\mathcal{M}\cup c}^{c,\pi^E}(s,a^E)\right] + \theta Q_{\mathcal{M}\cup c}^{c,\pi^E}(s,a')\right], \quad (18)$$

where $\tau' \in [0,\tau)$ denotes the leftover budget after applying $\pi'$. By subtracting Eq. (17) from Eq. (18), we have $\forall\, Q_{\mathcal{M}\cup c}^{c,\pi^E}(s,a') > \mathbb{E}_{\pi^E}\left[Q_{\mathcal{M}\cup c}^{c,\pi^E}(s,a^E)\right]$,

$$\theta = \frac{\tau'}{\beta(P_{\mathcal{T}},\pi^E,\gamma,c)\left[Q_{\mathcal{M}\cup c}^{c,\pi^E}(s,a') - \mathbb{E}_{\pi^E}\left[Q_{\mathcal{M}\cup c}^{c,\pi^E}(s',a^E)\right]\right]}. \quad (19)$$

With this analysis, if $A_{\mathcal{M}}^{r,\pi^E}(s,a') > 0$, which indicates 2) of Lemma 4.1 is not satisfied so 1) must be satisfied, $Q_{\mathcal{M}\cup c}^{c,\pi^E}(s,a') > \mathbb{E}_{\pi^E}\left[Q_{\mathcal{M}\cup c}^{c,\pi^E}(s,a^E)\right] = V_{\mathcal{M}\cup c}^{c,\pi^E}(s)$ suffices to let $\mathbb{E}_{\mu_0,\pi'',P_{\mathcal{T}}}\left[\sum_{t=0}^{\infty}\gamma^t c(s_t,a_t)\right] > \epsilon$ with $\pi''$ only differs from $\pi^E$ at state $s$ where $\pi''(s) = a'$, which is a constraint-violating condition. $\qquad\square$

## C.2. Proof of Lemma 4.4

*Proof.* In this proof, we distinguish two cases according to Assumption 4.3.
In the first case, the constraint (2) is hard, i.e., $\epsilon = 0$.

(i) By definition of expert policy $\pi^E$, we have $V_{\mathcal{M}\cup c}^{c,\pi^E}(s) = 0$. On one hand, if $c$ is feasible, $V_{\mathcal{M}\cup c}^{c,\pi^E} = \mathbb{E}_{\pi^E}[Q_{\mathcal{M}\cup c}^{c,\pi^E}] = 0$. Also, since $c \in [0,C_{\max}]^{\mathcal{S}\times\mathcal{A}}$, $Q_{\mathcal{M}\cup c}^{c,\pi^E} \geq 0$. As a result, $Q_{\mathcal{M}\cup c}^{c,\pi^E} = 0 = V_{\mathcal{M}\cup c}^{c,\pi^E}$. On the other hand, any $c \in [0,C_{\max}]^{\mathcal{S}\times\mathcal{A}}$ that satisfies $Q_{\mathcal{M}\cup c}^{c,\pi^E} = V_{\mathcal{M}\cup c}^{c,\pi^E} = 0$ makes $\pi^E$ an optimal policy under this condition.

(ii) By definition of expert policy $\pi^E$, we have $V_{\mathcal{M}\cup c}^{c,\pi^E}(s) = 0$. On one hand, since $A_{\mathcal{M}}^{r,\pi^E}(s,a) > 0$, if $c$ is feasible, $Q_{\mathcal{M}\cup c}^{c,\pi^E}(s,a) > 0$, otherwise $\pi^E$ is not optimal. On the other hand, any cost function $c \in [0,C_{\max}]^{\mathcal{S}\times\mathcal{A}}$ that satisfies $Q_{\mathcal{M}\cup c}^{c,\pi^E}(s,a) > 0 = V_{\mathcal{M}\cup c}^{c,\pi^E}(s)$ ensures action $a$ violates the constraint, and makes $\pi^E$ an optimal policy under this condition.

(iii) By definition of expert policy $\pi^E$, we have $V_{\mathcal{M}\cup c}^{c,\pi^E}(s) = 0$. On one hand, since $A_{\mathcal{M}}^{r,\pi^E}(s,a) \leq 0$, any $c \in [0,C_{\max}]^{\mathcal{S}\times\mathcal{A}}$ ensures the expert's optimality. However, in terms of the minimal set $\mathcal{C}_{\mathfrak{P}}$ in Definition 4.2, $c(s,a) = 0$ and $Q_{\mathcal{M}\cup c}^{c,\pi^E}(s,a) = 0 = V_{\mathcal{M}\cup c}^{c,\pi^E}(s)$. On the other hand, any $c(s,a) \in [0,C_{\max}]^{\mathcal{S}\times\mathcal{A}}$ that satisfies $Q_{\mathcal{M}\cup c}^{c,\pi^E}(s,a) = 0 = V_{\mathcal{M}\cup c}^{c,\pi^E}(s)$ ensures $\pi^E$ an optimal policy under this condition.

In the second case, the constraint in (2) is soft, i.e., $\epsilon > 0$, and the expert policy is deterministic.

(i) Since the expert policy $\pi^E$ is deterministic, we have $Q_{\mathcal{M}\cup c}^{c,\pi^E}(s,a) = V_{\mathcal{M}\cup c}^{c,\pi^E}(s)$. On one hand, if $c$ is feasible, $Q_{\mathcal{M}\cup c}^{c,\pi^E}(s,a) = V_{\mathcal{M}\cup c}^{c,\pi^E}(s)$. On the other hand, any $c \in [0,C_{\max}]^{\mathcal{S}\times\mathcal{A}}$ that satisfies $Q_{\mathcal{M}\cup c}^{c,\pi^E}(s,a) = V_{\mathcal{M}\cup c}^{c,\pi^E}(s)$ makes $\pi^E$ an optimal policy under this condition.

(ii) In this case, since $A_{\mathcal{M}}^{r,\pi^E}(s,a) > 0$, situation 2) of Lemma 4.1 is not satisfied. As a result, 1) of Lemma 4.1 must be satisfied. On one hand, if $c$ is feasible, $Q_{\mathcal{M}\cup c}^{c,\pi^E}(s,a) > Q_{\mathcal{M}\cup c}^{c,\pi^E}(s,a^E) = V_{\mathcal{M}\cup c}^{c,\pi^E}(s)$ suffices to let $\mathbb{E}_{\mu_0,\pi^E,P_{\mathcal{T}}}\left[\sum_{t=0}^{\infty}\gamma^t c(s_t,a_t)\right] > \epsilon$. On the other hand, any cost function $c \in [0,C_{\max}]^{\mathcal{S}\times\mathcal{A}}$ that satisfies $Q_{\mathcal{M}\cup c}^{c,\pi^E}(s,a) > V_{\mathcal{M}\cup c}^{c,\pi^E}(s)$ ensures action $a$ violates the constraint, and makes $\pi^E$ an optimal policy under this condition.

(iii) On one hand, since $A_{\mathcal{M}}^{r,\pi^E}(s,a) \leq 0$, any relationship between $Q_{\mathcal{M}\cup c}^{c,\pi^E}(s,a)$ and $V_{\mathcal{M}\cup c}^{c,\pi^E}(s)$ ensures the expert's optimality. However, in terms of the minimal set $\mathcal{C}_{\mathfrak{P}}$ in Definition 4.2, $Q_{\mathcal{M}\cup c}^{c,\pi^E}(s,a) \leq V_{\mathcal{M}\cup c}^{c,\pi^E}(s)$. On the other hand, any $c \in [0,C_{\max}]^{\mathcal{S}\times\mathcal{A}}$ that satisfies $Q_{\mathcal{M}\cup c}^{c,\pi^E}(s,a) \leq V_{\mathcal{M}\cup c}^{c,\pi^E}(s)$ ensures $\pi^E$ an optimal policy under this condition.

$\qquad\square$

## C.3. Proof of Lemma 4.5

**Lemma C.1.** *Let $\mathfrak{P} = (\mathcal{M}, \pi^E)$ be an ICRL problem. A Q-function satisfies the condition of Lemma 4.4 if and only if there exist $\zeta \in \mathbb{R}_{>0}^{\mathcal{S} \times \mathcal{A}}$ and $V^c \in \mathbb{R}_{\geq 0}^{\mathcal{S}}$ such that:*

$$Q_{\mathcal{M} \cup c}^c = A_{\mathcal{M}}^{r,\pi^E} \zeta + EV^c, \tag{20}$$

*where the expansion operator $E$ satisfies $(Ef)(s,a) = f(s)$.*

Here, the term $\zeta$ ensures 1) (when $A_{\mathcal{M}}^{r,\pi^E} > 0$) the constraint condition in (2) is violated at $(s,a)$ pairs that achieve larger rewards than the expert policy, and 2) (when $A_{\mathcal{M}}^{r,\pi^E} \leq 0$) only necessary cost functions are captured by feasible cost set $\mathcal{C}_{\mathfrak{P}}$.

*Proof.* We prove both the 'if' and 'only if' sides.

To demonstrate the "if" side, we can easily see that all the Q-functions of the form $Q_{\mathcal{M} \cup c}^c(s,a) = A_{\mathcal{M}}^{r,\pi^E}(s,a)\zeta(s,a) + EV^c(s)$ satisfies the conditions of Lemma 4.4 in the following:

1) Let $s \in \mathcal{S}$ and $a \in \mathcal{A}$ such that $\pi^E(a|s) > 0$, then we have $Q_{\mathcal{M} \cup c}^c(s,a) = V^c(s) = V_{\mathcal{M} \cup c}^c(s)$. This is the condition (i) in Lemma 4.4. Note that $V^c(s) = V_{\mathcal{M} \cup c}^c(s)$ holds true for the following two cases since each state $s \in \mathcal{S}$ has an expert policy such that $\pi^E(a|s) > 0$.

2) Let $s \in \mathcal{S}$ and $a \in \mathcal{A}$ such that $\pi^E(a|s) = 0$ and $Q_{\mathcal{M}}^{r,\pi^E}(s,a) > V_{\mathcal{M}}^{r,\pi^E}(s)$, then we have $Q_{\mathcal{M} \cup c}^c(s,a) = A_{\mathcal{M}}^{r,\pi^E}(s,a)\zeta(s,a) + V^c(s) = A_{\mathcal{M}}^{r,\pi^E}(s,a)\zeta(s,a) + V_{\mathcal{M} \cup c}^c(s) > V_{\mathcal{M} \cup c}^c(s)$. This is the case (ii) in Lemma 4.4.

3) Let $s \in \mathcal{S}$ and $a \in \mathcal{A}$ such that $\pi^E(a|s) = 0$ and $Q_{\mathcal{M}}^{r,\pi^E}(s,a) \leq V_{\mathcal{M}}^{r,\pi^E}(s)$, then we have $Q_{\mathcal{M} \cup c}^c(s,a) = A_{\mathcal{M}}^{r,\pi^E}(s,a)\zeta + V^c(s) = A_{\mathcal{M}}^{r,\pi^E}(s,a)\zeta(s,a) + V_{\mathcal{M} \cup c}^c(s) \leq V_{\mathcal{M} \cup c}^c(s)$. This is the case (iii) in Lemma 4.4.

To demonstrate the "only if" side, suppose that $Q_{\mathcal{M} \cup c}^c$ satisfies conditions of Lemma 4.4, we take $V^c(s) = V_{\mathcal{M} \cup c}^c(s)$ since we are proving the existence of $V^c \in \mathbb{R}_{\geq 0}^{\mathcal{S}}$.

1) In the critical region and follows the expert policy, where $Q_{\mathcal{M}}^{r,\pi^E}(s,a) = V_{\mathcal{M}}^{r,\pi^E}(s)$, $0\zeta(s,a) = Q_{\mathcal{M} \cup c}^c - EV_{\mathcal{M} \cup c}^c = 0$. Hence, there definitely exists $\zeta(s,a) > 0$.

2) In the constraint-violating region with more rewards, where $Q_{\mathcal{M}}^{r,\pi^E}(s,a) > V_{\mathcal{M}}^{r,\pi^E}(s)$, $A_{\mathcal{M}}^{r,\pi^E}(s,a)\zeta(s,a) = Q_{\mathcal{M} \cup c}^c - EV_{\mathcal{M} \cup c}^c > 0$. Hence, there definitely exists $\zeta(s,a) > 0$.

3) In the non-critical region with fewer rewards, where $Q_{\mathcal{M}}^{r,\pi^E}(s,a) \leq V_{\mathcal{M}}^{r,\pi^E}(s)$, $A_{\mathcal{M}}^{r,\pi^E}(s,a)\zeta(s,a) = Q_{\mathcal{M} \cup c}^c - EV_{\mathcal{M} \cup c}^c \leq 0$. Hence, there definitely exists $\zeta(s,a) > 0$. $\qquad\square$

### Final proof of Lemma 4.5

*Proof.* Recall that $Q_{\mathcal{M} \cup c}^c = (I_{\mathcal{S} \times \mathcal{A}} - \gamma P_{\mathcal{T}} \pi^E)^{-1} c$ and based on Lemma C.1, we can show that:

$$\begin{aligned} c &= \left(I_{\mathcal{S} \times \mathcal{A}} - \gamma P_{\mathcal{T}} \pi^E\right)\left(A_{\mathcal{M}}^{r,\pi^E} \zeta + EV^c\right) \\ &= A_{\mathcal{M}}^{r,\pi^E} \zeta + EV^c - \gamma P_{\mathcal{T}} \pi^E A_{\mathcal{M}}^{r,\pi^E} \zeta - \gamma P_{\mathcal{T}} \pi^E EV^c \end{aligned}$$

Since $\pi^E A_{\mathcal{M}}^{r,\pi^E} = \mathbf{0}_{\mathcal{S}}$ and $\pi^E E = I_{\mathcal{S}}$,

$$c = A_{\mathcal{M}}^{r,\pi^E} \zeta + (E - \gamma P_{\mathcal{T}})V^c$$

We now bound the infinity norm of $\zeta$ and $V^c$. First, from Eq. (20), we know that $EV^c(s) = Q_{\mathcal{M} \cup c}^c(s, a^E)$. Hence, intuitively $\|V^c(s)\|_\infty \leq \frac{C_{\max}}{1-\gamma}$. Second, from Eq. (6), $c(s,a) = A_{\mathcal{M}}^{r,\pi^E}(s,a)\zeta(s,a) + (E - \gamma P_{\mathcal{T}})V^c(s)$. 1) When $A_{\mathcal{M}}^{r,\pi^E}(s,a) > 0$, since for every $A_{\mathcal{M}}^{r,\pi^E}(s,a) > 0$, $\zeta(s,a)$ should satisfy the existence of a cost function in $[0, C_{\max}]^{\mathcal{S} \times \mathcal{A}}$, $\zeta(s,a) = (c(s,a) - (E - \gamma P_{\mathcal{T}})V^c(s))/A_{\mathcal{M}}^{r,\pi^E}(s,a) \leq C_{\max}/\max_{(s,a)}^+ A_{\mathcal{M}}^{r,\pi^E}(s,a)$. 2) When $A_{\mathcal{M}}^{r,\pi^E}(s,a) < 0$, $\zeta(s,a) = (-c(s,a) + (E - \gamma P_{\mathcal{T}})V^c(s))/(-A_{\mathcal{M}}^{r,\pi^E}(s,a))$. Since $(E - \gamma P_{\mathcal{T}})V^c(s) = c(s, a^E) \leq C_{\max}$, $\zeta \leq C_{\max}/\left(-\max_{(s,a)}^+ A_{\mathcal{M}}^{r,\pi^E}(s,a)\right)$. 3) When $A_{\mathcal{M}}^{r,\pi^E} = 0$, we define $\zeta(s,a) = 0$. To combine all the three conditions, $\|\zeta\|_\infty = C_{\max}/\max_{(s,a)}^+ |A_{\mathcal{M}}^{r,\pi^E}|$. $\qquad\square$

## C.4. Proof of Lemma 4.6

*Proof.* From Lemma 4.5, $\forall (s,a) \in \mathcal{S} \times \mathcal{A}$, we can express the cost functions belonging to $\mathcal{C}_{\mathfrak{P}}$ as:

$$c(s,a) = A_{\mathcal{M}}^{r,\pi^E} \zeta(s,a) + (E - \gamma P_{\mathcal{T}}) V^c(s,a).$$

Regarding $\widehat{\pi}^E$ and $\widehat{P_{\mathcal{T}}}$, we can express the estimated cost function belonging to $\mathcal{C}_{\widehat{\mathfrak{P}}}$ as:

$$\widehat{c}(s,a) = A_{\widehat{\mathcal{M}}}^{r,\widehat{\pi}^E} \widehat{\zeta}(s,a) + (E - \gamma \widehat{P_{\mathcal{T}}}) \widehat{V}^c(s,a),$$

What we need to do first is to provide a specific choice of $\widehat{\zeta}$ and $\widehat{V}$ under which $\widehat{c} \in [0, C_{\max}]^{\mathcal{S} \times \mathcal{A}}$. We construct

$$\widetilde{c}(s,a) = A_{\widehat{\mathcal{M}}}^{r,\widehat{\pi}^E} \zeta(s,a) + (E - \gamma \widehat{P_{\mathcal{T}}}) V^c(s,a).$$

We now define the absolute difference between $\widetilde{c}(s,a)$ and $c(s,a)$ as

$$\chi(s,a) = |\widetilde{c}(s,a) - c(s,a)| = \gamma \left| (P_{\mathcal{T}} - \widehat{P_{\mathcal{T}}}) V^c \right| (s,a) + \left| A_{\mathcal{M}}^{r,\pi^E} - A_{\widehat{\mathcal{M}}}^{r,\widehat{\pi}^E} \right| \zeta(s,a),$$

$$\chi = \max_{(s,a) \in \mathcal{S} \times \mathcal{A}} \chi(s,a).$$

$\forall (s,a) \in \mathcal{S} \times \mathcal{A}$, since $c(s,a) \in [0, C_{\max}]$ and $\widetilde{c}(s,a) - c(s,a) \in [-\chi, \chi]$, we have:

$$\widetilde{c}(s,a) = c(s,a) + (\widetilde{c}(s,a) - c(s,a)) \in [-\chi, C_{\max} + \chi] \tag{21}$$

Therefore, there is always a state-action pair $(s', a')$ such that

$$\min_{(s,a) \in \mathcal{S} \times \mathcal{A}} \widetilde{c}(s,a) = \widetilde{c}(s',a') = A_{\widehat{\mathcal{M}}}^{r,\widehat{\pi}^E} \zeta(s',a') + (E - \gamma \widehat{P_{\mathcal{T}}}) V^c(s',a') \geq -\chi.$$

To obtain $\widehat{c}(s,a) \in [0, C_{\max}]$, we distinguish two cases: 1) $\widetilde{c}(s',a') < 0$ and 2) $\widetilde{c}(s',a') \geq 0$.

**Case one:** $\widetilde{c}(s',a') < 0$:

By subtracting $\widetilde{c}(s',a')$ from all $\widetilde{c}(s,a)$, we have

$$
\begin{aligned}
\bar{c}(s,a) &= \widetilde{c}(s,a) - \widetilde{c}(s',a') \\
&= A_{\widehat{\mathcal{M}}}^{r,\widehat{\pi}^E} \zeta(s,a) + (E - \gamma \widehat{P_{\mathcal{T}}}) V^c(s,a) - A_{\widehat{\mathcal{M}}}^{r,\widehat{\pi}^E} \zeta(s',a') - (E - \gamma \widehat{P_{\mathcal{T}}}) V^c(s',a') \\
&= A_{\widehat{\mathcal{M}}}^{r,\widehat{\pi}^E} \left[ \zeta(s,a) - \frac{A_{\widehat{\mathcal{M}}}^{r,\widehat{\pi}^E}(s',a')}{A_{\widehat{\mathcal{M}}}^{r,\widehat{\pi}^E}(s,a)} \zeta(s'.a') \right] + (E - \gamma \widehat{P_{\mathcal{T}}}) \left[ V^c(s,a) - V^c(s',a') \right] \\
&\geq 0
\end{aligned}
$$

Also, note that $\forall (s,a) \in \mathcal{S} \times \mathcal{A}$, we have:

$$\widetilde{c}(s',a') < 0, \bar{c}(s,a) = \widetilde{c}(s,a) - \widetilde{c}(s',a') \leq |\widetilde{c}(s,a)| + |\widetilde{c}(s',a')| \leq C_{\max} + \chi(s,a) + \chi$$

Hence, $\forall (s,a) \in \mathcal{S} \times \mathcal{A}, \bar{c}(s,a) \in [0, C_{\max} + \chi(s,a) + \chi]$. Because we are looking for the existence of $\widehat{c} \in \mathcal{C}_{\widehat{\mathfrak{P}}}$ satisfying $\widehat{c} \in [0, C_{\max}]^{\mathcal{S} \times \mathcal{A}}$, we can now provide a specific choice of $\widehat{\zeta}$ and $\widehat{V}$ under which $\widehat{c} \in [0, C_{\max}]^{\mathcal{S} \times \mathcal{A}}$:

$$\widehat{\zeta}(s,a) = \frac{\zeta(s,a) - \frac{A_{\widehat{\mathcal{M}}}^{r,\widehat{\pi}^E}(s',a')}{A_{\widehat{\mathcal{M}}}^{r,\widehat{\pi}^E}(s,a)} \zeta(s',a')}{1 + (\chi(s,a) + \chi)/C_{\max}}, \widehat{V}^c(s,a) = \frac{V^c(s,a) - V^c(s',a')}{1 + (\chi(s,a) + \chi)/C_{\max}}, \widehat{c}(s,a) = \frac{\bar{c}(s,a)}{1 + (\chi(s,a) + \chi)/C_{\max}} \tag{22}$$

We then quantify the estimation error between $\widehat{c}(s,a)$ and $c(s,a)$.

$$|c(s,a) - \widehat{c}(s,a)| = \left| c(s,a) - \frac{\bar{c}(s,a)}{1 + (\chi(s,a) + \chi)/C_{\max}} \right|$$

$$= \frac{1}{1 + (\chi(s,a) + \chi)/C_{\max}} \Big[ |c(s,a) - \bar{c}(s,a)| + ((\chi(s,a) + \chi)/C_{\max})|c(s,a)| \Big]$$

$$= \frac{1}{1 + (\chi(s,a) + \chi)/C_{\max}} \Big[ |c(s,a) - \bar{c}(s,a)| + ((\chi(s,a) + \chi)/C_{\max})|c(s,a)| \Big]$$

$$\leq \frac{1}{1 + (\chi(s,a) + \chi)/C_{\max}} \Big[ |c(s,a) - \widetilde{c}(s,a)| + |\widetilde{c}(s,a) - \bar{c}(s,a)| + ((\chi(s,a) + \chi)/C_{\max})|c(s,a)| \Big]$$

$$\leq \frac{\chi(s,a) + \chi + ((\chi(s,a) + \chi)/C_{\max})C_{\max}}{1 + (\chi(s,a) + \chi)/C_{\max}}$$

$$\leq \frac{2\chi(s,a) + 2\chi}{1 + (\chi(s,a) + \chi)/C_{\max}}$$

**Case Two:** $\widetilde{c}(s',a') \geq 0$:

Note that $\forall (s,a) \in \mathcal{S} \times \mathcal{A}$, we have:

$$\widetilde{c}(s',a') \geq 0, \bar{c}(s,a) = \widetilde{c}(s,a) - \widetilde{c}(s',a') \leq \widetilde{c}(s,a) \leq C_{\max} + \chi(s,a) \tag{23}$$

Hence, $\forall (s,a) \in \mathcal{S} \times \mathcal{A}, \bar{c}(s,a) \in [0, C_{\max} + \chi(s,a)]$. Because we are looking for the existence of $\widehat{c} \in \mathcal{C}_{\widehat{\mathfrak{P}}}$ satisfying $\widehat{c} \in [0, C_{\max}]^{\mathcal{S} \times \mathcal{A}}$, we can now provide a specific choice of $\widehat{\zeta}$ and $\widehat{V}$ under which $\widehat{c} \in [0, C_{\max}]^{\mathcal{S} \times \mathcal{A}}$:

$$\widehat{\zeta}(s,a) = \frac{\zeta(s,a)}{1 + \chi(s,a)/C_{\max}}, \widehat{V}^c(s,a) = \frac{V^c(s,a)}{1 + \chi(s,a)/C_{\max}}, \widehat{c}(s,a) = \frac{\widetilde{c}(s,a)}{1 + \chi(s,a)/C_{\max}}. \tag{24}$$

We then quantify the estimation error between $\widehat{c}(s,a)$ and $c(s,a)$.

$$|c(s,a) - \widehat{c}(s,a)| = \left| c(s,a) - \frac{\widetilde{c}(s,a)}{1 + \chi(s,a)/C_{\max}} \right|$$

$$= \frac{1}{1 + \chi(s,a)/C_{\max}} \Big[ |c(s,a) - \widetilde{c}(s,a)| + (\chi(s,a)/C_{\max})|c(s,a)| \Big]$$

$$\leq \frac{\chi(s,a) + (\chi(s,a)/C_{\max})C_{\max}}{1 + \chi(s,a)/C_{\max}}$$

$$\leq \frac{2\chi(s,a)}{1 + \chi(s,a)/C_{\max}}$$

$$\leq \frac{2\chi(s,a) + 2\chi}{1 + (\chi(s,a) + \chi)/C_{\max}} \tag{25}$$

Combine the upper bound of the estimation error for cost functions in both cases, we finally derive:

$$|c(s,a) - \widehat{c}(s,a)| \leq \frac{2\chi(s,a) + 2\chi}{1 + (\chi(s,a) + \chi)/C_{\max}} = \frac{2(\chi(s,a) + \chi)}{1 + (\chi(s,a) + \chi)/C_{\max}}$$

$\square$

## C.5. Proof of Lemma 4.7

**Lemma C.2.** *(Simulation Lemma for action-value function.) Let* $\mathcal{M} = (\mathcal{S}, \mathcal{A}, P_{\mathcal{T}}, r, \mu_0, \gamma)$ *and* $\widehat{\mathcal{M}} = (\mathcal{S}, \mathcal{A}, \widehat{P_{\mathcal{T}}}, r, \mu_0, \gamma)$ *be two MDPs. Let* $\widehat{\pi} \in \Delta_{\mathcal{S}}^{\mathcal{A}}$ *be a policy. The following equality holds element-wise:*

$$Q_{\mathcal{M}}^{r,\widehat{\pi}} - Q_{\widehat{\mathcal{M}}}^{r,\widehat{\pi}} = \gamma(I_{\mathcal{S} \times \mathcal{A}} - \gamma P_{\mathcal{T}}\widehat{\pi})^{-1}(P_{\mathcal{T}} - \widehat{P_{\mathcal{T}}})V_{\widehat{\mathcal{M}}}^{r,\widehat{\pi}} \tag{26}$$

*Proof.* The proof can be shown as follows:

$$Q_{\mathcal{M}}^{r,\widehat{\pi}} - Q_{\widehat{\mathcal{M}}}^{r,\widehat{\pi}} = (I_{\mathcal{S} \times \mathcal{A}} - \gamma P_{\mathcal{T}}\widehat{\pi})^{-1}r - (I_{\mathcal{S} \times \mathcal{A}} - \gamma P_{\mathcal{T}}\widehat{\pi})^{-1}(I_{\mathcal{S} \times \mathcal{A}} - \gamma P_{\mathcal{T}}\widehat{\pi})Q_{\widehat{\mathcal{M}}}^{r,\widehat{\pi}}$$

$$= (I_{\mathcal{S} \times \mathcal{A}} - \gamma P_{\mathcal{T}}\widehat{\pi})^{-1}(I_{\mathcal{S} \times \mathcal{A}} - \gamma \widehat{P_{\mathcal{T}}}\widehat{\pi})Q_{\widehat{\mathcal{M}}}^{r,\widehat{\pi}} - (I_{\mathcal{S} \times \mathcal{A}} - \gamma P_{\mathcal{T}}\widehat{\pi})^{-1}(I_{\mathcal{S} \times \mathcal{A}} - \gamma P_{\mathcal{T}}\widehat{\pi})Q_{\widehat{\mathcal{M}}}^{r,\widehat{\pi}}$$

$$= \gamma(I_{\mathcal{S}\times\mathcal{A}} - \gamma P_{\mathcal{T}}\widehat{\pi})^{-1}(P_{\mathcal{T}} - \widehat{P_{\mathcal{T}}})\widehat{\pi}Q_{\widehat{\mathcal{M}}}^{r,\widehat{\pi}}$$

$$= \gamma(I_{\mathcal{S}\times\mathcal{A}} - \gamma P_{\mathcal{T}}\widehat{\pi})^{-1}(P_{\mathcal{T}} - \widehat{P_{\mathcal{T}}})V_{\widehat{\mathcal{M}}}^{r,\widehat{\pi}}$$

$\square$

**Lemma C.3.** *(Simulation Lemma for state-value function.) Let $\mathcal{M} = (\mathcal{S}, \mathcal{A}, P_{\mathcal{T}}, r, \mu_0, \gamma)$ and $\widehat{\mathcal{M}} = (\mathcal{S}, \mathcal{A}, \widehat{P_{\mathcal{T}}}, r, \mu_0, \gamma)$ be two MDPs. Let $\widehat{\pi} \in \Delta_{\mathcal{S}}^{\mathcal{A}}$ be a policy. The following equality holds element-wise:*

$$V_{\mathcal{M}}^{r,\widehat{\pi}} - V_{\widehat{\mathcal{M}}}^{r,\widehat{\pi}} = \gamma(I_{\mathcal{S}} - \gamma\widehat{\pi}P_{\mathcal{T}})^{-1}\widehat{\pi}(\widehat{P_{\mathcal{T}}} - P_{\mathcal{T}})V_{\widehat{\mathcal{M}}}^{r,\widehat{\pi}} \tag{27}$$

*Proof.* The proof can be shown as follows:

$$V_{\mathcal{M}}^{r,\widehat{\pi}} - V_{\widehat{\mathcal{M}}}^{r,\widehat{\pi}} = (I_{\mathcal{S}} - \gamma\widehat{\pi}P_{\mathcal{T}})^{-1}r - (I_{\mathcal{S}} - \gamma\widehat{\pi}P_{\mathcal{T}})^{-1}(I_{\mathcal{S}} - \gamma\widehat{\pi}P_{\mathcal{T}})V_{\widehat{\mathcal{M}}}^{r,\widehat{\pi}}$$

$$= (I_{\mathcal{S}} - \gamma\widehat{\pi}P_{\mathcal{T}})^{-1}(I_{\mathcal{S}} - \gamma\widehat{\pi}\widehat{P_{\mathcal{T}}})V_{\widehat{\mathcal{M}}}^{r,\widehat{\pi}} - (I_{\mathcal{S}} - \gamma\widehat{\pi}P_{\mathcal{T}})^{-1}(I_{\mathcal{S}} - \gamma\widehat{\pi}P_{\mathcal{T}})V_{\widehat{\mathcal{M}}}^{r,\widehat{\pi}}$$

$$= \gamma(I_{\mathcal{S}} - \gamma\widehat{\pi}P_{\mathcal{T}})^{-1}\widehat{\pi}(P_{\mathcal{T}} - \widehat{P_{\mathcal{T}}})V_{\widehat{\mathcal{M}}}^{r,\widehat{\pi}}$$

$$= \gamma(I_{\mathcal{S}} - \gamma\widehat{\pi}P_{\mathcal{T}})^{-1}\widehat{\pi}(P_{\mathcal{T}} - \widehat{P_{\mathcal{T}}})V_{\widehat{\mathcal{M}}}^{r,\widehat{\pi}}$$

$\square$

**Lemma C.4.** *(Policy Mismatch Lemma.) Let $\mathcal{M} = (\mathcal{S}, \mathcal{A}, P_{\mathcal{T}}, r, \mu_0, \gamma)$ be an MDP. Let $\pi, \widehat{\pi} \in \Delta_{\mathcal{S}}^{\mathcal{A}}$ be two policies. The following equality holds element-wise:*

$$V_{\mathcal{M}}^{r,\pi} - V_{\mathcal{M}}^{r,\widehat{\pi}} = \gamma(I_{\mathcal{S}} - \gamma\widehat{\pi}P_{\mathcal{T}})^{-1}(\pi - \widehat{\pi})P_{\mathcal{T}}V_{\mathcal{M}}^{r,\pi}$$

*Proof.* The proof can be shown as follows:

$$V_{\mathcal{M}}^{r,\pi} - V_{\mathcal{M}}^{r,\widehat{\pi}} = (I_{\mathcal{S}} - \gamma\widehat{\pi}P_{\mathcal{T}})^{-1}(I_{\mathcal{S}} - \gamma\widehat{\pi}P_{\mathcal{T}})V_{\mathcal{M}}^{r,\pi} - (I_{\mathcal{S}} - \gamma\widehat{\pi}P_{\mathcal{T}})^{-1}r$$

$$= (I_{\mathcal{S}} - \gamma\widehat{\pi}P_{\mathcal{T}})^{-1}(I_{\mathcal{S}} - \gamma\widehat{\pi}P_{\mathcal{T}})V_{\mathcal{M}}^{r,\pi} - (I_{\mathcal{S}} - \gamma\widehat{\pi}P_{\mathcal{T}})^{-1}(I_{\mathcal{S}} - \gamma\pi P_{\mathcal{T}})V_{\mathcal{M}}^{r,\pi}$$

$$= \gamma(I_{\mathcal{S}} - \gamma\widehat{\pi}P_{\mathcal{T}})^{-1}(\pi - \widehat{\pi})P_{\mathcal{T}}V_{\mathcal{M}}^{r,\pi}$$

$\square$

**Final proof of Lemma 4.7**

*Proof.* By utilizing the triangular inequality of norms, we can obtain:

$$\left|A_{\mathcal{M}}^{r,\pi^E} - A_{\widehat{\mathcal{M}}}^{r,\widehat{\pi}^E}\right| \leq \left|A_{\mathcal{M}}^{r,\widehat{\pi}^E} - A_{\widehat{\mathcal{M}}}^{r,\widehat{\pi}^E}\right| + \left|A_{\mathcal{M}}^{r,\pi^E} - A_{\mathcal{M}}^{r,\widehat{\pi}^E}\right|$$

$$\overset{I,II}{\leq} \frac{2\gamma}{1-\gamma}\left|(\widehat{P_{\mathcal{T}}} - P_{\mathcal{T}})V_{\widehat{\mathcal{M}}}^{r,\widehat{\pi}^E}\right| + \frac{\gamma(1+\gamma)}{1-\gamma}\left|(\pi^E - \widehat{\pi}^E)P_{\mathcal{T}}V_{\mathcal{M}}^{r,\pi^E}\right|, \tag{28}$$

where the second inequality is derived by the following two parts.

**Part I.** Let's consider the first part.

$$\left|A_{\mathcal{M}}^{r,\widehat{\pi}^E} - A_{\widehat{\mathcal{M}}}^{r,\widehat{\pi}^E}\right| \overset{(i)}{=} \left|\left(Q_{\mathcal{M}}^{r,\widehat{\pi}^E} - Q_{\widehat{\mathcal{M}}}^{r,\widehat{\pi}^E}\right) - E\left(V_{\mathcal{M}}^{r,\widehat{\pi}^E} - V_{\widehat{\mathcal{M}}}^{r,\widehat{\pi}^E}\right)\right|$$

$$\overset{(ii)}{\leq} \left|\left(Q_{\mathcal{M}}^{r,\widehat{\pi}^E} - Q_{\widehat{\mathcal{M}}}^{r,\widehat{\pi}^E}\right)\right| + \left|E\left(V_{\mathcal{M}}^{r,\widehat{\pi}^E} - V_{\widehat{\mathcal{M}}}^{r,\widehat{\pi}^E}\right)\right|$$

$$\overset{(iii)}{=} \gamma\left|(I_{\mathcal{S}\times\mathcal{A}} - \gamma P_{\mathcal{T}}\widehat{\pi}^E)^{-1}(\widehat{P_{\mathcal{T}}} - P_{\mathcal{T}})V_{\widehat{\mathcal{M}}}^{r,\widehat{\pi}^E}\right| + \gamma\left|(I_{\mathcal{S}} - \gamma\widehat{\pi}^E P_{\mathcal{T}})^{-1}\widehat{\pi}^E(\widehat{P_{\mathcal{T}}} - P_{\mathcal{T}})V_{\widehat{\mathcal{M}}}^{r,\widehat{\pi}^E}\right|$$

$$\overset{(iv)}{=} \gamma\left\|(I_{\mathcal{S}\times\mathcal{A}} - \gamma P_{\mathcal{T}}\widehat{\pi}^E)^{-1}\right\|_{\infty}\left|(\widehat{P_{\mathcal{T}}} - P_{\mathcal{T}})V_{\widehat{\mathcal{M}}}^{r,\widehat{\pi}^E}\right| + \gamma\left\|(I_{\mathcal{S}} - \gamma\widehat{\pi}^E P_{\mathcal{T}})^{-1}\right\|_{\infty}\|\widehat{\pi}^E\|_{\infty}\left|(\widehat{P_{\mathcal{T}}} - P_{\mathcal{T}})V_{\widehat{\mathcal{M}}}^{r,\widehat{\pi}^E}\right|$$

$$\overset{(v)}{\leq} \frac{2\gamma}{1-\gamma}\left|(\widehat{P_{\mathcal{T}}} - P_{\mathcal{T}})V_{\widehat{\mathcal{M}}}^{r,\widehat{\pi}^E}\right|$$

- (i) exploits the definition of the advantage function.

- (ii) applies the triangular inequality.

- (iii) applies the simulation Lemma for action-value function in Lemma C.2 (a variant of (Agarwal et al., 2019, Lemma 2.2)) and the simulation Lemma for state-value function in Lemma C.3.

- (iv) exploits Holder's inequality and the theorem of matrix infinity norm inequalities that $\|AB\|_\infty \leq \|A\|_\infty\|B\|_\infty$.

- (v) exploits the fact that $\|(I_{\mathcal{S}\times\mathcal{A}} - \gamma P_\mathcal{T}\widehat{\pi}^E)^{-1}\|_\infty \leq \frac{1}{1-\gamma}$, $\|(I_\mathcal{S} - \gamma\widehat{\pi}^E P_\mathcal{T})^{-1}\|_\infty \leq \frac{1}{1-\gamma}$, and $\|\pi^E\|_\infty \leq 1$.

**Part II.** Let's consider the second part:

$$
\begin{aligned}
\left|A_\mathcal{M}^{r,\pi^E} - A_\mathcal{M}^{r,\widehat{\pi}^E}\right| &= \left|\left(Q_\mathcal{M}^{r,\pi^E} - Q_\mathcal{M}^{r,\widehat{\pi}^E}\right) - E\left(V_\mathcal{M}^{r,\pi^E} - V_\mathcal{M}^{r,\widehat{\pi}^E}\right)\right| \\
&\overset{(i)}{=} \left|\gamma\left(P_\mathcal{T}V_\mathcal{M}^{r,\pi^E} - P_\mathcal{T}V_\mathcal{M}^{r,\widehat{\pi}^E}\right) - E\left(V_\mathcal{M}^{r,\pi^E} - V_\mathcal{M}^{r,\widehat{\pi}^E}\right)\right| \\
&\overset{(ii)}{=} \gamma\left|P_\mathcal{T}\left(V_\mathcal{M}^{r,\pi^E} - V_\mathcal{M}^{r,\widehat{\pi}^E}\right)\right| + \left|E\left(V_\mathcal{M}^{r,\pi^E} - V_\mathcal{M}^{r,\widehat{\pi}^E}\right)\right| \\
&\overset{(iii)}{\leq} (1+\gamma)\left|E\left(V_\mathcal{M}^{r,\pi^E} - V_\mathcal{M}^{r,\widehat{\pi}^E}\right)\right| \\
&\overset{(iv)}{\leq} \gamma(1+\gamma)\left|(I_\mathcal{S} - \gamma\widehat{\pi}^E P_\mathcal{T})^{-1}(\pi^E - \widehat{\pi}^E)P_\mathcal{T}V_\mathcal{M}^{r,\pi^E}\right| \\
&\leq \gamma(1+\gamma)\left\|(I_\mathcal{S} - \gamma\widehat{\pi}^E P_\mathcal{T})^{-1}\right\|_\infty\left|(\pi^E - \widehat{\pi}^E)P_\mathcal{T}V_\mathcal{M}^{r,\pi^E}\right| \\
&\overset{(v)}{\leq} \frac{\gamma(1+\gamma)}{1-\gamma}\left|(\pi^E - \widehat{\pi}^E)P_\mathcal{T}V_\mathcal{M}^{r,\pi^E}\right|
\end{aligned}
$$

- (i) applies the Bellman equation $Q = r + \gamma P_\mathcal{T}V$.

- (ii) applies the triangular inequality.

- (iii) holds since $\|P_\mathcal{T}\|_\infty \leq 1$.

- (iv) applies the policy mismatch Lemma for state-value function in Lemma C.4.

- (v) exploits the fact that $\|(I_\mathcal{S} - \gamma\widehat{\pi}^E P_\mathcal{T})^{-1}\|_\infty \leq \frac{1}{1-\gamma}$

$\square$

## C.6. Proof of Lemma 4.8

**Lemma C.5.** *(Good Event). Let $\delta \in (0,1)$, define the good event $\mathcal{E}_k$ as the event at iteration $k$ such that the following inequalities hold simultaneously for all $(s,a) \in \mathcal{S} \times \mathcal{A}$ and $k \geq 1$:*

$$
\left|(\widehat{P_\mathcal{T}}_k - P_\mathcal{T})V_{\widehat{\mathcal{M}}_k}^{r,\widehat{\pi}_k^E}\right|(s,a) \leq \frac{R_{\max}}{1-\gamma}\sqrt{\frac{\ell_k(s,a)}{2N_k^+(s,a)}},
$$

$$
\left|(P_\mathcal{T} - \widehat{P_\mathcal{T}}_k)V_\mathcal{M}^{r,\pi^E}\right|(s,a) \leq \frac{R_{\max}}{1-\gamma}\sqrt{\frac{\ell_k(s,a)}{2N_k^+(s,a)}},
$$

$$
\left|(\pi - \widehat{\pi}_k^E)P_\mathcal{T}V_\mathcal{M}^{r,\pi^E}\right|(s,a) \leq \frac{R_{\max}}{1-\gamma}\sqrt{\frac{\ell_k(s,a)}{2N_k^+(s,a)}},
$$

$$
\left|(\widehat{\pi}_k^E - \pi^E)\widehat{P_\mathcal{T}}_k V_{\widehat{\mathcal{M}}_k}^{r,\widehat{\pi}_k^E}\right|(s,a) \leq \frac{R_{\max}}{1-\gamma}\sqrt{\frac{\ell_k(s,a)}{2N_k^+(s,a)}},
$$

$$\left| (P_{\mathcal{T}} - \widehat{P_{\mathcal{T}}}_k)V^c \right| (s,a) \le \frac{C_{\max}}{1-\gamma} \sqrt{\frac{\ell_k(s,a)}{2N_k^+(s,a)}},$$

$$\left| (P_{\mathcal{T}} - \widehat{P_{\mathcal{T}}}_k)\widehat{V}_k^c \right| (s,a) \le \frac{C_{\max}}{1-\gamma} \sqrt{\frac{\ell_k(s,a)}{2N_k^+(s,a)}},$$

*where* $V_{\widehat{\mathcal{M}}_k}^{r,\widehat{\pi}^E}$, $V_{\mathcal{M}}^{r,\pi^E}$, $V^c$ *and* $\widehat{V}_k^c$ *are defined in Lemma 4.6 and Lemma 4.7.* $\ell_k(s,a) = \log(36SA(N_k^+(s,a))^2/\delta)$. *Then,* $\Pr(\mathcal{E}_k) \ge 1 - \delta$.

*Proof.* We show that each statement does not hold with probability less than $\delta/6$. Let us denote $\beta^3_{N_k^+(s,a)}(s,a) = \frac{C_{\max}}{1-\gamma}\sqrt{\frac{\ell_k(s,a)}{2N_k^+(s,a)}}$ and $\beta^3_m(s,a) = \frac{C_{\max}}{1-\gamma}\sqrt{\frac{\ell_k(s,a)}{2m}}$. Consider the second to last inequality. The probability that it does not hold is:

$$\Pr\left[ \exists k \ge 1, \exists(s,a) \in \mathcal{S} \times \mathcal{A} : \left| (P_{\mathcal{T}} - \widehat{P_{\mathcal{T}}}_k)V^c \right| (s,a) > \beta^3_{N_k^+(s,a)}(s,a) \right]$$

$$\overset{(a)}{\le} \sum_{(s,a)} \Pr\left[ \exists k \ge 1 : \left| (P_{\mathcal{T}} - \widehat{P_{\mathcal{T}}}_k)V^c \right| (s,a) > \beta^3_{N_k^+(s,a)}(s,a) \right]$$

$$\overset{(b)}{=} \sum_{(s,a)} \Pr\left[ \exists m \ge 0 : \left| (P_{\mathcal{T}} - \widehat{P_{\mathcal{T}}}_k)V^c \right| (s,a) > \beta^3_m(s,a) \right]$$

$$\overset{(c)}{\le} \sum_m \sum_{(s,a)} \Pr\left[ \left| (P_{\mathcal{T}} - \widehat{P_{\mathcal{T}}}_k)V^c \right| (s,a) > \beta^3_m(s,a) \right]$$

$$\overset{(d)}{\le} \sum_m \sum_{(s,a)} 2\exp\left( \frac{-2(\beta^3_m(s,a))^2 m^2}{m\left(\frac{C_{\max}}{1-\gamma}\right)^2} \right)$$

$$= \sum_m \sum_{(s,a)} 2\exp\left( -\ell_k(s,a) \right)$$

$$= \sum_m \sum_{(s,a)} \frac{2\delta}{36SA(m^+)^2}$$

$$= \frac{\delta}{18}\left(1 + \frac{\pi^2}{6}\right) \le \frac{\delta}{6} \tag{29}$$

- (a) and (c) use union-bound inequalities over $(s,a)$ and $m$.

- (b) assumes that we visit a state-action pair $(s,a)$ for $m$ times and only focus on these $m$ times that the transition model is updated.

- (d) applies the Hoeffding's inequality and $\|V^c\|_\infty \le C_{\max}/(1-\gamma)$ in Lemma 4.6. The factor $m^2$ in the numerator results from dividing by $1/m$ to average over samples, and the factor $m$ in the denominator results from the sum over $m$ in the denominator of Hoeffding's bound.

Similarly, we have $\beta^{1,2}_{N_k^+(s,a)}(s,a) = \frac{R_{\max}}{1-\gamma}\sqrt{\frac{\ell_k(s,a)}{2N_k^+(s,a)}}$ and $\beta^{1,2}_m(s,a) = \frac{R_{\max}}{1-\gamma}\sqrt{\frac{\ell_k(s,a)}{2m}}$ for Lemma's first and second, third and fourth inequalities, respectively. Lemma's last inequality employs $\beta^3_{N_k^+(s,a)}(s,a)$ and $\beta^3_m(s,a)$ again. A union bound over the six probabilities results in $\Pr(\bar{\mathcal{E}}_k) \le (\delta/6 + \delta/6 + \delta/6 + \delta/6 + \delta/6 + \delta/6) = \delta$. Thus, $\Pr(\mathcal{E}_k) = 1 - \Pr(\bar{\mathcal{E}}_k) \ge 1 - \delta$. $\square$

**Final proof of Lemma 4.8**

*Proof.*

$$\chi(s,a) \overset{(a)}{\leq} \gamma \left| (P_{\mathcal{T}} - \widehat{P_{\mathcal{T}}})V^c \right| + \left| A_{\mathcal{M}}^{r,\pi^E} - A_{\widehat{\mathcal{M}}}^{r,\widehat{\pi}^E} \right| \zeta$$

$$\overset{(b)}{\leq} \frac{\gamma \left( R_{\max}(3+\gamma)\zeta(s,a) + C_{\max}(1-\gamma) \right)}{(1-\gamma)^2} \sqrt{\frac{\ell_k(s,a)}{2N_k^+(s,a)}}$$

$$\leq \frac{\gamma \left( R_{\max}(3+\gamma)\|\zeta\|_\infty + C_{\max}(1-\gamma) \right)}{(1-\gamma)^2} \sqrt{\frac{\ell_k(s,a)}{2N_k^+(s,a)}}$$

$$\overset{(c)}{=} \frac{\gamma C_{\max} \left( R_{\max}(3+\gamma)/\max_{(s,a)}^+ |A_{\mathcal{M}}^{r,\pi^E}| + (1-\gamma) \right)}{(1-\gamma)^2} \sqrt{\frac{\ell_k(s,a)}{2N_k^+(s,a)}} \tag{30}$$

$$= \sigma \sqrt{\frac{\ell_k(s,a)}{2N_k^+(s,a)}} \tag{31}$$

where, for concision, we denote $\sigma = \frac{\gamma C_{\max} \left( R_{\max}(3+\gamma)/\max_{(s,a)}^+ |A_{\mathcal{M}}^{r,\pi^E}| + (1-\gamma) \right)}{(1-\gamma)^2}$.

- (a) uses Lemma 4.6 and the triangular inequality.

- (b) uses Lemma 4.7 and Lemma C.5.

- (c) uses Lemma results of $\|\zeta\|_\infty$ in Lemma 4.5

From Lemma 4.6, since $\frac{2(\chi(s,a)+\chi)}{1+(\chi(s,a)+\chi)/C_{\max}}$ increases monotonically with $\chi(s,a) + \chi$, we have

$$|c(s,a) - \widehat{c}_k(s,a)| \leq \frac{2(\chi(s,a)+\chi)}{1+(\chi(s,a)+\chi)/C_{\max}} = \frac{2\sigma \left( \sqrt{\frac{\ell_k(s,a)}{2N_k^+(s,a)}} + \max\limits_{(s,a)\in\mathcal{S}\times\mathcal{A}} \sqrt{\frac{\ell_k(s,a)}{2N_k^+(s,a)}} \right)}{1 + \sigma/C_{\max} \left( \sqrt{\frac{\ell_k(s,a)}{2N_k^+(s,a)}} + \max\limits_{(s,a)\in\mathcal{S}\times\mathcal{A}} \sqrt{\frac{\ell_k(s,a)}{2N_k^+(s,a)}} \right)}. \tag{32}$$

Also, note that

$$|c(s,a) - \widehat{c}_k(s,a)| \leq \max\{c(s,a), \widehat{c}_k(s,a)\} \leq C_{\max} \tag{33}$$

Thus, the following formula holds true,

$$|c(s,a) - \widehat{c}_k(s,a)| \leq \mathcal{C}_k(s,a), \forall\, (s,a) \in \mathcal{S} \times \mathcal{A}, \tag{34}$$

$$\mathcal{C}_k(s,a) = \min \left\{ \frac{2\sigma \left( \sqrt{\frac{\ell_k(s,a)}{2N_k^+(s,a)}} + \max\limits_{(s,a)\in\mathcal{S}\times\mathcal{A}} \sqrt{\frac{\ell_k(s,a)}{2N_k^+(s,a)}} \right)}{1 + \sigma/C_{\max} \left( \sqrt{\frac{\ell_k(s,a)}{2N_k^+(s,a)}} + \max\limits_{(s,a)\in\mathcal{S}\times\mathcal{A}} \sqrt{\frac{\ell_k(s,a)}{2N_k^+(s,a)}} \right)}, C_{\max} \right\}, \tag{35}$$

$\square$

## C.7. Proof of Lemma 5.1

*Proof.*

$$\|e_k(s,a;\pi^*)\|_\infty \overset{(a)}{\leq} \left\| (I_{\mathcal{S}\times\mathcal{A}} - \gamma P_{\mathcal{T}}\pi^*)^{-1}|c - \widehat{c}_k| \right\|_\infty \overset{(b)}{\leq} \left\| \mu_0^T (I_{\mathcal{S}\times\mathcal{A}} - \gamma P_{\mathcal{T}}\pi)^{-1}\mathcal{C}_k \right\|_\infty. \tag{36}$$

- (a) follows Lemma C.6 (treat $\pi = \pi^*$ and $\widehat{c} = \widehat{c}_k$).

- (b) follows Lemma 4.8.

$\square$

## C.8. Proof of Lemma 5.2

**Lemma C.6.** *For every given policy $\pi$, the first inequality below holds element-wise. For every optimal policies $\pi^* \in \Pi^*_{\mathcal{M} \cup c}$ and $\widehat{\pi}^* \in \Pi^*_{\widehat{\mathcal{M}} \cup \widehat{c}}$ of CMDPs $\mathcal{M} \cup c$ and $\widehat{\mathcal{M}} \cup \widehat{c}$ respectively, the second inequality below holds.*

$$\left| Q^{c,\pi}_{\mathcal{M} \cup c} - Q^{\widehat{c},\pi}_{\mathcal{M} \cup \widehat{c}} \right| \le \left| (I_{\mathcal{S} \times \mathcal{A}} - \gamma P_{\mathcal{T}} \pi)^{-1} | c - \widehat{c} | \right|,$$

$$\max_{\pi \in \{\widehat{\pi}^*, \pi^*\}} \left\| Q^{c,\pi}_{\mathcal{M} \cup c} - Q^{\widehat{c},\pi}_{\mathcal{M} \cup \widehat{c}} \right\|_\infty \le \frac{1}{1-\gamma} \| c - \widehat{c} \|_\infty.$$

*Proof.* We can show that:

$$\left| Q^{c,\pi}_{\mathcal{M} \cup c} - Q^{\widehat{c},\pi}_{\mathcal{M} \cup \widehat{c}} \right| \overset{(a)}{=} \left| (I_{\mathcal{S} \times \mathcal{A}} - \gamma P_{\mathcal{T}} \pi)^{-1} c - (I_{\mathcal{S} \times \mathcal{A}} - \gamma P_{\mathcal{T}} \pi)^{-1} \widehat{c} \right|$$

$$= \left| (I_{\mathcal{S} \times \mathcal{A}} - \gamma P_{\mathcal{T}} \pi)^{-1} | c - \widehat{c} | \right| \tag{37}$$

- (a) results from the matrix representation of Bellman equation, i.e., $Q^{c,\pi}_{\mathcal{M} \cup c} = (I_{\mathcal{S} \times \mathcal{A}} - \gamma P_{\mathcal{T}} \pi)^{-1} c$.

By definition of infinity norm, we have

$$|Q^{c,\pi}_{\mathcal{M} \cup c} - Q^{\widehat{c},\widehat{\pi}}_{\mathcal{M} \cup \widehat{c}}| \le \|Q^{c,\pi}_{\mathcal{M} \cup c} - Q^{\widehat{c},\pi}_{\mathcal{M} \cup \widehat{c}}\|_\infty. \tag{38}$$

Further, we derive the error upper bound of the action-value function by that of cost.

$$\|Q^{c,\pi}_{\mathcal{M} \cup c} - Q^{\widehat{c},\pi}_{\mathcal{M} \cup \widehat{c}}\|_\infty \overset{(b)}{=} \left\| (I_{\mathcal{S} \times \mathcal{A}} - \gamma P_{\mathcal{T}} \pi)^{-1} | c - \widehat{c} | \right\|_\infty$$

$$\overset{(c)}{=} \left\| (I_{\mathcal{S} \times \mathcal{A}} - \gamma P_{\mathcal{T}} \pi)^{-1} \right\|_\infty \| c - \widehat{c} \|_\infty$$

$$\overset{(d)}{\le} \frac{1}{1-\gamma} \| c - \widehat{c} \|_\infty$$

- (b) uses Eq. (37)

- (c) exploits the theorem of matrix infinity norm inequalities that $\|AB\|_\infty \le \|A\|_\infty \|B\|_\infty$

- (d) results from $\|(I_{\mathcal{S} \times \mathcal{A}} - \gamma \pi P_{\mathcal{T}})^{-1}\|_\infty \le \frac{1}{1-\gamma}$.

$\square$

### Final proof of Lemma 5.2

*Proof.* For statement (1),

$$\inf_{\widehat{c}_k \in \mathcal{C}_{\widehat{\mathfrak{P}}_k}} \sup_{\pi^* \in \Pi^*_{\mathcal{M} \cup c}} \left| Q^{c,\pi^*}_{\mathcal{M} \cup c}(s_0, a) - Q^{\widehat{c}_k, \pi^*}_{\mathcal{M} \cup \widehat{c}_k}(s_0, a) \right| \le \inf_{\widehat{c}_k \in \mathcal{C}_{\widehat{\mathfrak{P}}_k}} \sup_{\pi^* \in \Pi^*_{\mathcal{M} \cup c}} \| Q^{c,\pi^*}_{\mathcal{M} \cup c}(s, a) - Q^{\widehat{c}_k, \pi^*}_{\mathcal{M} \cup \widehat{c}_k}(s, a) \|_\infty$$

$$\overset{(a)}{\le} \inf_{\widehat{c}_k \in \mathcal{C}_{\widehat{\mathfrak{P}}_k}} \frac{1}{1-\gamma} \| c(s, a) - \widehat{c}_k(s, a) \|_\infty$$

$$\overset{(b)}{=} \frac{1}{1-\gamma} \max_{(s,a) \in \mathcal{S} \times \mathcal{A}} \mathcal{C}_k(s, a) \le \varepsilon,$$

$$\inf_{c \in \mathcal{C}_{\mathfrak{P}}} \sup_{\widehat{\pi}^*_k \in \Pi^*_{\widehat{\mathcal{M}}_k \cup \widehat{c}_k}} \left| Q^{c,\widehat{\pi}^*_k}_{\mathcal{M} \cup c}(s_0, a) - Q^{\widehat{c}_k, \widehat{\pi}^*_k}_{\mathcal{M} \cup \widehat{c}_k}(s_0, a) \right| \le \inf_{c \in \mathcal{C}_{\mathfrak{P}}} \sup_{\widehat{\pi}^*_k \in \Pi^*_{\widehat{\mathcal{M}}_k \cup \widehat{c}_k}} \| Q^{c,\widehat{\pi}^*_k}_{\mathcal{M} \cup c}(s, a) - Q^{\widehat{c}_k, \widehat{\pi}^*_k}_{\mathcal{M} \cup \widehat{c}_k}(s, a) \|_\infty$$

$$\overset{(c)}{\le} \inf_{c \in \mathcal{C}_{\mathfrak{P}}} \frac{1}{1-\gamma} \| c(s, a) - \widehat{c}_k(s, a) \|_\infty$$

$$\stackrel{(d)}{=} \frac{1}{1 - \gamma} \max_{(s,a) \in \mathcal{S} \times \mathcal{A}} \mathcal{C}_k(s,a) \leq \varepsilon,$$

where steps (a) and (c) use Lemma C.6, and steps (b) and (d) use Lemma 4.8.

For statement (2),

$$\inf_{\widehat{c}_k \in \mathcal{C}_{\widehat{\mathfrak{P}}_k}} \sup_{\pi^* \in \Pi^*_{\mathcal{M} \cup c}} \left| Q^{c,\pi^*}_{\mathcal{M} \cup c}(s_0, a) - Q^{\widehat{c}_k, \pi^*}_{\mathcal{M} \cup \widehat{c}_k}(s_0, a) \right| \stackrel{(e)}{\leq} \inf_{\widehat{c}_k \in \mathcal{C}_{\widehat{\mathfrak{P}}_k}} \max_{\pi \in \Pi^\dagger} \left| (I_{\mathcal{S} \times \mathcal{A}} - \gamma P_{\mathcal{T}} \pi)^{-1} | c - \widehat{c}_k | \right|$$

$$\stackrel{(f)}{\leq} \max_{\pi \in \Pi^\dagger} \left| (I_{\mathcal{S} \times \mathcal{A}} - \gamma P_{\mathcal{T}} \pi)^{-1} \mathcal{C}_k \right|$$

$$\leq \max_{\pi \in \Pi^\dagger} \max_{\mu_0 \in \Delta^{\mathcal{S}}} \left| \mu_0^T (I_{\mathcal{S} \times \mathcal{A}} - \gamma P_{\mathcal{T}} \pi)^{-1} \mathcal{C}_k \right| \leq \varepsilon,$$

$$\inf_{c \in \mathcal{C}_{\mathfrak{P}}} \sup_{\widehat{\pi}^*_k \in \Pi^*_{\widehat{\mathcal{M}}_k \cup \widehat{c}_k}} \left| Q^{c,\widehat{\pi}^*_k}_{\mathcal{M} \cup c}(s_0, a) - Q^{\widehat{c}_k, \widehat{\pi}^*_k}_{\mathcal{M} \cup \widehat{c}_k}(s_0, a) \right| \stackrel{(g)}{\leq} \inf_{c \in \mathcal{C}_{\mathfrak{P}}} \max_{\pi \in \Pi^\dagger} \left| (I_{\mathcal{S} \times \mathcal{A}} - \gamma P_{\mathcal{T}} \pi)^{-1} | c - \widehat{c}_k | \right|$$

$$\stackrel{(h)}{\leq} \max_{\pi \in \Pi^\dagger} \left| (I_{\mathcal{S} \times \mathcal{A}} - \gamma P_{\mathcal{T}} \pi)^{-1} \mathcal{C}_k \right|$$

$$\leq \max_{\pi \in \Pi^\dagger} \max_{\mu_0 \in \Delta^{\mathcal{S}}} \left| \mu_0^T (I_{\mathcal{S} \times \mathcal{A}} - \gamma P_{\mathcal{T}} \pi)^{-1} \mathcal{C}_k \right| \leq \varepsilon,$$

where steps (e) and (g) use Eq. (37) and the fact that each cost function in the feasible cost set must ensure the feasibility of expert policy, steps (f) and (h) use Lemma 4.8. $\qquad\square$

### C.9. Uniform Sampling Strategy for ICRL with a Generative Model

In this part, we additionally consider the problem setting where the agent does not employ any exploration strategy to acquire desired information but utilizes a uniform sampling strategy to query a generative model. The problem setting is based on the following assumption, which is stronger than the assumption in the main paper.

**Assumption C.7.** The agent have access to the *generative model* of $\mathcal{M}$;

More specifically, the agent can always query a generative model about a state-action pair $(s, a)$ to receive a next state $s' \sim P(\cdot | s, a)$ and about a state $s$ to receive an expert action $a^E \sim \pi^E(\cdot | s)$. We first present Alg. 2 for uniform sampling strategy with the generative model and study the sample complexity of this algorithm in Theorem C.9.

---

**Algorithm 2** Uniform Sampling Strategy for ICRL

**Input:** significance $\delta \in (0, 1)$, target accuracy $\varepsilon$, maximum number of samples per iteration $n_{\max}$
Initialize $k \leftarrow 0$, $\varepsilon_0 = \frac{1}{1-\gamma}$
**while** $\varepsilon_k > \varepsilon$ **do**
    Collect $\lceil \frac{n_{\max}}{SA} \rceil$ samples from each $(s, a) \in \mathcal{S} \times \mathcal{A}$
    Update accuracy $\varepsilon_{k+1} = \frac{1}{1-\gamma} \max_{(s,a) \in \mathcal{S} \times \mathcal{A}} \mathcal{C}_{k+1}(s, a)$
    Update $\widehat{\pi}^E_{k+1}(a|s)$ and $\widehat{P}_{\mathcal{T}k+1}(s'|s, a)$ in (7)
    $k \leftarrow k + 1$
**end while**

---

**Lemma C.8.** *(Metelli et al., 2021, Lemma B.8). Let $a, b \geq 0$ such that $2a\sqrt{b} > $ e. Then, the inequality $x \geq a \log(bx^2)$ is satisfied for all $x \geq -2aW_{-1}\left(-\frac{1}{2a\sqrt{b}}\right)$, where $W_{-1}$ is the secondary component of the Lambert W function. Moreover, $-2aW_{-1}\left(-\frac{1}{2a\sqrt{b}}\right) \leq 4a \log(2a\sqrt{b})$.*

**Theorem C.9.** *(Sample Complexity of Uniform Sampling Strategy). If Algorithm 2 stops at iteration $K$ with accuracy $\varepsilon_K$, then with probability at least $1 - \delta$, it fulfills Definition 4.9 with a number of samples upper bounded by,*

$$n \leq \widetilde{\mathcal{O}}\left( \frac{\sigma^2 SA}{(1 - \gamma)^2 \varepsilon_K^2} \right), \tag{39}$$

*where* $\sigma = \dfrac{\gamma C_{\max}\left(R_{\max}(3+\gamma)/\max^+_{(s,a)}|A^{r,\pi^E}_{\mathcal{M}}|+(1-\gamma)\right)}{(1-\gamma)^2}$ *and* $\widetilde{\mathcal{O}}$ *notation suppresses logarithmic terms.*

*Proof.* We start from Lemma 5.2. We further bound:

$$
\begin{aligned}
\frac{1}{1-\gamma}\max_{(s,a)\in\mathcal{S}\times\mathcal{A}}\mathcal{C}_k(s,a) &= \frac{1}{1-\gamma}\max_{(s,a)\in\mathcal{S}\times\mathcal{A}}\min\left\{\frac{2\sigma\left(\sqrt{\frac{\ell_k(s,a)}{2N_k^+(s,a)}}+\max_{(s,a)}\sqrt{\frac{\ell_k(s,a)}{2N_k^+(s,a)}}\right)}{1+\frac{\sigma}{C_{\max}}\left(\sqrt{\frac{\ell_k(s,a)}{2N_k^+(s,a)}}+\max_{(s,a)}\sqrt{\frac{\ell_k(s,a)}{2N_k^+(s,a)}}\right)},C_{\max}\right\}\\
&= \frac{1}{1-\gamma}\min\left\{\frac{4\sigma\max_{(s,a)}\sqrt{\frac{\ell_k(s,a)}{2N_k^+(s,a)}}}{1+\frac{2\sigma}{C_{\max}}\max_{(s,a)}\sqrt{\frac{\ell_k(s,a)}{2N_k^+(s,a)}}},C_{\max}\right\}\\
&\overset{(a)}{=} \frac{1}{1-\gamma}\min\left\{\frac{4\sigma\sqrt{\frac{\ell_k(s',a')}{2N_k^+(s',a')}}}{1+\frac{2\sigma}{C_{\max}}\sqrt{\frac{\ell_k(s',a')}{2N_k^+(s',a')}}},C_{\max}\right\}
\end{aligned}
\tag{40}
$$

where step (a) supposes at state-action pair $(s',a')$, $\sigma\sqrt{\frac{\ell_k(s',a')}{2N_k^+(s',a')}}=\sigma\max_{(s',a')}\sqrt{\frac{\ell_k(s',a')}{2N_k^+(s',a')}}$.

After $K$ iterations, based on uniform sampling strategy, we know that $N_K \geq 1$ for any $(s,a)\in\mathcal{S}\times\mathcal{A}$. To terminate at iteration $K$, it suffices to enforce every $(s,a)\in\mathcal{S}\times\mathcal{A}$:

$$
\frac{1}{1-\gamma}\frac{4\sigma\sqrt{\frac{\ell_k(s',a')}{2N_k^+(s',a')}}}{1+\frac{2\sigma}{C_{\max}}\sqrt{\frac{\ell_k(s',a')}{2N_k^+(s',a')}}} = \varepsilon_K
$$

$$
\sigma\sqrt{\frac{\ell_k(s',a')}{2N_k^+(s',a')}} = \frac{C_{\max}\varepsilon_K(1-\gamma)}{4C_{\max}-2\varepsilon_K(1-\gamma)}
$$

By $\sigma = \dfrac{\gamma C_{\max}\left(R_{\max}(3+\gamma)/\max^+_{(s,a)}|A^{r,\pi^E}_{\mathcal{M}}|+(1-\gamma)\right)}{(1-\gamma)^2}$, we have

$$
\frac{\gamma C_{\max}\left(R_{\max}(3+\gamma)/\max^+_{(s,a)}|A^{r,\pi^E}_{\mathcal{M}}|+(1-\gamma)\right)}{(1-\gamma)^3}\sqrt{\frac{\ell_k(s',a')}{2N_k^+(s',a')}} = \frac{C_{\max}\varepsilon_K}{4C_{\max}-2\varepsilon_K(1-\gamma)}
$$

$$
\implies N_K(s',a') = \frac{2\gamma^2(2C_{\max}-\varepsilon_K(1-\gamma))^2\left(R_{\max}(3+\gamma)/\max^+_{(s,a)}|A^{r,\pi^E}_{\mathcal{M}}|+(1-\gamma)\right)^2\ell_k(s',a')}{(1-\gamma)^6\varepsilon_K^2}
$$

From Lemma C.8, we derive

$$
\begin{aligned}
&N_K(s',a')\\
&= -\frac{4\gamma^2(2C_{\max}-\varepsilon_K(1-\gamma))^2\left(R_{\max}(3+\gamma)/\max^+_{(s,a)}|A^{r,\pi^E}_{\mathcal{M}}|+(1-\gamma)\right)^2}{(1-\gamma)^6\varepsilon_K^2}\times\\
&\quad W_{-1}\left(-\frac{(1-\gamma)^6\varepsilon_K^2}{4\gamma^2(2C_{\max}-\varepsilon_K(1-\gamma))^2\left(R_{\max}(3+\gamma)/\max^+_{(s,a)}|A^{r,\pi^E}_{\mathcal{M}}|+(1-\gamma)\right)^2}\sqrt{\frac{\delta}{36SA}}\right)\\
&\leq \frac{8\gamma^2(2C_{\max}-\varepsilon_K(1-\gamma))^2\left(R_{\max}(3+\gamma)/\max^+_{(s,a)}|A^{r,\pi^E}_{\mathcal{M}}|+(1-\gamma)\right)^2}{(1-\gamma)^6\varepsilon_K^2}\times
\end{aligned}
$$

$$\log\left(\frac{4\gamma^2(2C_{\max}-\varepsilon_K(1-\gamma))^2\left(R_{\max}(3+\gamma)/\max_{(s,a)}^+|A_{\mathcal{M}}^{r,\pi^E}|+(1-\gamma)\right)^2}{(1-\gamma)^6\varepsilon_K^2}\sqrt{\frac{36SA}{\delta}}\right)$$

$$=\widetilde{\mathcal{O}}\left(\frac{\gamma^2(2C_{\max}-\varepsilon_K(1-\gamma))^2\left(R_{\max}(3+\gamma)/\max_{(s,a)}^+|A_{\mathcal{M}}^{r,\pi^E}|+(1-\gamma)\right)^2}{(1-\gamma)^6\varepsilon_K^2}\right)$$

$$=\widetilde{\mathcal{O}}\left(\frac{\gamma^2(2C_{\max}-\varepsilon_K(1-\gamma))^2\left(R_{\max}(3+\gamma)/\max_{(s,a)}^+|A_{\mathcal{M}}^{r,\pi^E}|+(1-\gamma)\right)^2}{(1-\gamma)^6\varepsilon_K^2}\right) \tag{41}$$

By summing $n=\sum_{(s,a)\in\mathcal{S}\times\mathcal{A}}N_K(s,a)$, we obtain the upper bound.

$$n\leq\widetilde{\mathcal{O}}\left(\frac{\gamma^2(2C_{\max}-\varepsilon_K(1-\gamma))^2\left(R_{\max}(3+\gamma)/\max_{(s,a)}^+|A_{\mathcal{M}}^{r,\pi^E}|+(1-\gamma)\right)^2SA}{(1-\gamma)^6\varepsilon_K^2}\right) \tag{42}$$

Since $\sigma=\frac{\gamma C_{\max}\left(R_{\max}(3+\gamma)/\max_{(s,a)}^+|A_{\mathcal{M}}^{r,\pi^E}|+(1-\gamma)\right)}{(1-\gamma)^2}$, we have

$$n\leq\widetilde{\mathcal{O}}\left(\frac{\sigma^2(2C_{\max}-\varepsilon_K(1-\gamma))^2SA}{(1-\gamma)^2\varepsilon_K^2C_{\max}^2}\right). \tag{43}$$

Regarding the sample complexity in the RL phase, since the reward function is known, according to Corollary 2.7 in Section 2.3.1 from the book 'Reinforcement Learning: Theory and Algorithms' (Agarwal et al., 2019), the sample complexity of obtaining a $\varepsilon$-optimal policy is $O(SA/(1-\gamma)^3\varepsilon^2)$, which is dominated by the sample complexity in Theorem 5.5. Note that $\sigma$ also contains $1/(1-\gamma)$. As a result, Eq. (43) still holds true after including the sample complexity of the RL phase. $\quad\square$

### C.10. Proof of Lemma 5.4

*Proof.* This result generalizes (Kaufmann et al., 2021, Lemma 7) to our setting. We define event $\mathcal{G}^{\mathrm{cnt}}$ as:

$$\mathcal{G}^{\mathrm{cnt}}\quad=\quad\left\{\forall k\in\mathbb{N}^\star,\forall(s,a)\in\mathcal{S}\times\mathcal{A}:N_k(s,a)\geq\frac{1}{2}\bar{N}_k(s,a)-\log\left(\frac{2SA}{\delta}\right)\right\}. \tag{44}$$

We calculate the probability of the complement of event $\mathcal{G}^{\mathrm{cnt}}$.

$$\mathbb{P}\left((\mathcal{G}^{\mathrm{cnt}})^c\right)$$
$$\overset{(a)}{\leq}\sum_{(s,a)\in\mathcal{S}\times\mathcal{A}}\mathbb{P}\left(\exists k\in\mathbb{N}:N_k(s,a)\leq\frac{1}{2}\bar{N}_k(s,a)-\log\left(\frac{2SA}{\delta}\right)\right)$$
$$\overset{(b)}{\leq}\sum_{(s,a)\in\mathcal{S}\times\mathcal{A}}\mathbb{P}\left(\exists k\in\mathbb{N}:\sum_{h=1}^{n_{\max}}\sum_{i=1}^k\mathbb{1}\left((s_i^h,a_i^h)=(s,a)\right)\leq\frac{1}{2}\sum_{s_0}\sum_{h=1}^{n_{\max}}\sum_{i=1}^k\mu_0(s_0)\eta_i^h(s,a|s_0)-\log\left(\frac{2SA}{\delta}\right)\right)$$
$$\overset{(c)}{\leq}\sum_{(s,a)\in\mathcal{S}\times\mathcal{A}}\frac{\delta}{2SA}=\frac{\delta}{2}, \tag{45}$$

- (a) results from a union bound over $(s,a)$.

- (b) results from Definition 5.3.

- (c) results from (Kaufmann et al., 2021, Lemma 9).

As a result, we have with probability at least $1 - \delta/2$:

$$N_k(s,a) \geq \frac{1}{2}\bar{N}_k(s,a) - \beta_{\mathrm{cnt}}(\delta), \tag{46}$$

where $\beta_{\mathrm{cnt}}(\delta) = \log\left(2SA/\delta\right)$.

The following proof is adapted from Lemma B.18 in (Lindner et al., 2022). Distinguish two cases. First, let $\beta_{\mathrm{cnt}}(\delta) \leq \frac{1}{4}\bar{N}_k(s,a)$. Then $N_k(s,a) \geq \frac{1}{4}\bar{N}_k(s,a)$, and

$$
\begin{aligned}
\min\left\{\sigma\sqrt{\frac{\ell_k(s,a)}{2N_k^+(s,a)}}, C_{\max}\right\} &\leq \sigma\sqrt{\frac{\ell_k(s,a)}{2N_k^+(s,a)}} = \sigma\sqrt{\frac{\log(36SA(N_k^+(s,a))^2/\delta)}{2N_k^+(s,a)}} \\
&\leq \sigma\sqrt{\frac{\log(36SA(\bar{N}_k^+(s,a)/4)^2/\delta)}{\bar{N}_k^+(s,a)/2}} \leq \sigma\sqrt{\frac{2\bar{\ell}_k(s,a)}{\bar{N}_k^+(s,a)}},
\end{aligned}
\tag{47}
$$

where we use that $\log(36SAx^2/\delta)/x$ is non-increasing for $x > \mathrm{e}\sqrt{\frac{\delta}{36SA}}$, where $\mathrm{e}$ is Euler's number.

For the second case, let $\beta_{\mathrm{cnt}}(\delta) > \frac{1}{4}\bar{N}_k(s,a)$. Then,

$$\min\left\{\sigma\sqrt{\frac{\ell_k(s,a)}{2N_k^+(s,a)}}, C_{\max}\right\} \leq C_{\max} < C_{\max}\sqrt{\frac{4\beta_{\mathrm{cnt}}(\delta)}{\bar{N}_k^+(s,a)}} \leq C_{\max}\sqrt{\frac{4\bar{\ell}_k(s,a)}{\bar{N}_k^+(s,a)}}, \tag{48}$$

where we use $\bar{\ell}_k(s,a) = \log\left(36SA(\bar{N}_k^+(s,a))^2/\delta\right) = \beta_{\mathrm{cnt}}(\delta) + \log\left(18(\bar{N}_k^+(s,a))^2\right) \geq \beta_{\mathrm{cnt}}(\delta)$. By combining the two cases, we obtain

$$\min\left\{\sigma\sqrt{\frac{\ell_k(s,a)}{2N_k^+(s,a)}}, C_{\max}\right\} \leq \max\{\sigma, \sqrt{2}C_{\max}\}\sqrt{\frac{2\bar{\ell}_k(s,a)}{\bar{N}_k^+(s,a)}} = \check{\sigma}\sqrt{\frac{2\bar{\ell}_k(s,a)}{\bar{N}_k^+(s,a)}}, \tag{49}$$

where we denote $\check{\sigma} = \max\{\sigma, \sqrt{2}C_{\max}\}$. $\qquad\square$

### C.11. Proof of Theorem 5.5

*Proof.* We assume BEAR exploration strategy terminates after $K$ iterations, then

$$
\begin{aligned}
\frac{1}{1-\gamma}\max_{(s,a)}\mathcal{C}_K(s,a) &\overset{(a)}{=} \frac{1}{1-\gamma}\max_{(s,a)\in\mathcal{S}\times\mathcal{A}}\min\left\{\frac{2\sigma\left(\sqrt{\frac{\ell_k(s,a)}{2N_k^+(s,a)}} + \max\limits_{(s,a)}\sqrt{\frac{\ell_k(s,a)}{2N_k^+(s,a)}}\right)}{1+\frac{\sigma}{C_{\max}}\left(\sqrt{\frac{\ell_k(s,a)}{2N_k^+(s,a)}} + \max\limits_{(s,a)}\sqrt{\frac{\ell_k(s,a)}{2N_k^+(s,a)}}\right)}, C_{\max}\right\} \\
&\overset{(b)}{\leq} \frac{1}{1-\gamma}\max_{(s,a)\in\mathcal{S}\times\mathcal{A}}\min\left\{\frac{2\check{\sigma}\left(\sqrt{\frac{2\bar{\ell}_K(s,a)}{\bar{N}_K^+(s,a)}} + \max\limits_{(s,a)}\sqrt{\frac{2\bar{\ell}_K(s,a)}{\bar{N}_K^+(s,a)}}\right)}{1+\frac{\check{\sigma}}{C_{\max}}\left(\sqrt{\frac{2\bar{\ell}_K(s,a)}{\bar{N}_K^+(s,a)}} + \max\limits_{(s,a)}\sqrt{\frac{2\bar{\ell}_K(s,a)}{\bar{N}_K^+(s,a)}}\right)}, C_{\max}\right\} \\
&\overset{(c)}{=} \frac{1}{1-\gamma}\min\left\{\frac{4\check{\sigma}\sqrt{\frac{2\bar{\ell}_K(s',a')}{\bar{N}_K^+(s',a')}}}{1+\frac{2\check{\sigma}}{C_{\max}}\sqrt{\frac{2\bar{\ell}_K(s',a')}{\bar{N}_K^+(s',a')}}}, C_{\max}\right\}
\end{aligned}
$$

where step (a) follows Lemma 4.8, step (b) results from Lemma 5.4 and step (c) assumes at state-action pair $(s',a')$ $\sqrt{\frac{2\bar{\ell}_K(s',a')}{\bar{N}_K^+(s',a')}} = \max\limits_{(s,a)}\sqrt{\frac{2\bar{\ell}_K(s,a)}{\bar{N}_K^+(s,a)}}$. Hence, we obtain,

$$\varepsilon_K = \frac{1}{1-\gamma}\frac{4\check{\sigma}\sqrt{\frac{2\bar{\ell}_K(s',a')}{\bar{N}_K^+(s',a')}}}{1+\frac{2\check{\sigma}}{C_{\max}}\sqrt{\frac{2\bar{\ell}_K(s',a')}{\bar{N}_K^+(s',a')}}} \tag{50}$$

$$\frac{C_{\max}\varepsilon_K}{4C_{\max} - 2\varepsilon_K(1-\gamma)} = \frac{\check{\sigma}}{1-\gamma}\sqrt{\frac{2\bar{\ell}_K(s',a')}{\bar{N}_K^+(s',a')}} = \frac{\check{\sigma}}{1-\gamma}\sqrt{\frac{2\log(36SA(\bar{N}_K^+(s',a'))^2/\delta)}{\bar{N}_K^+(s',a')}}$$

Thus,

$$\bar{N}_K^+(s,a) \geq \frac{8\check{\sigma}^2(2C_{\max} - \varepsilon_K(1-\gamma))^2\log(36SA(\bar{N}_K^+(s,a))^2/\delta)}{(1-\gamma)^2\varepsilon_K^2 C_{\max}^2}$$

From Lemma C.8, we have

$$\begin{aligned}
\bar{N}_K^+(s,a) &= -\frac{16\check{\sigma}^2(2C_{\max} - \varepsilon_K(1-\gamma))^2}{(1-\gamma)^2\varepsilon_K^2 C_{\max}^2}W_{-1}\left(-\frac{(1-\gamma)^2\varepsilon_K^2 C_{\max}^2}{16\check{\sigma}^2(2C_{\max} - \varepsilon_K(1-\gamma))^2}\sqrt{\frac{\delta}{36SA}}\right) \\
&\leq \frac{32\check{\sigma}^2(2C_{\max} - \varepsilon_K(1-\gamma))^2}{(1-\gamma)^2\varepsilon_K^2 C_{\max}^2}\log\left(\frac{32\check{\sigma}^2(2C_{\max} - \varepsilon_K(1-\gamma))^2}{(1-\gamma)^2\varepsilon_K^2 C_{\max}^2}\sqrt{\frac{36SA}{\delta}}\right) \\
&= \widetilde{\mathcal{O}}\left(\frac{\check{\sigma}^2(2C_{\max} - \varepsilon_K(1-\gamma))^2}{(1-\gamma)^2\varepsilon_K^2 C_{\max}^2}\right)
\end{aligned} \tag{51}$$

By summing over $n = \sum_{(s,a)\in\mathcal{S}\times\mathcal{A}}\bar{N}_K^+(s,a)$, we obtain the upper bound.

$$n \leq \widetilde{\mathcal{O}}\left(\frac{\check{\sigma}^2(2C_{\max} - \varepsilon_K(1-\gamma))^2}{(1-\gamma)^2\varepsilon_K^2 C_{\max}^2}\right), \tag{52}$$

where $\check{\sigma} = \max\{\sigma, \sqrt{2}C_{\max}\}$.

Regarding the sample complexity in the RL phase, since the reward function is known, according to Corollary 2.7 in Section 2.3.1 from the book 'Reinforcement Learning: Theory and Algorithms' (Agarwal et al., 2019), the sample complexity of obtaining a $\varepsilon$-optimal policy is $O(SA/(1-\gamma)^3\varepsilon^2)$, which is dominated by the sample complexity in Theorem 5.5. Note that $\sigma$ also contains $1/(1-\gamma)$. As a result, Eq. (43) still holds true after including the sample complexity of the RL phase. $\qquad\square$

## C.12. Theoretical Results on Policy-Constrained Strategic Exploration (PCSE)

**Definition C.10.** We define the optimal policy w.r.t. cost, reward, and safety as follows:

- The cost minimization policy: $\pi^{c,*} = \arg\min_{\pi\in\Pi}\mathbb{E}[\sum_t \gamma^t c(s_t,a_t)]$.

- The reward maximization policy: $\pi^{r,*} = \arg\max_{\pi\in\Pi}\mathbb{E}[\sum_t \gamma^t r(s_t,a_t)]$.

- The optimal safe policy: $\pi^* = \arg\max_{\pi\in\Pi_{safe}}\mathbb{E}[\sum_t \gamma^t r(s_t,a_t)]$ where $\Pi_{safe} = \{\pi : \mathbb{E}[\sum_t \gamma^t c(s_t,a_t)] \leq \epsilon\}$

Accordingly, we can have the following relations:

- $\mathbb{E}_{\mu_0}[V^{c,\pi^{c,*}}(s_0)] \leq \mathbb{E}_{\mu_0}[V^{c,\pi^*}(s_0)] \leq \mathbb{E}_{\mu_0}[V^{c,\pi^{c,*}}(s_0)] + \epsilon$ where the equality normally holds that $V^{c,\pi^{c,*}}(s_0) = 0$.

- $\mathbb{E}_{\mu_0}[V^{r,\pi^*}(s_0)] \leq \mathbb{E}_{\mu_0}[V^{r,\pi^{r,*}}(s_0)]$.

Let's define the following symbols:

- $\varepsilon_0 = \frac{1}{4(1-\gamma)}$.

- $\varepsilon_k^\pi = \sup_{\mu_0\in\Delta^{S\times A}}\mu_0^T(I_{\mathcal{S}\times\mathcal{A}} - \gamma P_{\mathcal{T}}\pi)\mathcal{C}_k$

- $\varepsilon_k = \max_{\pi\in\Pi_{k-1}}\varepsilon_k^\pi$

We can construct a set of plausibly optimal policies as

$$\Pi_k = \Pi_k^c \cap \Pi_k^r$$

$$\Pi_k^c = \left\{ \pi \in \Delta_{\mathcal{S}}^{\mathcal{A}} : \sup_{\mu_0 \in \Delta^{\mathcal{S}}} \mu_0^T (V_{\widehat{\mathcal{M}} \cup \widehat{c}_k}^{c,\pi} - V_{\widehat{\mathcal{M}} \cup \widehat{c}_k}^{c,*}) \leq 4\varepsilon_k + 2\epsilon \right\}$$

$$\Pi_k^r = \left\{ \pi \in \Delta_{\mathcal{S}}^{\mathcal{A}} : \inf_{\mu_0 \in \Delta^{\mathcal{S}}} \mu_0^T \left( V_{\widehat{\mathcal{M}}}^{r,\pi} - V_{\widehat{\mathcal{M}}}^{r,\widehat{\pi}^*} \right) \geq \mathfrak{R}_k \right\},$$

where $\mathfrak{R}_k = \frac{2\gamma R_{\max}}{(1-\gamma)^2} \|P_{\mathcal{T}} - \widehat{P_{\mathcal{T}}}\|_\infty + \frac{\gamma R_{\max}}{(1-\gamma)^2} \|(\pi^* - \widehat{\pi}^*)\|_\infty$.

**Lemma C.11.** *($\pi^*$ propagation). Under the good event $\mathcal{E}_k$, if $\pi^*, \widehat{\pi}_k^* \in \Pi_{k-1}^c$ then $\pi^* \in \Pi_k^c$*

*Proof.* Given a $c \in \mathcal{C}_{\mathfrak{P}}$, we can show:

$$\sup_{\mu_0 \in \Delta^{\mathcal{S}}} \mu_0^T \left( V_{\widehat{\mathcal{M}} \cup \widehat{c}_k}^{c,\pi^*} - V_{\widehat{\mathcal{M}} \cup \widehat{c}_k}^{c,*} \right) = \sup_{\mu_0 \in \Delta^{\mathcal{S}}} \mu_0^T \left( V_{\widehat{\mathcal{M}} \cup \widehat{c}_k}^{c,\pi^*} - V_{\widehat{\mathcal{M}} \cup c}^{c,\pi^*} + V_{\widehat{\mathcal{M}} \cup c}^{c,\pi^*} - V_{\widehat{\mathcal{M}} \cup \widehat{c}_k}^{c,*} \right) \tag{53}$$

$$\overset{(i)}{\leq} \sup_{\mu_0 \in \Delta^{\mathcal{S}}} \mu_0^T \left( \varepsilon_k + V_{\widehat{\mathcal{M}} \cup c}^{c,\pi^*} - V_{\widehat{\mathcal{M}} \cup \widehat{c}_k}^{c,*} \right)$$

$$\overset{(ii)}{\leq} 2\varepsilon_k + 2\epsilon,$$

which demonstrates that $\pi^* \in \Pi_k^c$.

- (i) holds since

$$|V_{\widehat{\mathcal{M}} \cup \widehat{c}_k}^{c,\pi^*} - V_{\widehat{\mathcal{M}} \cup c}^{c,\pi^*}| \leq (I_{\mathcal{S}} - \gamma \pi^* P_{\mathcal{T}})^{-1} \pi^* |\widehat{c}_k - c|$$

$$\leq (I_{\mathcal{S}} - \gamma \pi^* P_{\mathcal{T}})^{-1} \pi^* \mathcal{C}_k,$$

  where

  – The first inequality follows (Metelli et al., 2021, Lemma B.2) (treat $\widehat{r}_k = -\widehat{c}_k$ and $r = -c$).
  – The second inequality is due to the good event definition in Lemma C.5.

  As a result:

$$\sup_{\mu_0 \in \Delta^{\mathcal{S}}} \mu_0^T \left( V_{\widehat{\mathcal{M}} \cup \widehat{c}_k}^{c,\pi^*} - V_{\widehat{\mathcal{M}} \cup c}^{c,\pi^*} \right) = \varepsilon_k^{\pi^*} \leq \max_{\pi \in \Pi_{k-1}^c} \varepsilon_k^\pi = \varepsilon_k \tag{54}$$

- (ii) holds since

$$V_{\widehat{\mathcal{M}} \cup c}^{c,\pi^*} - V_{\widehat{\mathcal{M}} \cup \widehat{c}_k}^{c,*} = V_{\widehat{\mathcal{M}} \cup c}^{c,\pi^*} - V_{\widehat{\mathcal{M}} \cup \widehat{c}_k}^{c,\widehat{\pi}_k^{c,*}}$$

$$\leq V_{\widehat{\mathcal{M}} \cup c}^{c,\pi^{c,*}} - V_{\widehat{\mathcal{M}} \cup \widehat{c}_k}^{c,\widehat{\pi}_k^{c,*}} + \epsilon$$

$$= \min_\pi V_{\widehat{\mathcal{M}} \cup c}^{c,\pi} - \min_\pi V_{\widehat{\mathcal{M}} \cup \widehat{c}_k}^{c,\pi} + \epsilon$$

$$\leq \min_{\pi' \in \Pi_{k-1}^c} V_{\widehat{\mathcal{M}} \cup c}^{c,\pi'} - \min_{\pi' \in \Pi_{k-1}^c} V_{\widehat{\mathcal{M}} \cup \widehat{c}_k}^{c,\pi'} + 2\epsilon$$

$$\leq \max_{\pi \in \Pi_{k-1}^c} \left| V_{\widehat{\mathcal{M}} \cup \widehat{c}_k}^{c,\pi} - V_{\widehat{\mathcal{M}} \cup c}^{c,\pi} \right| + 2\epsilon,$$

  where

  – The first inequality utilizes $\mathbb{E}_{\mu_0}[V^{c,\pi^{c,*}}(s_0)] + \epsilon \geq \mathbb{E}_{\mu_0}[V^{c,\pi^*}(s_0)]$.
  – The second inequality utilizes $\forall c, \mathbb{E}_{\mu_0}[V^{c,\pi^{c,*}}(s_0)] \leq \mathbb{E}_{\mu_0}[V^{c,\pi^*}(s_0)] \leq \mathbb{E}_{\mu_0}[V^{c,\pi^{c,*}}(s_0)] + \epsilon$ for $\epsilon > 0$ and the assumption that $\pi^*, \widehat{\pi}_k^* \in \Pi_{k-1}^c$.

– The third inequality results from Lemma C.12.

By following the inequality (54), we have:

$$\max_{\pi \in \Pi_{k-1}^c} \sup_{\mu_0 \in \Delta^S} \mu_0^T \left( V_{\widehat{\mathcal{M}} \cup \widehat{c}_k}^{c,\pi} - V_{\widehat{\mathcal{M}} \cup c}^{c,\pi} \right) + 2\epsilon = \varepsilon_k + 2\epsilon$$

$\square$

**Lemma C.12.**

$$\max_x f(x) - \max_x g(x) \leq \max_x (f(x) - g(x))$$
$$\min_x f(x) - \min_x g(x) \leq \max_x (f(x) - g(x))$$

*Proof.* For the first inequality, suppose $x_1 = \arg\max f(x)$ and $x_2 = \arg\max g(x)$, then we have,

$$\max_x f(x) - \max_x g(x) = f(x_1) - g(x_2) \leq f(x_1) - g(x_1) \leq \max_x (f(x) - g(x))$$

For the second inequality, suppose $x_3 = \arg\min f(x)$ and $x_4 = \arg\min g(x)$, then we have,

$$\min_x f(x) - \min_x g(x) = f(x_3) - g(x_4) \leq f(x_4) - g(x_4) \leq \max_x (f(x) - g(x))$$

$\square$

**Lemma C.13.** *Under the good event $\mathcal{E}_k$, if $\widehat{\pi}_k^*, \xi \in \Pi_{k-1}^c$ and $\xi \notin \Pi_k^c$, then $\xi$ is suboptimal for some cost $\widehat{c}_{k'} \in \mathcal{R}_{\widehat{\mathfrak{P}}_{k'}}$ for all $k' \geq k$.*

*Proof.* Let's consider the following decomposition:

$$
\begin{aligned}
V_{\widehat{\mathcal{M}} \cup \widehat{c}_{k'}}^{c,\xi} - V_{\widehat{\mathcal{M}} \cup \widehat{c}_{k'}}^{c,*} &\overset{(i)}{\geq} V_{\widehat{\mathcal{M}} \cup \widehat{c}_{k'}}^{c,\xi} - V_{\widehat{\mathcal{M}} \cup \widehat{c}_{k'}}^{c,\widehat{\pi}_k^{c,*}} \\
&= V_{\widehat{\mathcal{M}} \cup \widehat{c}_{k'}}^{c,\xi} - V_{\widehat{\mathcal{M}} \cup \widehat{c}_k}^{c,\xi} + V_{\widehat{\mathcal{M}} \cup \widehat{c}_k}^{c,\xi} - V_{\widehat{\mathcal{M}} \cup \widehat{c}_k}^{c,\widehat{\pi}_k^{c,*}} + V_{\widehat{\mathcal{M}} \cup \widehat{c}_k}^{c,\widehat{\pi}_k^{c,*}} - V_{\widehat{\mathcal{M}} \cup \widehat{c}_{k'}}^{c,\widehat{\pi}_k^{c,*}} \\
&\overset{(ii)}{\geq} -4\varepsilon_k + V_{\widehat{\mathcal{M}} \cup \widehat{c}_k}^{c,\xi} - V_{\widehat{\mathcal{M}} \cup \widehat{c}_k}^{c,\widehat{\pi}_k^{c,*}} \\
&\overset{(iii)}{>} 2\epsilon
\end{aligned}
$$

which indicates that $\xi$ cannot be optimal for $k' \geq k$.

- (i) holds since $V_{\widehat{\mathcal{M}} \cup \widehat{c}_{k'}}^{c,\widehat{\pi}_k^{c,*}} \geq V_{\widehat{\mathcal{M}} \cup \widehat{c}_{k'}}^{c,\widehat{\pi}_{k'}^{c,*}} = V_{\widehat{\mathcal{M}} \cup \widehat{c}_{k'}}^{c,*}$

- (ii) holds by following (Metelli et al., 2021, Lemma B.5) (treat $\pi = \widehat{\pi}_k^{c,*}$ and $\pi = \xi$ respectively, while $\widehat{r}_k = \widehat{c}_k$ and $\widehat{r}_{k'} = \widehat{c}_{k'}$)

$$\sup_{\mu_0 \in \Delta^S} \mu_0^T \left( V_{\widehat{\mathcal{M}} \cup \widehat{c}_{k'}}^{c,\widehat{\pi}_k^{c,*}} - V_{\widehat{\mathcal{M}} \cup \widehat{c}_k}^{c,\widehat{\pi}_k^{c,*}} \right) \leq 2\varepsilon_k^{\widehat{\pi}_k^{c,*}} \leq 2\varepsilon_k$$
$$\sup_{\mu_0 \in \Delta^S} \mu_0^T \left( V_{\widehat{\mathcal{M}} \cup \widehat{c}_k}^{c,\xi} - V_{\widehat{\mathcal{M}} \cup \widehat{c}_{k'}}^{c,\xi} \right) \leq 2\varepsilon^\xi \leq 2\varepsilon_k$$

- (iii) holds since according to the definition of $\Pi_k^c$ and considering our assumption that $\xi \notin \Pi_k^c$, we have:

$$\sup_{\mu_0 \in \Delta^S} \mu_0^T \left( V_{\widehat{\mathcal{M}} \cup \widehat{c}_k}^{c,\xi} - V_{\widehat{\mathcal{M}} \cup \widehat{c}_k}^{c,*} \right) > 4\varepsilon_k + 2\epsilon$$

$\square$

**Lemma C.14.** *If $\varepsilon_0 = \frac{1}{4(1-\gamma)}$, then for every $k \geq 0$, it holds that $\pi^*, \widehat{\pi}^*_{k+1} \in \Pi^c_k$.*

*Proof.* We prove the result by induction on $k$. For $k = 0$, for every policy $\pi \in \Delta^{\mathcal{A}}_{\mathcal{S}}$, we have $\sup_{\mu_0 \in \Delta^{\mathcal{S}}} \mu_0^T \left( V^{c,\pi}_{\widehat{\mathcal{M}} \cup \widehat{c}_0} - V^{c,*}_{\widehat{\mathcal{M}} \cup \widehat{c}_0} \right) \leq \frac{1}{1-\gamma} \leq 4\varepsilon_0 \leq 4\varepsilon_0 + \epsilon$. Thus, $\Pi^c_0$ contains all the policies, i.e., $\Pi^c_0 = \Delta^{\mathcal{A}}_{\mathcal{S}}$, and in particular $\pi^*, \widehat{\pi}^*_1 \in \Pi^c_0$. Suppose that for every $1 \leq k' < k$ the statement holds, we aim to prove that the statement also holds for $k$. Let $k' = k - 1$, from the inductive hypothesis we have that $\pi^*, \widehat{\pi}^*_k \in \Pi^c_{k-1}$. Then, from Lemma C.11, it holds that $\pi^* \in \Pi^c_k$. If $\widehat{\pi}^*_{k+1} \in \Pi^c_k$, the proof is finished. If $\widehat{\pi}^*_{k+1} \notin \Pi^c_k$, we prove by contradiction. Let $1 \leq j \leq k$ be the iteration such that $\widehat{\pi}^*_{k+1} \in \Pi^c_{j-1}$ and $\widehat{\pi}^*_{k+1} \notin \Pi^c_j$. Note that this assumption always holds, since $\Pi^c_0$ contains all policies. Recalling the inductive hypothesis, we have that $\widehat{\pi}^*_j \in \Pi^c_{j-1}$. Thus, from Lemma C.13, it must be that $\widehat{\pi}^*_{k+1}$ is suboptimal for all $j' \geqslant j$, in particular for $j' = k + 1$, which brings about a contradiction. $\qquad\square$

**Lemma C.15.** *It holds that $\pi^* \in \Pi^r_k$, where:*

$$\Pi^r_k = \left\{ \pi \in \Delta^{\mathcal{A}}_{\mathcal{S}} : \inf_{\mu_0 \in \Delta^{\mathcal{S}}} \mu_0^T \left( V^{r,\pi}_{\widehat{\mathcal{M}}} - V^{r,\widehat{\pi}^*}_{\widehat{\mathcal{M}}} \right) \geq \mathfrak{R}_k \right\} where$$

$$\mathfrak{R}_k = \frac{2\gamma R_{\max}}{(1-\gamma)^2} \| P_{\mathcal{T}} - \widehat{P_{\mathcal{T}}} \|_\infty + \frac{\gamma R_{\max}}{(1-\gamma)^2} \| (\pi^* - \widehat{\pi}^*) \|_\infty$$

*Proof.* We should show if $\pi \in \Pi^r_k$, we will have $V^{r,\pi}_{\mathcal{M}} \geq V^{r,\pi^*}_{\mathcal{M}}$.

$$V^{r,\pi}_{\widehat{\mathcal{M}}} - V^{r,\widehat{\pi}^*}_{\widehat{\mathcal{M}}} = V^{r,\pi}_{\widehat{\mathcal{M}}} - V^{r,\pi}_{\mathcal{M}} + V^{r,\pi}_{\mathcal{M}} - V^{r,\pi^*}_{\mathcal{M}} + V^{r,\pi^*}_{\mathcal{M}} - V^{r,\widehat{\pi}^*}_{\mathcal{M}} + V^{r,\widehat{\pi}^*}_{\mathcal{M}} - V^{r,\widehat{\pi}^*}_{\widehat{\mathcal{M}}}$$

$$\overset{(i,ii,iii)}{\leq} \frac{2\gamma R_{\max}}{(1-\gamma)^2} \| P_{\mathcal{T}} - \widehat{P_{\mathcal{T}}} \|_\infty + \frac{\gamma R_{\max}}{(1-\gamma)^2} \| (\pi^* - \widehat{\pi}^*) \|_\infty + V^{r,\pi}_{\mathcal{M}} - V^{r,\pi^*}_{\mathcal{M}}$$

$$= \mathfrak{R}_k + V^{r,\pi}_{\mathcal{M}} - V^{r,\pi^*}_{\mathcal{M}}$$

Since $\inf_{\mu_0 \in \Delta^{\mathcal{S}}} \mu_0^T \left( V^{r,\pi}_{\widehat{\mathcal{M}}} - V^{r,\widehat{\pi}^*}_{\widehat{\mathcal{M}}} \right) \geq \mathfrak{R}_k$, it must hold that $\inf_{\mu_0 \in \Delta^{\mathcal{S}}} \mu_0^T \left( V^{r,\pi}_{\mathcal{M}} - V^{r,\pi^*}_{\mathcal{M}} \right) \geq 0$

- To show (i), we first follow the simulation Lemma for the state-value function:

$$V^{r,\pi}_{\widehat{\mathcal{M}}} - V^{r,\pi}_{\mathcal{M}} = \gamma (I_{\mathcal{S}} - \gamma \pi \widehat{P_{\mathcal{T}}})^{-1} \pi (\widehat{P_{\mathcal{T}}} - P_{\mathcal{T}}) V^{r,\pi}_{\mathcal{M}}$$

Then we derive an upper bound for the difference between these state-values as follows:

$$V^{r,\pi}_{\widehat{\mathcal{M}}} - V^{r,\pi}_{\mathcal{M}} \leq \frac{\gamma}{1-\gamma} \| \pi (\widehat{P_{\mathcal{T}}} - P_{\mathcal{T}}) V^{r,\pi}_{\mathcal{M}} \|_\infty$$

$$\leq \frac{\gamma R_{\max}}{(1-\gamma)^2} \| \pi (\widehat{P_{\mathcal{T}}} - P_{\mathcal{T}}) \|_\infty$$

$$\leq \frac{\gamma R_{\max}}{(1-\gamma)^2} \| \widehat{P_{\mathcal{T}}} - P_{\mathcal{T}} \|_\infty$$

- (ii) holds due to the policy mismatch Lemma C.4:

$$V^{r,\pi^*}_{\mathcal{M}} - V^{r,\widehat{\pi}^*}_{\mathcal{M}} = \gamma (I_{\mathcal{S}} - \gamma \widehat{\pi}^* P_{\mathcal{T}})^{-1} (\pi^* - \widehat{\pi}^*) P_{\mathcal{T}} V^{r,\pi^*}_{\mathcal{M}}$$

Then we derive an upper bound for the difference between these state-values as follows:

$$V^{r,\pi^*}_{\mathcal{M}} - V^{r,\widehat{\pi}^*}_{\mathcal{M}} \leq \frac{\gamma}{1-\gamma} \| (\pi^* - \widehat{\pi}^*) P_{\mathcal{T}} V^{r,\pi^*}_{\mathcal{M}} \|_\infty$$

$$\leq \frac{\gamma R_{\max}}{(1-\gamma)^2} \| (\pi^* - \widehat{\pi}^*) P_{\mathcal{T}} \|_\infty$$

$$\leq \frac{\gamma R_{\max}}{(1-\gamma)^2} \| (\pi^* - \widehat{\pi}^*) \|_\infty$$

- (iii) holds due to the derivation to (i):

$$V_{\mathcal{M}}^{r,\widehat{\pi}^*} - V_{\widehat{\mathcal{M}}}^{r,\widehat{\pi}^*} \leq \frac{\gamma R_{\max}}{(1-\gamma)^2} \|P_{\mathcal{T}} - \widehat{P_{\mathcal{T}}}\|_{\infty}$$

Since we can guarantee $V_{\mathcal{M}}^{\pi} \geq V_{\mathcal{M}}^{\pi^*}$, we know $\pi^* \in \{\pi | V_{\mathcal{M}}^{\pi} \geq V_{\mathcal{M}}^{\pi^*}\}$. Subsequently, according to our Lemma 4.4, to find the feasible cost set, the exploration policy should follow the $\pi$ that visits states with larger cumulative rewards. $\qquad\square$

**Lemma C.16.** *Under the good event $\mathcal{E}_k$, let $\tilde{c} \in \arg\min_{c \in \mathcal{C}_{\mathfrak{P}}} \max_{(s,a) \in \mathcal{S} \times \mathcal{A}} |c(s,a) - \widehat{c}_k(s,a)|$. If $\pi \in \Pi_k$ and $\pi^* \in \Pi_{k-1}$, then $\sup_{\mu_0 \in \Delta^{\mathcal{S}}} \mu_0^T \left( V_{\widehat{\mathcal{M}} \cup \tilde{c}}^{c,\pi} - V_{\widehat{\mathcal{M}} \cup \tilde{c}}^{c,*} \right) \leq 6\varepsilon_k + \epsilon.$*

*Proof.*

$$\sup_{\mu_0 \in \Delta^{\mathcal{S}}} \mu_0^T \left( V_{\widehat{\mathcal{M}} \cup \tilde{c}}^{c,\pi} - V_{\widehat{\mathcal{M}} \cup \tilde{c}}^{c,*} \right)$$

$$\leq \underbrace{\sup_{\mu_0 \in \Delta^{\mathcal{S}}} \mu_0^T \left( V_{\widehat{\mathcal{M}} \cup \tilde{c}}^{c,\pi} - V_{\widehat{\mathcal{M}} \cup \widehat{c}_k}^{c,\pi} \right)}_{(a)} + \underbrace{\sup_{\mu_0 \in \Delta^{\mathcal{S}}} \mu_0^T \left( V_{\widehat{\mathcal{M}} \cup \widehat{c}_k}^{c,\pi} - V_{\widehat{\mathcal{M}} \cup \widehat{c}_k}^{c,*} \right)}_{(b)} + \underbrace{\sup_{\mu_0 \in \Delta^{\mathcal{S}}} \mu_0^T \left( V_{\widehat{\mathcal{M}} \cup \widehat{c}_k}^{c,*} - V_{\widehat{\mathcal{M}} \cup \tilde{c}}^{c,*} \right)}_{(c)},$$

$$\leq (\varepsilon_k) + (4\varepsilon_k + \epsilon) + (\varepsilon_k)$$

$$= 6\varepsilon_k + \epsilon$$

where

- (a) holds due to $\sup_{\mu_0 \in \Delta^{\mathcal{S}}} \mu_0^T \left( V_{\widehat{\mathcal{M}} \cup \tilde{c}}^{c,\pi} - V_{\widehat{\mathcal{M}} \cup \widehat{c}_k}^{c,\pi} \right) \leq \varepsilon_k^{\pi} \leq \varepsilon_k.$

- (b) results from $\sup_{\mu_0 \in \Delta^{\mathcal{S}}} \mu_0^T \left( V_{\widehat{\mathcal{M}} \cup \widehat{c}_k}^{c,\pi} - V_{\widehat{\mathcal{M}} \cup \widehat{c}_k}^{c,*} \right) \leq 4\varepsilon_k + \epsilon$, since $\pi \in \Pi_k$.

- (c) follows Eq. (54), recalling the definition of $\tilde{c}$.

$\qquad\square$

## C.13. Proof of Theorem 5.6

*Proof.* Suppose we have derived a value of $\bar{N}_K(s,a)$ so that for all $(s,a) \in \mathcal{S} \times \mathcal{A}$ (the rationale is discussed later), it holds that:

$$\mathcal{C}_K(s,a) = \min\left\{ \sigma \sqrt{\frac{\ell_K(s,a)}{2N_K^+(s,a)}}, C_{\max} \right\} \leq \check{\sigma} \sqrt{\frac{2\bar{\ell}_K(s,a)}{\bar{N}_K^+(s,a)}} \leq \frac{-\min_{a' \in \mathcal{A}} A_{\widehat{\mathcal{M}} \cup \tilde{c}}^{c,*}(s,a')\varepsilon_K}{6\varepsilon_{K-1} + \epsilon}. \tag{55}$$

From Lemma C.8, we obtain

$$\begin{aligned}
\bar{N}_k^+(s,a) &= \frac{2\check{\sigma}^2(6\varepsilon_{K-1} + \epsilon)^2 \bar{\ell}_K(s,a)}{(\min_{a' \in \mathcal{A}} A_{\widehat{\mathcal{M}} \cup \tilde{c}}^{c,*}(s,a'))^2 \varepsilon_K^2} \\
&= -\frac{4\sigma^2(6\varepsilon_{K-1} + \epsilon)^2}{(\min_{a' \in \mathcal{A}} A_{\widehat{\mathcal{M}} \cup \tilde{c}}^{c,*}(s,a'))^2 \varepsilon_K^2} W_{-1}\left( \frac{(\min_{a' \in \mathcal{A}} A_{\widehat{\mathcal{M}} \cup \tilde{c}}^{c,*}(s,a'))^2 \varepsilon_K^2}{4\sigma^2(6\varepsilon_{K-1} + \epsilon)^2} \sqrt{\frac{\delta}{36SA}} \right) \\
&= \frac{8\sigma^2(6\varepsilon_{K-1} + \epsilon)^2}{(\min_{a' \in \mathcal{A}} A_{\widehat{\mathcal{M}} \cup \tilde{c}}^{c,*}(s,a'))^2 \varepsilon_K^2} \log\left( \frac{4\sigma^2(6\varepsilon_{K-1} + \epsilon)^2}{(\min_{a' \in \mathcal{A}} A_{\widehat{\mathcal{M}} \cup \tilde{c}}^{c,*}(s,a'))^2 \varepsilon_K^2} \sqrt{\frac{36SA}{\delta}} \right) \\
&= \widetilde{\mathcal{O}}\left( \frac{\sigma^2(6\varepsilon_{K-1} + \epsilon)^2}{(\min_{a' \in \mathcal{A}} A_{\widehat{\mathcal{M}} \cup \tilde{c}}^{c,*}(s,a'))^2 \varepsilon_K^2} \right).
\end{aligned} \tag{56}$$

Summing over $n = \sum_{(s,a) \in \mathcal{S} \times \mathcal{A}} \bar{N}_k^+(s,a)$, since the convergence of BEAR is stricter than PCSE, i.e., (13) is always satisfied if (12) is satisfied, the sample complexity of BEAR constitutes a lower bound for that of PCSE. Recall the sample complexity of BEAR exploration strategy in Theorem 5.5, we obtain

$$n \leq \widetilde{\mathcal{O}} \left( \min \left\{ \widetilde{\mathcal{O}} \left( \frac{\check{\sigma}^2 (2C_{\max} - \varepsilon_K(1-\gamma))^2}{(1-\gamma)^2 \varepsilon_K^2 C_{\max}^2} \right), \frac{\sigma^2 (6\varepsilon_{K-1} + \epsilon)^2 SA}{(\min_{(s,a)} A_{\widehat{\mathcal{M}} \cup \tilde{c}}^{c,*}(s,a))^2 \varepsilon_K^2} \right\} \right). \tag{57}$$

Next, we explain the rationale for the assumption in Eq. (55). We have for every $\pi \in \Pi_k$,

$$\begin{aligned}
\|e_k(s,a;\pi^*,\widehat{\pi}^*)\|_\infty &\overset{(a)}{\leq} \gamma \mu_0^T (I_{\mathcal{S}} - \gamma \pi \widehat{P_{\mathcal{T}}})^{-1} \pi \mathcal{C}_k \\
&\overset{(b)}{\leq} \frac{\gamma \varepsilon_K}{6\varepsilon_K + \epsilon} \mu_0^T (I_{\mathcal{S}} - \gamma \pi \widehat{P_{\mathcal{T}}})^{-1} \pi \left( -A_{\widehat{\mathcal{M}} \cup \tilde{c}}^{c,*} \right) \\
&\overset{(c)}{=} \frac{\gamma \varepsilon_K}{6\varepsilon_{K-1} + \epsilon} \mu_0^T \left( V_{\widehat{\mathcal{M}} \cup \tilde{c}}^{c,\pi} - V_{\widehat{\mathcal{M}} \cup \tilde{c}}^{c,*} \right) \overset{(e)}{\leq} \varepsilon_K,
\end{aligned} \tag{58}$$

- (a) follows the matrix from the Bellman equation for the value function.

- (b) is based on the assumption in Eq. (55).

- (c) follows (Metelli et al., 2021, Lemma B.3), where we treat $r = -\tilde{c}$ and note that $V_{\widehat{\mathcal{M}} \cup (-\tilde{c})}^\pi = -V_{\widehat{\mathcal{M}} \cup \tilde{c}}^\pi$, $Q_{\widehat{\mathcal{M}} \cup (-\tilde{c})}^\pi = -Q_{\widehat{\mathcal{M}} \cup \tilde{c}}^\pi$ and $A_{\widehat{\mathcal{M}} \cup (-\tilde{c})}^\pi = -A_{\widehat{\mathcal{M}} \cup \tilde{c}}^\pi$.

- (d) results from Lemma C.16 and $\gamma < 1$.

$\square$

### C.14. Optimization Problem and the Two-Timescale Stochastic Approximation

We can now formulate the optimization problem.

$$\begin{aligned}
\varepsilon_{k+1} = \sup_{\substack{\mu_0 \in \Delta^{\mathcal{S}} \\ \pi \in \Pi_k}} & \mu_0^T (I_{\mathcal{S} \times \mathcal{A}} - \gamma P_{\mathcal{T}} \pi) \mathcal{C}_{k+1} \\
\text{s.t.} \quad \Pi_k &= \Pi_k^c \cap \Pi_k^r \\
\Pi_k^c &= \left\{ \pi \in \Delta_{\mathcal{S}}^{\mathcal{A}} : \sup_{\mu_0 \in \Delta^{\mathcal{S}}} \mu_0^T (V_{\widehat{\mathcal{M}} \cup \widehat{c}_k}^{c,\pi} - V_{\widehat{\mathcal{M}} \cup \widehat{c}_k}^{c,*}) \leq 4\varepsilon_k + 2\epsilon \right\} \\
\Pi_k^r &= \left\{ \pi \in \Delta_{\mathcal{S}}^{\mathcal{A}} : \inf_{\mu_0 \in \Delta^{\mathcal{S}}} \mu_0^T \left( V_{\widehat{\mathcal{M}}}^{r,\pi} - V_{\widehat{\mathcal{M}}}^{r,\widehat{\pi}^*} \right) \geq \mathfrak{R}_k \right\}
\end{aligned} \tag{59}$$

where $\mathfrak{R}_k = \frac{2\gamma R_{\max}}{(1-\gamma)^2} \|P_{\mathcal{T}} - \widehat{P_{\mathcal{T}}}\|_\infty + \frac{\gamma R_{\max}}{(1-\gamma)^2} \|(\pi^* - \widehat{\pi}^*)\|_\infty$.

Recall that the discounted normalized occupancy measure is defined by

$$\rho_{\mathcal{M}}^\pi(s,a) = (1-\gamma) \sum_{t=0}^\infty \gamma^t \mathbb{P}_{\mu_0}^\pi(s_t = s, a_t = a), \tag{60}$$

where the normalizer $(1-\gamma)$ makes $\rho_{\mathcal{M}}^\pi(s,a)$ a probability measure, i.e., $\sum_{(s,a)} \rho_{\mathcal{M}}^\pi(s,a) = 1$.

The promised relationship between reward value function and occupancy measure is as follows:

$$\begin{aligned}
(1-\gamma) V_{\mathcal{M}}^{r,\pi} &\overset{(a)}{=} (1-\gamma) \mathbb{E}_{\mu_0, \pi, P_{\mathcal{T}}} \left[ \sum_{t=0}^\infty \gamma^t r(s_t, a_t) \right] \\
&= (1-\gamma) \sum_{t=0}^\infty \gamma^t \sum_{(s,a)} \mathbb{P}_{\mu_0}^\pi(s_t = s, a_t = a) r(s_t = s, a_t = a)
\end{aligned}$$

$$\overset{(b)}{=} \sum_{(s,a)} \left[ (1-\gamma) \sum_{t=0}^{\infty} \gamma^t \mathbb{P}_{\mu_0}^{\pi}(s_t = s, a_t = a) \right] \cdot \left[ r(s_t = s, a_t = a) \right]$$

$$= \langle \rho_{\mathcal{M}}^{\pi}, r \rangle, \tag{61}$$

where step (a) follows the definition of the reward state-value function, and step (b) exchanges the order of two summations.

Similarly, concerning the cost function, the relationship between the cost value function and (the same) occupancy measure is as follows:

$$(1-\gamma)V_{\mathcal{M}}^{c,\pi} = (1-\gamma)\mathbb{E}_{\pi,P_{\mathcal{T}}} \left[ \sum_{t=0}^{\infty} \gamma^t c(s_t, a_t) \right]$$

$$= (1-\gamma) \sum_{t=0}^{\infty} \gamma^t \sum_{(s,a)} \mathbb{P}_{\mu_0}^{\pi}(s_t = s, a_t = a) c(s_t = s, a_t = a)$$

$$= \sum_{(s,a)} \left[ (1-\gamma) \sum_{t=0}^{\infty} \gamma^t \mathbb{P}_{\mu_0}^{\pi}(s_t = s, a_t = a) \right] \cdot \left[ c(s_t = s, a_t = a) \right]$$

$$= \langle \rho_{\mathcal{M}}^{\pi}, c \rangle. \tag{62}$$

For simplicity, denote the occupancy measure vector $\rho_{\mathcal{M}}^{\pi}$ as vector $x$. As a result, the optimization problem (59) can be recast as a linear program.

$$\begin{aligned} \min_{x} \quad & -\langle x, \mathcal{C}_{k+1} \rangle \\ \text{s.t.} \quad & -(1-\gamma)(V_{\widehat{\mathcal{M}} \cup \widehat{c}_k}^{c,*} + 4\varepsilon_k + 2\epsilon) + \langle x, \widehat{c}_k \rangle \leq 0 \\ & (1-\gamma)(V_{\widehat{\mathcal{M}}}^{r,\widehat{\pi}^*} + \mathfrak{R}_k) - \langle x, r \rangle \leq 0 \end{aligned} \tag{63}$$

To solve this linear program, we introduce the Lagrangian function and calculate its saddle points by solving the dual problem. The Lagrangian of this primal problem is defined as:

$$L(x,\lambda) = -\langle x, \mathcal{C}_{k+1} \rangle + \lambda_1 \left( -(1-\gamma)(V_{\widehat{\mathcal{M}} \cup \widehat{c}_k}^{c,*} + 4\varepsilon_k + 2\epsilon) + \langle x, \widehat{c}_k \rangle \right)$$
$$+ \lambda_2 \left( (1-\gamma)(V_{\widehat{\mathcal{M}}}^{r,\widehat{\pi}^*} + \mathfrak{R}_k) - \langle x, r \rangle \right), \tag{64}$$

where $\lambda = [\lambda_1, \lambda_2]^T$ is a nonnegative real vector, composed of so-called Lagrangian multipliers. The dual problem is defined as:

$$\min_{x} \max_{\lambda \geq 0} L(x, \lambda). \tag{65}$$

To solve this dual problem, we follow a gradient-based approach known as the two-timescale stochastic approximation (Szepesvári, 2021). At time step $k$, the following updates are conducted,

$$x_{k+1} - x_k = -a_k(L'_x(x_k, \lambda_k) + W_k), \tag{66}$$
$$\lambda_{k+1} - \lambda_k = b_k(L'_\lambda(x_k, \lambda_k) + U_k), \tag{67}$$

where the two coefficients $a_k \ll b_k$, satisfying $\sum_k a_k = \sum b_k = \infty$, $\sum a_k^2 < \infty$ and $\sum b_k^2 < \infty$. Under this condition, the convergence is guaranteed in the limit. As an option, we can set $a_k = c/k$, $b_k = c/k^{0.5+\kappa}$, with $c$ being a constant and $0 < \kappa < 0.5$. $W_k$ and $U_k$ are two zero-mean noise sequences. The two gradients are:

$$L'_x(x_k, \lambda_k) = -\mathcal{C}_{k+1} + \lambda_1 \widehat{c}_k - \lambda_2 r, \tag{68}$$

$$L'_\lambda(x_k, \lambda_k) = \begin{bmatrix} L'_{\lambda_1}(x_k, \lambda_k) \\ L'_{\lambda_2}(x_k, \lambda_k) \end{bmatrix} = \begin{bmatrix} -(1-\gamma)(V_{\widehat{\mathcal{M}} \cup \widehat{c}_k}^{c,*} + 4\varepsilon_k + 2\epsilon) + \langle x, \widehat{c}_k \rangle \\ (1-\gamma)(V_{\widehat{\mathcal{M}}}^{r,\widehat{\pi}^*} + \mathfrak{R}_k) - \langle x, r \rangle \end{bmatrix}. \tag{69}$$

At each time step $k$, the exploration policy can be calculated as,

$$\pi_k(a|s) = \frac{x_k(s,a)}{\sum_a x_k(s,a)}. \tag{70}$$

# D. Experimental Details

We ran experiments on a desktop computer with Intel(R) Core(TM) i5-14400F and NVIDIA GeForce RTX 2080 Ti.

## D.1. Discrete Environment

**More details about Gridworld.** In this paper, we create a map with dimensions of $7 \times 7$ units and define four distinct settings, as illustrated in Figure 2. We use two coordinates to represent the location, where the first coordinate corresponds to the vertical axis, and the second coordinate corresponds to the horizontal axis. The agent aims to navigate from a starting location to a target location while avoiding the given constraints. The agent starts in the lower left cell $(0, 0)$, and it has 8 actions that correspond to 8 adjacent directions, including four cardinal directions (up, down, left, right) as well as the four diagonal directions (upper-left, lower-left, upper-right, lower-right). The reward and target locations are the same, which is located in the upper right cell $(6, 6)$ for the first, second, and fourth Gridworld environment or located in the upper left cell $(6, 0)$ for the third Gridworld environment. If the agent takes an action, then with probability $0.05$ this action fails, and the agent moves in any viable random direction (including the direction this action leads to) with uniform probabilities. The reward in the reward state cell is $1$, while all other cells have a $0$ reward. The cost in a constraint location is also $1$. The game continues until a maximum time step of $50$ is reached.

**Comparison Methods.** The upper confidence bound exploration strategy is derived from the UCB algorithm, which selects an action with the highest upper bound. The maximum-entropy strategy selects an action on a state with the maximum entropy given previous choices of actions. The random strategy uniformly randomly selects a viable action on a state $s$. The $\epsilon$-greedy strategy selects an action based on the $\epsilon$-greedy algorithm, balancing exploration and exploitation with the exploration parameter $\epsilon = 1/\sqrt{k}$.

**More details about Figure 3.** In Figure 3, we plot the mean and $68\%$ confidence interval (1-sigma error bar) computed with 5 random seeds ($123456, 123, 1234, 36, 34$) and exploration episodes $n_e = 1$. The six exploration strategies compared in Figure 3 include: upper confidence bound, maximum-entropy, random, BEAR, $\epsilon$-greedy, and PCSE. Meanwhile, we utilize the running score to make the training process more resilient to environmental stochasticity: $running\_score = 0.2 * running\_score + 0.8 * iteration\_rewards$ (or $iteration\_costs$) (Luo et al., 2022).

## D.2. Weighted Generalized Intersection over Union (WGIoU)

In this section, we present the methodology for designing the metric that assesses the similarity between the estimated and ground-truth cost functions, which we refer to as WGIoU. We commence our discussion by explaining IoU, followed by GIoU, and ultimately introduce the novel concept of WGIoU for ICRL.

Intersection Over Union (IoU) score is a commonly used metric in the field of object detection, which measures how similar two sets are. The IoU score is bounded in $[0, 1]$ ($0$ being no overlap between two sets and $1$ being complete overlap). Suppose there are two sets $X$ and $Y$,

$$\text{IoU} = \frac{|X \cap Y|}{|X \cup Y|}.$$

Note that IoU equals to zero for all two sets with no overlap, which is a rough metric and incurs the problem of vanishing gradients. To further measure the difference between two sets with no overlap, Signed IoU (SIoU) (Simonelli et al., 2019) and Generalized IoU (GIoU) (Rezatofighi et al., 2019) are proposed. Both SIoU and GIoU are bounded in $[-1, 1]$. However, SIoU is constrained to a rectangular bounding box, which is not the case for the cost function. By contrast, GIoU is not limited to rectangular boxes. Thus, GIoU is more suitable for comparing the distance between the estimated cost function and the ground-truth cost function.

$$\text{GIoU} = \text{IoU} - \frac{|Z \backslash (X \cup Y)|}{|Z|},$$

where set $Z$ is the minimal enclosing convex set that contains both $X$ and $Y$. Taking cost function into account, the difference between $\widehat{c}_k$ the estimated cost function at iteration $k$ and $c$ the ground-truth cost function is calculated as,

$$\text{GIoU} = \frac{|c \cap \widehat{c}_k|}{|c \cup \widehat{c}_k|} - \frac{|(c \oplus \widehat{c}_k) \backslash (c \cup \widehat{c}_k)|}{|c \oplus \widehat{c}_k|},$$

where $\widehat{c}_k \oplus c$ denotes the enclosing convex matrix of $c$ and $\widehat{c}_k$.

Note that the estimated cost function $\widehat{c}_k$ could have different values, but GIoU only reflects spatial relationship and is unable to represent weight features. To accommodate our settings, weighted GIoU (WGIoU) is proposed, where we measure the distance between a weighted estimated cost function and a uniformly valued (or weighted) ground-truth cost function. WGIoU is also bounded in $[-1, 1]$. To calculate WGIoU, first, remap the cost function to $(\{0\} \cup [1, +\infty))^{\mathcal{S} \times \mathcal{A}}$,

$$\widehat{c}_k^{\circledast}(s, a) = \frac{\widehat{c}_k(s, a)}{\min\left\{\min^+_{(s,a)\in\mathcal{S}\times\mathcal{A}} \widehat{c}_k(s, a), \min^+_{(s,a)\in\mathcal{S}\times\mathcal{A}} c(s, a)\right\}}, \tag{71}$$

$$c^{\circledast}(s, a) = \frac{c(s, a)}{\min\left\{\min^+_{(s,a)\in\mathcal{S}\times\mathcal{A}} \widehat{c}_k(s, a), \min^+_{(s,a)\in\mathcal{S}\times\mathcal{A}} c(s, a)\right\}}. \tag{72}$$

where $\min^+_{(s,a)\in\mathcal{S}\times\mathcal{A}}$ returns the minimum positive value of $\widehat{c}_k$ or $c$ over all $(s, a)$ pairs. Note that $c$ must exceed $0$ at certain $(s, a)$. Otherwise, the cost function is zero, indicating an absence of constraint anywhere. Also note that if $\widehat{c}_k$ is zero, let $\widehat{c}_k^{\circledast}(s, a) = 0$ and $c^{\circledast}(s, a) = c(s, a)/\min^+_{(s,a)\in\mathcal{S}\times\mathcal{A}} c(s, a)$. Besides the two trivial situations, the above two equations (71 and 72) can be applied naturally.

Then, WGIoU is defined as:

$$\text{WGIoU} = \frac{\langle \widehat{c}_k^{\circledast}, c^{\circledast} \rangle}{\langle \mathbf{1}, \max\{\widehat{c}_k^{\circledast}, c^{\circledast}, \langle \widehat{c}_k^{\circledast}, c^{\circledast}\rangle\}\rangle} + \left(e^{-\langle \mathbf{1}, \max\{\widehat{c}_k^{\circledast}, c^{\circledast}\}\rangle} - 1\right)\mathbb{1}\left\{\langle \widehat{c}_k^{\circledast}, c^{\circledast}\rangle = 0\right\},$$

where $\mathbf{1}$ denotes the vector with appropriate length whose elements are all 1s. The rationale here can be understood by distinguishing two cases. For the first case, there is an overlap between $\widehat{c}_k$ and $c$, so the second term in WGIoU is 0. For the first term, for some $(s, a)$, 1) if both $\widehat{c}_k^{\circledast}(s, a) \geq 1$ and $c^{\circledast}(s, a) \geq 1$, WGIoU approaches 1; 2) if either $\widehat{c}_k^{\circledast}(s, a) = 0$ or $c^{\circledast}(s, a) = 0$, WGIoU approaches 0. For the second case, so there is no overlap between $\widehat{c}_k$ and $(s, a)$, the first term in WGIoU is 0. The second term is always below 0 and approaches $-1$ if the estimated and ground-truth cost functions contain large values.

### D.3. Continuous Environment

**Density model.** Recall that in Definition 5.3, the concept of pseudo-counts is introduced to analyze the uncertainty of the transition dynamics without a generative model. Here, we abuse the concept of pseudo-counts for generalizing count-based exploration algorithms to the non-tabular settings (Bellemare et al., 2016). Let $\rho$ be a density model on a finite space $\mathcal{X}$, and $\rho_n(x)$ the probability assigned by the model to $x$ after being trained on a sequence of states $x_1, \ldots, x_n$. Assume $\rho_n(x) > 0$ for all $x, n$. The recoding probability $\rho'_n(x)$ is then the probability the model would assign to $x$ if it was trained on that same $x$ one more time. We call $\rho$ *learning-positive* if $\rho'_n(x) \geq \rho_n(x)$ for all $x_1, \ldots, x_n, x \in \mathcal{X}$. A learning-positive $\rho$ implies $\text{PG}_n(x) \geq 0$ for all $x \in \mathcal{X}$. For learning-positive $\rho$, we define the *pseudo-count* as $\hat{N}_n(x) = \rho_n(x) \cdot n$, where $n$ is the total count. The pseudo-count generalizes the usual state visitation count function $N_n(x)$, also called the empirical count function or simply empirical count, which equals the number of occurrences of a state in the sequence.

**Methods.** We first train a Deep Q Network (DQN) in advance that stores the Q values of the constrained Point Maze environment. This DQN induces the expert policy at any given state. We also train a density model that accounts for calculating the pseudo-count of any given state-action pairs. The agent then collects samples from an unconstrained Point Maze environment where it could violate constraints. For algorithm BEAR, Proximal Policy Optimization (PPO) is utilized to obtain the exploration policy $\pi_k$. For algorithm PCSE, we rank 8 permissible actions for the exploration policy. The action that has a high estimated cost or a high reward is assigned with more probability to choose from. After the rollout of this exploration policy, the density model and accuracy are updated for the selection of the next exploration policy. Multiple rounds of iterations are conducted until the target accuracy is achieved.

**Point Maze.** In this environment, we create a map of $5\text{m} \times 5\text{m}$, where the area of each cell is $1\text{m} \times 1\text{m}$. The center of the map is the original point, i.e. $(0, 0)$. The constraint is initially set at the cell centered at $(-1, 0)$. The agent is a 2-DoF ball, force-actuated in the cartesian directions x and y. The reward obtained by the agent depends on where the agent reaches a target goal in a closed maze. The ball is considered to have reached the goal if the Euclidean distance between the ball and the goal is smaller than $0.5\text{m}$. The reward in the reward state cell is 1, while all other cells have a 0 reward. The cost in a constraint location is also 1. The game terminates when a maximum time step of 500 is reached. The state space dimension

is continuous and consists of 4 dimensions (two as the x and y coordinates of the agent and two as the linear velocity in the x and y direction). The action space is discrete, and at each state, there are 8 permissible actions (8 directions to add a linear force), similar to the action space of the Gridworld environment. The environment has a certain degree of stochasticity because there is a sampled noise from a uniform distribution to the cell's $(x, y)$ coordinates.

# E. More Experimental Results

## E.1. Gridworld Environments

Figure 7, 8, 9 and 10 show the constraint learning process of six exploration strategies in four Gridworld environments, i.e., Gridwworld-1, 2, 3, and 4. Note that in Figure 8 (Gridworld-2) and Figure 10 (Gridworld-4), only a fraction of ground-truth constraint is learned. This is attributed to ICRL's emphasis on identifying the minimum set of constraints necessary to explain expert behavior. Venturing into unidentified part of ground-truth constraints will not yield any advantages for cumulative rewards.

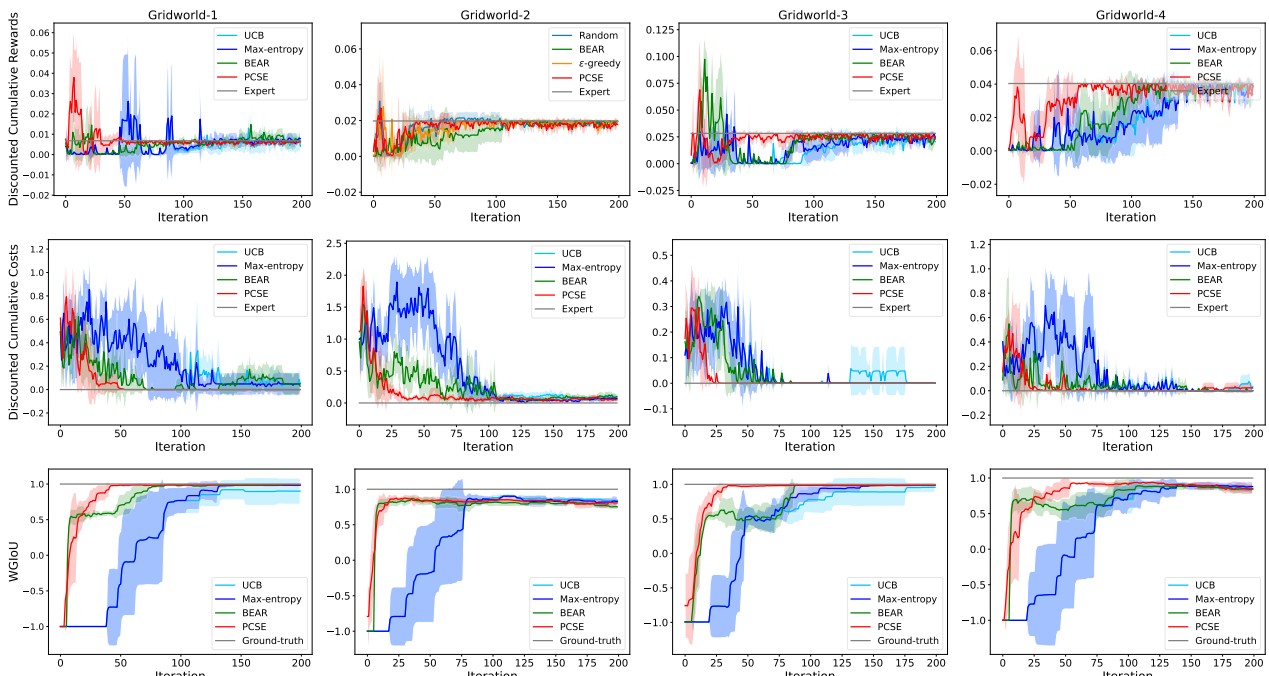

*Figure 5.* Training curves of discounted cumulative rewards (top), costs (middle), and WGIoU (bottom) for two other exploration strategies in four Gridworld environments.

## E.2. Point Maze Environment

Figure 6 shows the constraint learning process of PCSE in the Point Maze environment.

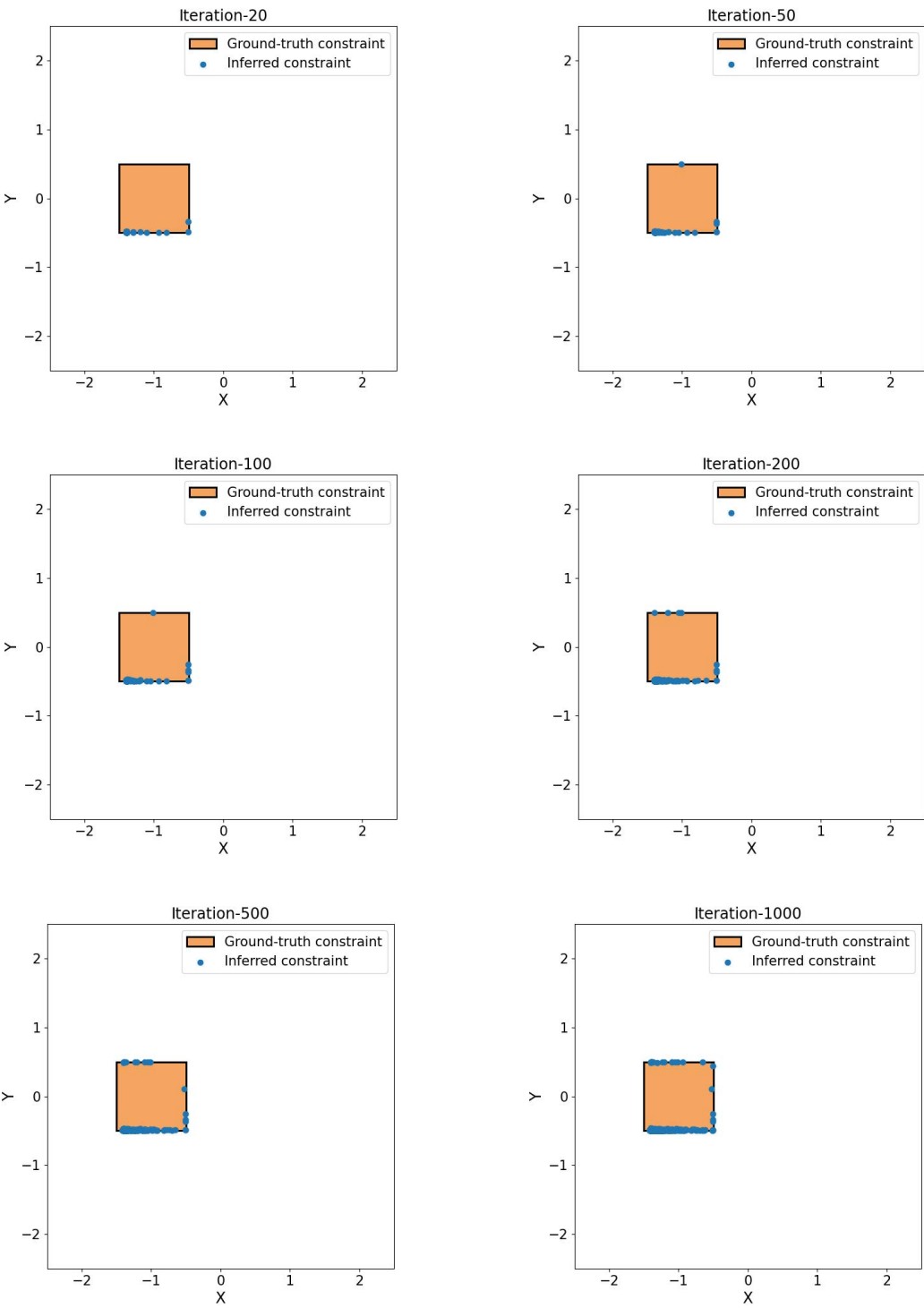

*Figure 6.* Constraint learning performance of PCSE for ICRL in the Point Maze environment.

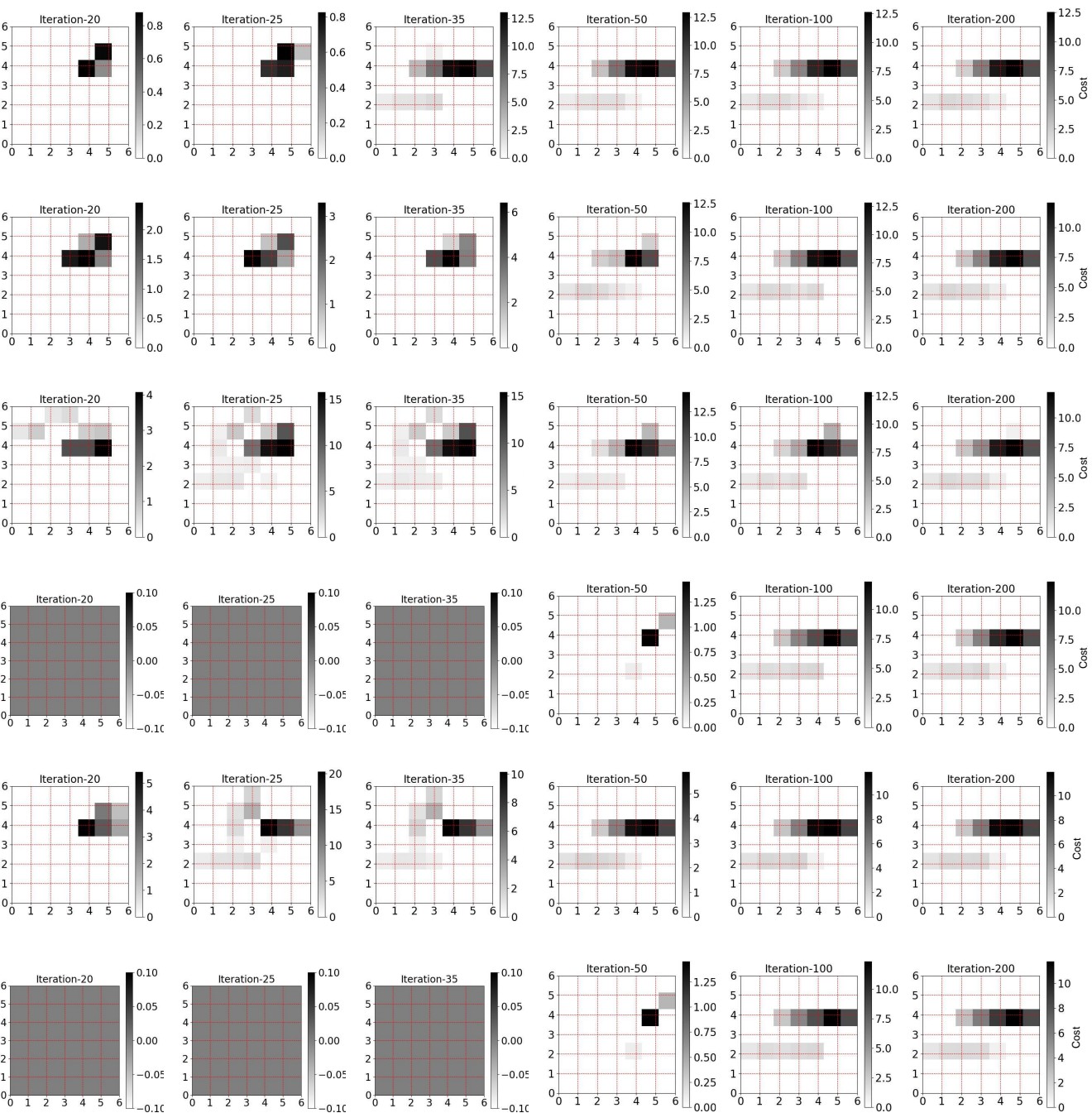

*Figure 7.* Constraint learning performance of six exploration strategies for ICRL in Gridworld-1. PCSE (1st row), BEAR strategy (2nd row), $\epsilon$-greedy exploration strategy (3rd row), maximum-entropy exploration strategy (4th row), random exploration strategy (5th row), upper confidence bound exploration strategy (bottom row).

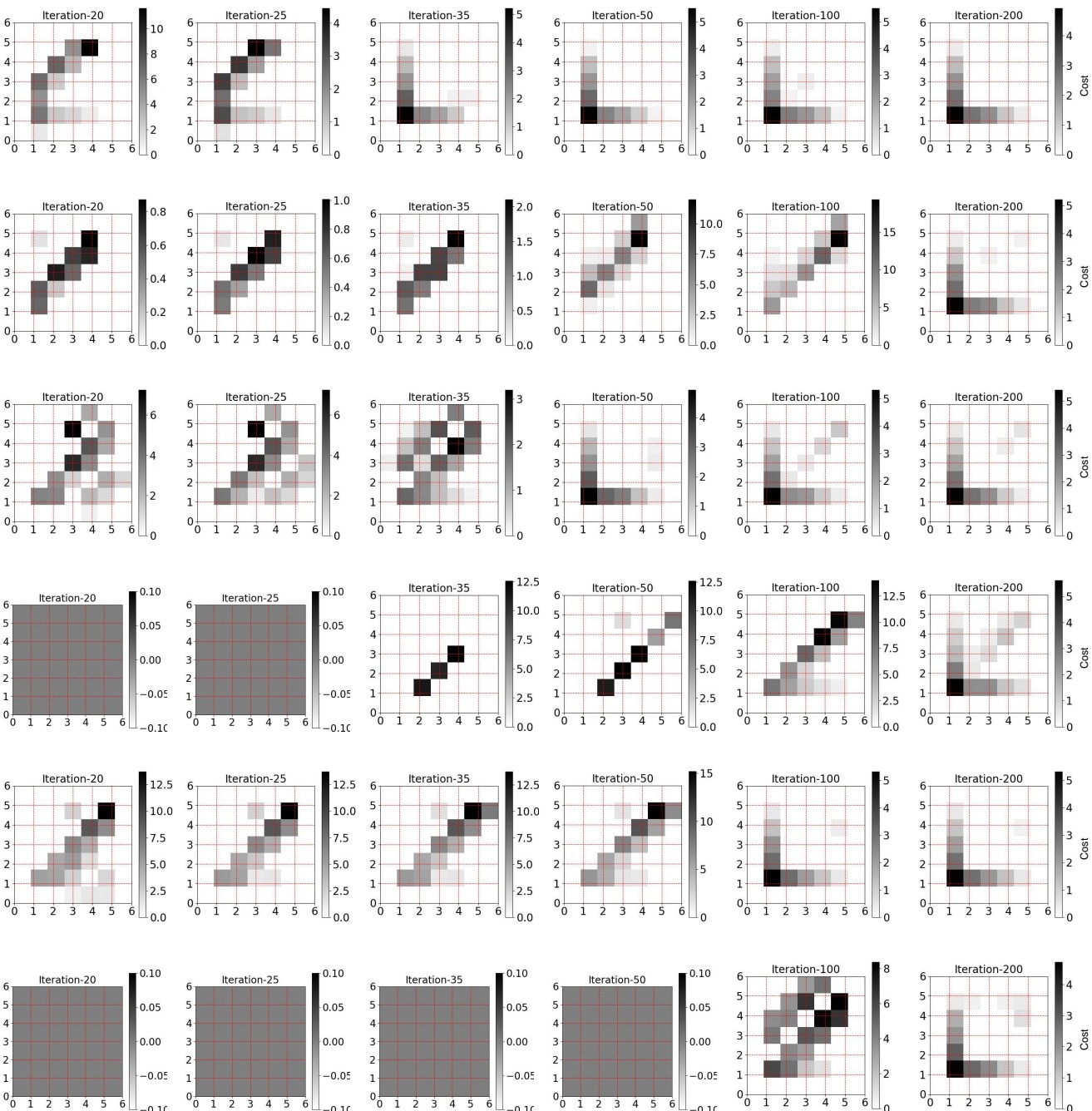

*Figure 8.* Constraint learning performance of six exploration strategies for ICRL in Gridworld-2. PCSE (1st row), BEAR strategy (2nd row), $\epsilon$-greedy exploration strategy (3rd row), maximum-entropy exploration strategy (4th row), random exploration strategy (5th row), upper confidence bound exploration strategy (bottom row).

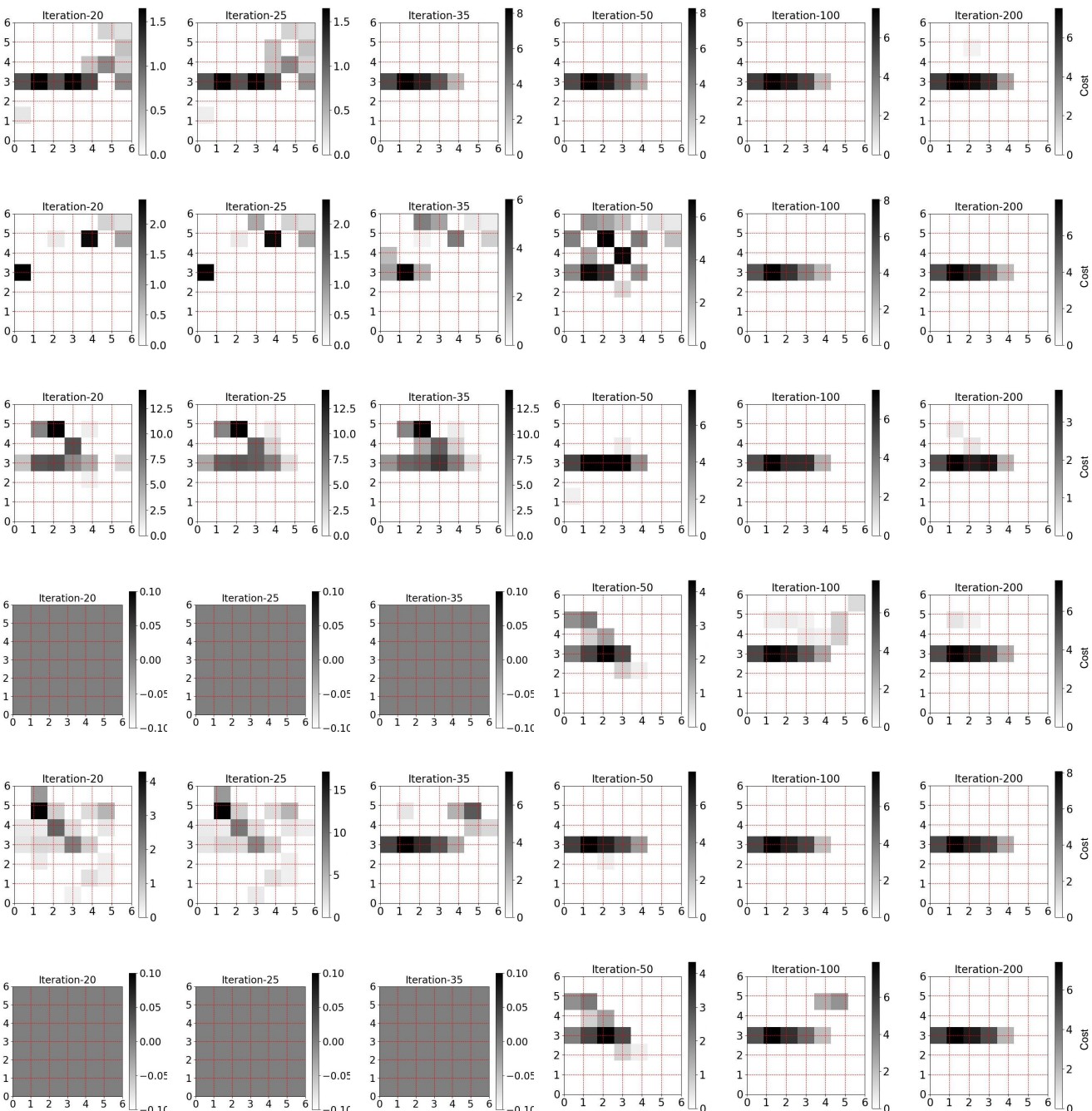

*Figure 9.* Constraint learning performance of six exploration strategies for ICRL in Gridworld-3. PCSE (1st row), BEAR strategy (2nd row), $\epsilon$-greedy exploration strategy (3rd row), maximum-entropy exploration strategy (4th row), random exploration strategy (5th row), upper confidence bound exploration strategy (bottom row).

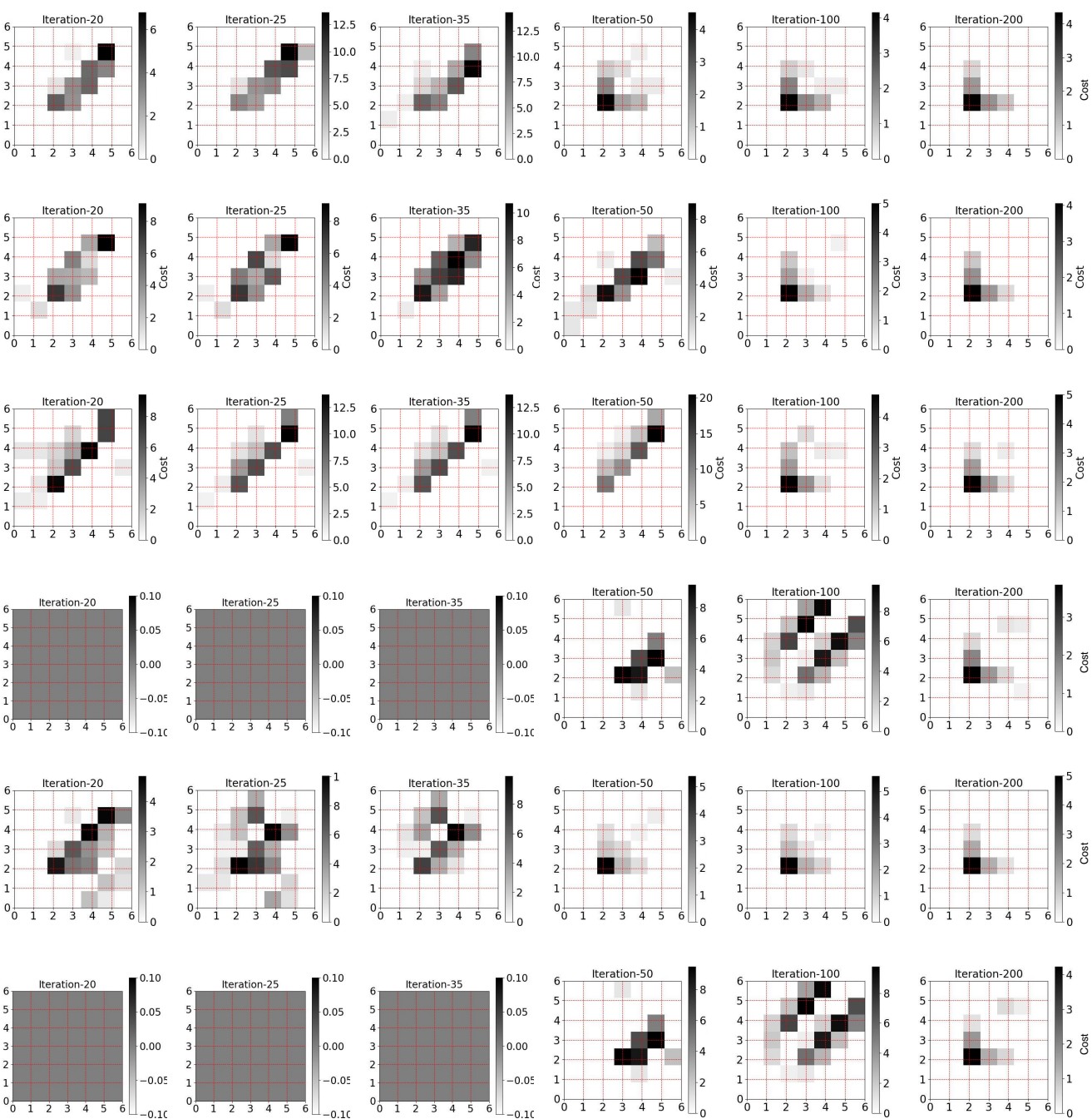

*Figure 10.* Constraint learning performance of six exploration strategies for ICRL in Gridworld-4. PCSE (1st row), BEAR strategy (2nd row), $\epsilon$-greedy exploration strategy (3rd row), maximum-entropy exploration strategy (4th row), random exploration strategy (5th row), upper confidence bound exploration strategy (bottom row).

