# OpenReview forum: "Provably Efficient Exploration in Inverse Constrained Reinforcement Learning"
_ICML.cc/2025/Conference — ICML 2025 poster_

### Official Review · Reviewer_p7jo · 2025-03-06

**Overall Recommendation:** 4

**Summary:**

This paper tackles ICRL, adding safety constraints in addition to classical IRL.
It claims to stand out by handling unknown environments, unlike many recent ICRL studies that assume known conditions.
The paper focuses on balancing expert imitation with exploration in Inverse Constrained Reinforcement Learning (ICRL), introducing efficient strategies for learning constraints. In particular, the paper's main contribution is a strategic exploration framework with two algorithms: Bounded Error Aggregate Reduction (BEAR) and Policy-Constrained Strategic Exploration (PCSE), both backed by theoretical guarantees.

Both BEAR and PCSE are supported by rigorous theoretical analyses, providing tractable upper bounds on sample complexity. The paper uses tools like the Hoeffding inequality to derive these bounds, ensuring that the algorithms achieve Probably Approximately Correct (PAC) optimality.

The sample complexity analysis is a significant contribution, as it quantifies the number of samples needed for accurate constraint inference, addressing a gap in previous ICRL literature where such guarantees were often absent or limited to specific settings.

**Claims And Evidence:**

The primary claim is that the proposed BEAR and PCSE algorithms achieve efficient constraint inference in ICRL with unknown dynamics.

This is theoretically backed by rigorous sample complexity analyses (e.g., Theorems 5.5 and 5.6) using tools like Hoeffding’s inequality, demonstrating PAC optimality with high probability.

Empirical results in Gridworld and Point Maze environments further corroborate this, showing PCSE outperforming baselines like maximum-entropy and ε-greedy in terms of rewards, costs, and constraint similarity. One issue is that those baseline algorithms are general-purpose algorithms. Another issue is that the environments are too simple. However, the theoretical claim mitigates this crude empirical result issue.

**Essential References Not Discussed:**

N/A

**Experimental Designs Or Analyses:**

* Point Maze has a 4D continuous state space (x, y coordinates plus velocities), which is a step up from Gridworld’s discrete 2D grid. However, the environment is still relatively simple—a flat 5m x 5m square with a single constraint at (-1,0) and a goal within 0.5m. This lacks the intricate obstacles, walls, or multi-goal complexity of typical maze benchmarks (e.g., OpenAI Gym’s Maze environments), limiting its ability to test navigation in truly continuous, cluttered spaces.

**Methods And Evaluation Criteria:**

The two algorithms—Bounded Error Aggregate Reduction (BEAR) and Policy-Constrained Strategic Exploration (PCSE)—are thoughtfully designed to address the challenge of efficient exploration without relying on generative models, a common limitation in prior ICRL work. BEAR minimizes cost estimation errors across all state-action pairs, while PCSE focuses exploration on plausibly optimal policies, leveraging a constrained optimization approach (Section 5.2).

The use of a Probably Approximately Correct (PAC) optimality criterion (Definition 4.9) to evaluate the closeness of inferred constraints to the true feasible set is a rigorous and appropriate metric.

Evaluation is conducted using benchmark datasets—Gridworld for discrete settings and Point Maze for continuous environments—which are standard in RL research and reasonable choices for testing ICRL. Gridworld (7x7 grid, Section D.1) offers a controlled, interpretable environment to assess exploration strategies, while Point Maze (5m x 5m, Section D.3) introduces continuous state spaces and stochasticity, reflecting real-world complexity.

Testing on additional continuous or larger-scale benchmarks could strengthen claims of broader applicability.

**Other Comments Or Suggestions:**

I don't think the Linear MDPs for future research is a particularly an interesting idea, as Linear MDPs are almost never practical.

**Other Strengths And Weaknesses:**

The work is significant for advancing ICRL in safety-critical domains where constraints are paramount but ill-defined. The theoretical sample complexity bounds (Theorems 5.5 and 5.6) fill a gap in ICRL literature. While not an application-driven ML paper per se, its focus on unknown dynamics and minimal constraint sets aligns with real-world use cases where expert data is available but environmental models are not.

Despite overall clarity, the omission of the effect of RL solver’s sample complexity in Theorem 5.5 (Section 5.1) on the overall sample complexity is glossed over, potentially confusing readers expecting a full efficiency analysis.

**Questions For Authors:**

How do you think about the effect of sample complexity of RL solver on the overall sample complexity?

**Relation To Broader Scientific Literature:**

PCSE’s restriction to a candidate policy set extends ideas from Bayesian IRL’s posterior sampling (Ramachandran and Amir, 2007), where a distribution over reward functions guides policy inference. However, PCSE focuses on constraints and leverages a structured policy set, akin to constrained policy optimization in Achiam et al. (2017). Unlike Liu et al. (2023), who apply bi-level optimization in ICRL without strategic exploration, PCSE’s targeted policy constraint offers a novel efficiency boost.

PCSE’s selective exploration aligns with active sampling in Balakrishnan et al. (2020), who use Bayesian optimization to explore reward functions in IRL. The paper advances this by applying it to constraints in ICRL and eliminating generative model reliance, addressing scalability issues noted in Chan and van der Schaar (2021, "Scalable Bayesian Inverse Reinforcement Learning"), where BIRL struggles with large state spaces due to MCMC sampling demands.

**Theoretical Claims:**

* When the minimum cost advantage (the smallest difference in cost between the optimal action and any other action across all state-action pairs in the constrained Markov Decision Process (CMDP)) becomes small, the sample complexity can be indefinitely large. Maybe add an assumption?
* RL solver needs its own sample complexity. This part is missing in the sample complexity.

---

> ### Author Rebuttal · Authors · 2025-04-01
>
> Dear Reviewer p7jo, we sincerely appreciate your constructive feedback and thank you for recognizing the significance of our work. We have carefully considered your suggestions and hope the following response can address your concerns.
>
> > *Q1. Empirical results ..., showing PCSE outperforming baselines ... in terms of rewards, costs, and constraint similarity. One issue is that those baseline algorithms are general-purpose algorithms.*
>
> **A1.** Thank you for raising this concern.
> The four baselines we selected—random, $\varepsilon$-greedy, upper confidence bound, and max-entropy—are well-established and effective exploration methods in RL. In our setting, the exploration strategy prioritizes states requiring frequent visits to improve constraint estimation. Our approach, which takes into account the estimation of unknown dynamics and the expert policy via exploration, is underexplored in ICRL literature. Prior ICRL works typically assume a maximum entropy framework, and we include it as a baseline in our experiments for Gridworld and PointMaze.
>
> ---
> > *Q2. Testing on additional continuous or larger-scale benchmarks could strengthen claims of broader applicability.*
>
> **A2.**
> Thank you for this advice. We have a relevant discussion in Appendix F. We acknowledge that additional experiments on larger-scale continuous benchmarks could offer deeper insights into the practical challenges and opportunities associated with inferring and transferring constraint information. Such experiments would ultimately help guide the development of more robust and scalable algorithms for ICI.
>
> However, it is important to note that sample complexity analysis primarily focuses on discrete state-action spaces [1]. Extending these analyses to continuous spaces presents a significant challenge in the field. Existing algorithms for learning feasible sets [2, 3, 4] struggle when scaling to problems involving large or continuous state spaces. This difficulty arises because their sample complexity is directly tied to the state space size, posing a substantial limitation, particularly since real-world problems often involve large or continuous domains. Continuous environments typically require the use of function approximation techniques, additional assumptions (such as smoothness or linearity), and more sophisticated exploration strategies. Moreover, generalizability remains a key concern, as continuous domains are infinite and depend heavily on effectively approximating value functions, policies, and constraints. We plan to address the development of a scalable approach for sample complexity analysis in future work.
>
> ---
> > *Q3. When the minimum cost advantage (...) becomes small, the sample complexity can be indefinitely large. Maybe add an assumption?*
>
> **A3.** Thank you for your valuable feedback. We agree with the reviewer that adding such an assumption is more rigorous. We additionally assume $\min_{(s,a)}A^{c,\ast}_{\widehat{\mathcal{M}}\cup\tilde{c}}(s,a)\geq\psi>0$, and replace this advantage with $\psi$ in the sample complexity of Theorem 5.6 accordingly. The manuscript has also been revised.
>
> ---
> > *Q4. RL solver needs its own sample complexity. This part is missing in the sample complexity.*
>
> **A4.** Thank you for raising this concern. We have considered the sample complexity of the RL phase in the overall sample complexity for BEAR and PCSE. This can be verified in lines 304-306 (left column) in the main paper and lines 1511-1514 in the proof for Theorem 5.5 in the Appendix (page 28).
>
> To highlight this point, we have revised the relevant paragraph in the manuscript accordingly.
>
> ---
> > *Q5. The reviewer shares some important insights into PCSE from a Bayesian perspective in Relation To Broader Scientific Literature.*
>
> **A5.** Thank you for your valuable feedback. We appreciate the reviewer’s justification of PCSE from a Bayesian perspective. We value this insight and have included these relevant studies in the related work section. The details are not presented here due to 5000 character limit.
>
> ---
> > *Q6. I don't think linear MDPs for future research are particularly interesting, as linear MDPs are almost never practical.*
>
> **A6.** Thank you for this suggestion. The key assumption in a linear MDP is that both the dynamics and rewards are linear with respect to underlying features of the state and action space. We agree with the reviewer that this assumption is strong and, as a result, may not be very practical in real-world scenarios. In response, we have revised the relevant section of the manuscript in Appendix F.
>
> ---
> **References**
>
> [1] Reinforcement learning: Theory and algorithms. CS Dept., UW Seattle, Seattle, WA, USA, Tech. Rep 32 (2019): 96.
>
> [2] Towards theoretical understanding of inverse reinforcement learning. ICML, 2023.
>
> [3] Is inverse reinforcement learning harder than standard reinforcement learning? ICML, 2024.
>
> [4] Offline inverse RL: New solution concepts and provably efficient algorithms. ICML, 2024.

---

### Official Review · Reviewer_fGXy · 2025-03-12

**Overall Recommendation:** 3

**Summary:**

The paper presents a new exploration approach for inverse constrained reinforcement learning (ICRL). in ICRL, the goal is to identify (safety) constraints and a well-performing policy from expert demonstrations resp. an interactive environment. The paper proposes a theoretically motivated way for efficient exploration, i.e. sampling strategies, to learn a good and robust policy.
The proposed method is introduced, formalized and discussed; experiments on several environments are performed to compare the performance against baseline techniques.

**Claims And Evidence:**

- The methods are supported by a theoretical foundation
- BEAR and PCSE and claimed to be more efficient than other baselines, which is maybe supported, but not clear (see comments on experiments below).

**Essential References Not Discussed:**

N/A

**Experimental Designs Or Analyses:**

BEAR and PCSE are evaluated both in multiple discrete and continuous environments and against an expert policy with groundtruth data and four baseline exploration approaches.
The execution and design of the experiments is reasonable and sufficiently broad.
The presentation of the results is, however, a bit strange and mostly done through Figure 3 and a textual interpretation.
It is stated that PCSE (red line) converges much faster and is therefore better than the baselines, but this is not really apparent from the Figure itself. I would have preferred a more thorough evaluation, e.g., is the improvement statistically significant over the other techniques? From the plots it appears as if all techniques perform more or less the same without major differences, but maybe this is an artifact of the presentation and not the results themselves.

Edit after rebuttal: The authors addressed my concerns in their rebuttal and pointed to supplementary information.

**Methods And Evaluation Criteria:**

The selection of discrete and continuous environments are good and sufficiently representative. One can always wish for more, of course, e.g. mujoco with safety constraints or safety-gymnasium, but the selected ones are okay to proof the point of the experiments IMHO.

**Other Comments Or Suggestions:**

N/A

**Other Strengths And Weaknesses:**

I'm glad to see the effort towards provably efficient exploration. Any sound + justified foundation of (inverse) RL is a great step forward.

**Questions For Authors:**

This is not my area and I'm not familiar with the state of the art and prior works.

**Relation To Broader Scientific Literature:**

references to existing works are given and seem reasonable, however I can't say whether they are complete.

**Theoretical Claims:**

The argumentation and motivation for the theoretical claims appear sound, but I have not checked them in detail.

---

> ### Author Rebuttal · Authors · 2025-04-01
>
> Dear Reviewer fGXy, we sincerely appreciate your valuable and constructive comments. We have carefully considered your comments and hope the following responses address your concerns satisfactorily.
>
> > *Q1. ...PCSE (red line) converges much faster and is therefore better than the baselines, but this is not really apparent from Figure 3.  ... is the improvement statistically significant over the other techniques? From the plots, it appears as if all techniques perform more or less the same without major differences, but maybe this is an artifact of the presentation and not the results themselves.*
>
> **A1.** Thank you for raising this concern. We argue that the improvement in PCSE's convergence is significant from several key perspectives.
>
> First, since ICRL recovers safety constraints, both cumulative rewards (top row) and WGIoU (bottom row) are considered valid only after the corresponding cumulative costs of an exploration strategy converge to those of the expert policy. As shown in the middle row of Figure 3, which illustrates the discounted cumulative costs, PCSE (red line) converges faster to the costs of the expert policy (grey line) compared to other baselines.
> In Gridworld 1 and Gridworld 3, the WGIoU score (bottom row) of PCSE further confirms this convergence, which measures the similarity between the recovered and ground-truth constraints. In Gridworld 2 and Gridworld 4, the discounted cumulative rewards (top row) of PCSE further support this convergence.
>
> Second, we present quantitative visualization results in Figures 7, 8, 9, and 10 in the Appendix from page 39 to 42. We do not include them in the main part due to page limit. These figures depict the learned constraints at selected iterations, further demonstrating that PCSE learns feasible constraints faster than the other exploration methods across the four gridworld environments.
>
> Finally, we provide experimental results for comparison of PCSE and two additional baselines, max entropy and upper confidence bound, in Figure 5 in the Appendix (page 37).

---

> > ### Comment · Reviewer_fGXy · 2025-04-04
> >
> > Dear authors,
> >
> > thank you for your response and addressing my concerns. Under consideration of your provided information I raise my score to 3 - weak accept.

---

> > > ### Author Response · Authors · 2025-04-07
> > >
> > > Dear Reviewer fGXy, thank you for your support and guidance in helping us refine our work. Your contributions are truly appreciated.

---

### Official Review · Reviewer_SXpM · 2025-03-15

**Overall Recommendation:** 4

**Summary:**

The authors propose a pair of elegant exploration methods for a variant of the inverse constrained RL problem where one wants to recover the entire set of feasible constraints and provide corresponding sample complexity benefits.


## Update After Rebuttal
The authors added in a discussion of some of the points of confusion I originally had to the paper. I already factored this into my evaluation of the paper, and hence maintain my score

**Claims And Evidence:**

Yes.

**Essential References Not Discussed:**

Could you add in a discussion of https://openreview.net/forum?id=T5Cerv7PT2 and https://arxiv.org/abs/2501.15618 in the paper? Also, there definitely has been prior work on sample complexity in ICRL (e.g. https://arxiv.org/abs/2309.00711), so I would reword the last paragraph in Sec. 2.

**Experimental Designs Or Analyses:**

Yes, they seemed correct but limited.

**Methods And Evaluation Criteria:**

There is fairly limited experimental evaluations -- I think it would be relatively easy to run more thorough experiments.

**Other Comments Or Suggestions:**

- One of the key issues with entropy regularization in ICRL is that it frequently leads to the recovery of constraints that forbid *all* behavior the expert didn't take (as the learner visits all reachable states with nonzero probability). It might be good to mention this in your third paragraph.

- The experiments here are extremely limited. While I don't think it is required, it would of course make things a stronger paper to see if some of these methods could be adapted to more high-dimensional settings, perhaps using open source code like https://github.com/konwook/mticl.

**Other Strengths And Weaknesses:**

- I appreciated the clarity of the justification for the second algorithm's sample complexity benefits over the first (restricting the set of policies consider to only those that could be optimal). It would be good to perhaps repeat this message at other parts of the paper as I thought it was particularly interesting upon reflection.

**Questions For Authors:**

1. Most (if not all) of the prior work you cite in ICRL recovers a single constraint, rather than a set of feasible constraints. Then, when faced with a novel task, it is trivial to figure out which constraint to enforce. This is much less true for set-recovery approaches. Could you comment on this fact or how you would select within the set (potentially reducing the complexity of the overall estimation problem)?

2. There are several assumptions made in this paper that are fairly unusual in the literature, including a deterministic expert, a known constraint tolerance, the expert actually being the constraint-saturating policy, and if I'm reading it correctly, the ability to query the expert (rather than a fixed set of demonstrations). Could you (a) call these out more explicitly / contrast them with the assumptions in work like that of https://arxiv.org/abs/2309.00711 (who also derive some elementary sample complexity results) and (b) explain why I wouldn't want to analysis closer to that of DAgger if I can freely query the expert? The usual justification for the latter is transferring the constraint to new tasks (as otherwise you could directly replay the observed action frequencies and get strong sample complexity guarantees as the expert is assumed to be the optimal safe policy), but I'm not quite sure how to make that argument in this constraint set recovery setting.

3. At heart, RL / CRL are solving a linear program. IRL / ICRL can be seen as solving the inverse of these problems. Could you comment on whether this perspective provides any insights on your results?

4. In prior work, it is common to assume access to a parametric class of functions to avoid "too expressive" constraints that forbid an unnecessarily wide set of expert behavior. Is it possible to adapt your work to this more practical setting (with the current results being a special case with the full set of constraints)? It would be interesting if this would provide tighter sample complexity guarantees (which I suspect it does).

5. This is a bit of a vague question but your analysis bears some similarity to the standard simulation lemma analysis. However, when you have access to an expert, the analysis in https://arxiv.org/abs/1203.1007 and more modern variants like https://arxiv.org/abs/2303.00694 or  https://arxiv.org/abs/2402.08848 are known to provide stronger guarantees. I'd be curious to know if this is also true in the ICRL setting.

**Relation To Broader Scientific Literature:**

Essentially, the authors considered a setting (ICRL) that has received much study before but provided an interesting pair of exploration strategies to solve the problem with tighter sample complexity guarantees than prior approaches.

**Theoretical Claims:**

I read all the theorems and skimmed the proofs -- nothing seemed egregiously wrong to me.

---

> ### Author Rebuttal · Authors · 2025-04-01
>
> Dear Reviewer SXpM, we sincerely value your time and effort in evaluating our work. We appreciate your recognition and have prepared comprehensive responses and clarifications to address each point you raised. We hope these responses can resolve your concerns.
>
> >*Q1. Prior work in ICRL recovers a single constraint ... When faced with a novel task, it is trivial to figure out which constraint to enforce. This is much less true for set-recovery approaches. Could you comment on this fact or how you would select within the set (potentially reducing the complexity of the overall estimation problem)?*
>
> **A1.** Thank you for raising this concern. Compared to prior ICRL works, the set-recovery approach delays selecting specific constraints, enabling analysis of the intrinsic complexity in inverse constraint inference (ICI) problems. Next, we identify two cases of constraint selection when faced with a novel task.
>
> For hard constraints, all constraints in the set are equivalent for a novel task, as the cost function value does not matter ($c(s,a)=1$ and $c(s,a)=2$ both prohibit $(s,a)$). Thus, any feasible constraint can be selected.
>
> For soft constraints, constraints in the set differ for a novel task. The value of each cost function matters due to task differences in dynamics and rewards. Therefore, a generalizable learned constraint should come from the intersection of feasible sets from old and novel tasks. The selection criterion should depend on differences in dynamics and rewards. As a result, visits to some states are no longer necessary given novel task specifications.
>
> ---
> >*Q2. There are several assumptions made in this paper that are fairly unusual in the literature ... Could you (a) call these out more explicitly / contrast them with the assumptions in [3] and (b) explain why I wouldn't want to analysis closer to that of DAgger if I can freely query the expert?*
>
> **A2.** Thank you for this suggestion. We have revised Assumption 4.3 to highlight all these assumptions explicitly.
>
> Next, we distinguish two key differences. First, unlike [3], which assumes the expert policy is safe but not necessarily optimal, we assume it is both safe and optimal. Aligning with a suboptimal safe policy might degrade reward performance, as it might exclude constraints that ensure the safety of a safe-optimal policy since a safe optimal policy makes the best use of constraint tolerance while suboptimal safe policies might not. Second, we adopt an online setting for flexibility and real-time adaptation, while [3] adopts an offline setting. We acknowledge that offline setting is an intriguing research direction for ICRL.
>
> For (b), could the reviewer explain what is DAgger in part (b)?
>
> ---
> >*Q3. At heart, RL / CRL solves a linear program. IRL / ICRL solves the inverse of these problems. Does this perspective provide any insights into your results?*
>
> **A3.** Thank you for this advice. In essence, ICRL alternates between updating an imitating policy with CRL and learning constraints via ICI until the imitation policy reproduces expert demonstrations. In our setting, the estimation of the expert policy and dynamics reproduces expert demonstrations, while constraints are inferred through subsequent updates of advantage functions.
>
> ---
> >*Q4. Prior work commonly assumes access to a parametric class of functions to avoid "too expressive" constraints that forbid an unnecessarily wide set of expert behavior. Is it possible to adapt your work to this more practical setting (with the current results being a special case with the full set of constraints)? It would be interesting if this would provide tighter sample complexity guarantees (which I suspect it does).*
>
> **A4.** Thank you for this advice. We beg to differ that we parametrically define the feasible cost set in Lemma 4.5 as
> $c=A^{r,\pi^E}_ {\mathcal{M}}\zeta+(E-\gamma P_{\tau})V^c$, where $\zeta$ and $V^c$ can alter. In addition, we avoid "too expressive" constraints by not penalizing $(s,a)$ with fewer rewards than the expert in case (iii) in Lemma 4.4.
>
> ---
> >*Q5. 1) Discuss [1,2,3] and reword the last paragraph in Sec.2. 2) Repeat the message of 'constraining candidate policies' in PCSE. 3) Mention the drawbacks of entropy regularization in ICRL in the 3rd paragraph. 4) Can [4-6] adapt to ICRL settings?*
>
> **A5.** Thanks for your valuable feedback. We have revised the paper accordingly. The details are not presented here due to 5000 rebuttal character limit.
>
> ---
> **References**
>
> [1] Simplifying constraint inference with inverse reinforcement learning. NeurIPS, 2024.
>
> [2] Your learned constraint is secretly a backward reachable tube. arXiv:2501.15618.
>
> [3] Learning shared safety constraints from multi-task demonstrations. NeurIPS, 2023.
>
> [4] Agnostic system identification for model-based reinforcement learning. ICML, 2012.
>
> [5] The virtues of laziness in model-based RL: a unified objective and algorithms. ICML, 2023.
>
> [6] Hybrid inverse reinforcement learning. ICML, 2024.

---

> > ### Comment · Reviewer_SXpM · 2025-04-06
> >
> > Hi,
> >
> > 1. It might be good to add a note on how one could use this recovered hard / soft constraint set for a downstream optimization procedure to the discussion section of the paper.
> >
> > 2. I'd argue that needing to assume the data is generated by an expert who is both safe and optimal is a strong assumption. As you say, it'd of course be better if we had access to such data. That said, it might be good to both (a) explicitly note this assumption and (b) note that relaxing it would be interesting for future work. Re: DAgger -- I think I mean the same thing as you mean by online / offline: that you can't query the expert for their action distribution at an arbitrary state. My point is that this isn't a standard assumption in ICL, so being explicit about this fact is important, so I'd suggest the same two points as above.
> >
> > 3. I know what ICL is :). My point is that a lot of the ICL machinery really boils down to looking at the linear program of constrained RL, considering the inverse problem, and then making linear algebraic statements about this inverse problem. For example, this is where the affine subspace you derive is actually coming from, right? It might be fun to think about that perspective more, I've found it illuminating when thinking about ICL.
> >
> > 4. So, often times, we don't want to be able to penalize an arbitrary $(s, a)$ pair in practice for ICL as it leads to overly restrictive constraint and instead consider doing inference over some function class $\mathcal{F}$ that incorporates some prior knowledge about what are the sorts of $(s, a)$ we want to forbid in the first place. I was noting that rather than searching over some affine subspace that is mostly a function of the MDP's dynamics, it would be closer to practice to consider restricting the set of functions you're searching over. It might be worth mentioning this in the discussion section as future work if there's space.

---

> > > ### Author Response · Authors · 2025-04-07
> > >
> > > Dear Reviewer SXpM, thank you for providing additional constructive feedback on our paper. We greatly appreciate the time and effort you dedicated to refining our work. Your insights have been crucial in guiding our revisions, and we will carefully incorporate your suggestions into the final version to improve its quality.
> > >
> > > **A-1.** Thanks for this advice. We agree with the reviewer on this point.
> > > We have added a note that discusses this point to the revised manuscript.
> > >
> > > ---
> > >
> > > **A-2.** Thank you for this comment and for providing further explanations. In the revised manuscript, 1) we have explicitly stated the assumption of access to a safe and optimal expert policy in Assumption 4.3; 2) We have also pointed out that relaxing it (either to a safe expert policy or offline expert demonstrations) would be an interesting direction for future work. We believe it is also valuable to investigate how the sub-optimality of expert agents influences constraint inference and transferability.
> > >
> > > ---
> > >
> > > **A-3.** Thanks for this feedback. We agree with the reviewer on this point. Linear algebraic analyses are indeed more rigorous and inherent, and thus worth investigating for the ICL. For instance, by defining a subspace $\mathcal{U} = \mathrm{im}(E-\gamma{P}_{\mathcal{T}})$, cost functions in a feasible cost set are equivalent on the quotient space $\mathbb{R}^{\mathcal{S}\times\mathcal{A}}/\mathcal{U}$. Furthermore, we can measure the distance between the recovered and expert costs within this quotient space. A discussion of this has been included in the revised manuscript.
> > >
> > > ---
> > >
> > > **A-4.** Thank you for your valuable advice. As noted in [3], the ground-truth constraints can be recovered within a multi-task framework by limiting $\mathcal{F}$ to certain function classes, such as DNNs or functions based on state observations. We agree with the reviewer that restricting the set of constraint functions is more practical than using state-action-wise penalties, making it a compelling direction for future research. Additionally, we recognize that employing a multi-task setting leads to a more generalizable constraint that is closer to the ground truth. In the set-recovery approach, this can be achieved by intersecting multiple feasible cost sets within tasks that are sufficiently distinct from one another. In the revised version, we have included a discussion of this point in the conclusion section.
> > >
> > > ---
> > >
> > > We have also incorporated the revisions from A5 in the previous rebuttal. Thank you again for your insightful guidance!

---

### Decision · Program_Chairs · 2025-05-01

**Decision:**

Accept (poster)

**Comment:**

All three reviewers recommended the acceptance of paper and appreciated the authors' responses to their questions and concerns. The theoretical contributions are novel and interesting, and the proposed algorithms show advance our current approaches to Inverse Constrained Reinforcement Learning (ICRL). For a paper with a strong theoretical focus, the empirical evaluation is adequate but could benefit from being more comprehensive overall. Based on the above, I recommend accepting the paper.
To improve the paper, the authors should explicitly state and discuss the strong assumptions made in the theoretical analysis (as already indicated in the rebuttal). Additionally, they should incorporate a more detailed discussion on the implications of these assumptions and potential directions for relaxing them in future work.